# Near-Optimal and Efficient First-Order Algorithm for Multi-Task Learning with Shared Linear Representation

**Shihong Ding** [1]  **Fangyu Du** [2]  **Cong Fang** [1]

## Abstract

Multi-task learning (MTL) has emerged as a pivotal paradigm in machine learning by leveraging shared structures across multiple related tasks. Despite its empirical success, the development of likelihood-based efficiently solvable algorithms—even for shared linear representations—remains largely underdeveloped, primarily due to the non-convex structure intrinsic to matrix factorization. This paper introduces a first-order algorithm that jointly learns a shared representation and task-specific parameters, with guaranteed efficiency. Notably, it converges in $\widetilde{\mathcal{O}}(1)$ iterations and attains a *near-optimal* estimation error of $\widetilde{\mathcal{O}}(dk/(TN))$, *improving* over existing likelihood-based methods by a factor of $k$, where $d$, $k$, $T$, $N$ denote input dimension, representation dimension, task count, and samples per task, respectively. Our results justify that likelihood-based first-order methods can efficiently solve the MTL problem.

## 1. Introduction

Multi-task learning (MTL) has become a fundamental paradigm in machine learning (Argyriou et al., 2008; Liu et al., 2007; Maurer et al., 2012; Zhang, 2013; Zhao et al., 2025b), addressing a core challenge of real-world intelligence: the ability to learn multiple related tasks while leveraging their inherent commonalities. Unlike isolated single-task models, MTL frameworks are designed to exploit a shared structure that are learned using whole data across the tasks, thereby improving data efficiency. Related paradigms including transfer learning (TL) and meta-learning (Maurer et al., 2016; Finn et al., 2017; Snell et al., 2017; Aqeel

et al., 2025; Liu et al., 2025) also build on leveraging shared structures that are then transferred to new tasks to improve generalization. Among various MTL and TL approaches, a central and widely adopted strategy is to assume that all tasks share a common or similar underlying representation, on which predictions for each task are based. This formulation transforms the problem into the joint learning of both a shared representation function and individual task-specific predictors. Gradient descent based algorithm applied to the corresponding log-likelihood objective is the most common approach for solving the problem. In neural network models, this is typically implemented through forward and backward propagation, a procedure central to modern deep learning practice. The effectiveness of shared representation learning has been empirically witnessed across diverse domains, including object detection (Zhang et al., 2014), language understanding evaluation (Wang et al., 2018), affective computing (Kollias et al., 2024), and drug design (Allenspach et al., 2024).

On the theoretical side, most studies focus on establishing statistical guarantees for multi-task representation learning. Pioneering works by Baxter (2000), Ando et al. (2005), and Maurer et al. (2016), for example, analyze general function classes and demonstrate that the statistical error for the shared function depends on samples from all tasks, thereby offering provable generalization improvements in various settings. Subsequent studies (Agarwal et al., 2020; Cheng et al., 2022; Collins et al., 2024) examine more specific function classes—such as linear or neural network representations—and derive improved or near-optimal sample complexity bounds. Another line of research explores more complex frameworks, including models with similar rather than identical representations, multi-task reinforcement learning in Markov decision processes, and distributed learning settings.

In machine learning, we value not only statistical efficiency but also the availability of efficient and practical algorithms to solve a given problem. However, the solvability of multi-task representation learning—even for shared linear representations—remains largely open. This challenge stems from its underlying mathematical structure: the joint learning of a shared representation and task-specific predictors

---

[1]State Key Lab of General AI, School of Intelligence Science and Technology, Peking University, China [2]School of Mathematical Sciences, Peking University, China. Correspondence to: Cong Fang <fangcong@pku.edu.cn>.

*Proceedings of the 43rd International Conference on Machine Learning*, Seoul, South Korea. PMLR 306, 2026. Copyright 2026 by the author(s).

*Table 1.* A comparison of different methods for Linear MTL in high-dimensional regime. In the table, the "Algorithm Type" column uses the following abbreviations: MBM for moment-based method, SM for spectral method, and LBM for likelihood-based method. "IC (Iteration Complexity)" refers only to the outer-loop count. Here, "-" indicates that the algorithm lacks a convergence guarantee and computational analysis. "AC (Additional Per-Iteration Cost)" notes if there are significant inner-loop operations, such as SVD or QR decomposition.

| Algorithm | $\frac{1}{T}\sum_{t=1}^{T}\|\widehat{\mathbf{v}}_t - \mathbf{v}_t^*\|^2$ | IC | AC | Algorithm Type |
|---|---|---|---|---|
| MoM (Tripuraneni et al., 2021) | $\widetilde{\mathcal{O}}\left(\frac{dk^2}{NT}\right)$ | $\widetilde{\mathcal{O}}(1)$ | SVD | MBM |
| MoM (Niu et al., 2024) | $\widetilde{\mathcal{O}}\left(\frac{dk}{NT}\right)$ | $\widetilde{\mathcal{O}}(1)$ | SVD | MBM |
| SE (Tian et al., 2025) | $\widetilde{\mathcal{O}}\left(\frac{dk}{NT}\right)$ | $\widetilde{\mathcal{O}}(1)$ | SVD | SM |
| AltMinGD (Thekumparampil et al., 2021) | $\widetilde{\mathcal{O}}\left(\frac{dk^2}{NT}\right)$ | $\widetilde{\mathcal{O}}(1)$ | QR | LBM |
| ERM (Du et al., 2021) | $\mathcal{O}\left(\frac{dk^2}{NT}\right)$ | - | - | LBM |
| ARMUL (Duan & Wang, 2023) | $\mathcal{O}\left(\frac{dk^2}{NT}\right)$ | - | - | LBM |
| pERM (Tian et al., 2025) | $\mathcal{O}\left(\frac{dk^2}{NT}\right)$ | - | - | LBM |
| TPGD (This Work) | $\widetilde{\mathcal{O}}\left(\frac{dk}{NT}\right)$ | $\widetilde{\mathcal{O}}(1)$ | - | LBM |
| **Lower Bound** (Tripuraneni et al., 2021; Tian et al., 2025) | $\Omega\left(\frac{dk}{NT}\right)$ | - | - | - |

typically takes the form of a matrix factorization problem, which is inherently non-convex and often computationally difficult to solve globally. For linear representation learning, the first work to study the solvability through efficient algorithms is Tripuraneni et al. (2021). It examines the global convergence of first-order methods based on the likelihood function. However, this analysis relies on a technical assumption that the point of convergence is an interior point of a predetermined region. Although Thekumparampil et al. (2021) further proposes a likelihood-based method with a simpler update rule, the resulting estimation error bound $\widetilde{\mathcal{O}}(\frac{dk^2}{NT})$ remains sub-optimal, as the known information-theoretic lower bound for this setting is $\widetilde{\mathcal{O}}(\frac{dk}{NT})$ (Tripuraneni et al., 2021; Tian et al., 2025), where $d, k, T, N$ denote input dimension, representation dimension, task count, and samples per task, respectively. To ensure computational tractability and achieve optimal sample complexity, subsequent research has turned to methods of moments (Niu et al., 2024) or spectral approaches (Tian et al., 2025), avoiding the non-convex matrix factorization structure. However, these types of methods often rely heavily on problem-specific structures, such as noise assumption, and require careful algorithmic design.

In contrast, likelihood-based methods (LBM), which typically adopt maximum likelihood estimation as the training objective, possibly augmented with regularization, are among the most widely used approaches in machine learning practice. Within the problem, applying first-order algorithm over the factorization structure in order to jointly learn representations and predictors is ubiquitous in machine learning. However, it remains an open challenge to *design a computationally efficient likelihood-based algorithm that attains statistical optimality, even for the linear representation setting*.

This paper focuses on this canonical high-dimensional MTL with low-rank linear representations (Linear MTL) and designs the *first* first-order likelihood-based algorithm that jointly learns a shared representation and task-specific parameters, with guaranteed computational and statistical efficiency. Specifically, consider $T$ tasks and $N$ samples $\{(\mathbf{x}_t^{(i)}, y_t^{(i)})\}_{i\in[N]}$ on each task $t \in [T]$. For task $t \in [T]$, with $\mathbf{x}_t^{(i)} \in \mathbb{R}^d$ being the covariate and $y_t^{(i)}$ being its response, we assume a linear relationship:

$$y_t^{(i)} = \left\langle \mathbf{x}_t^{(i)}, \mathbf{v}_t^* \right\rangle + z_t^{(i)},$$

where $\mathbf{v}_t^* \in \mathbb{R}^d$ is the task specific parameter and $z_t^{(i)} \in \mathbb{R}$ is an additive noise. We suppose a low-dimensional structure on the parameters, where there is an matrix $\mathbf{B}^* \in \mathbb{R}^{d\times k}$ with $k \le T \le d$ and vectors $\mathbf{w}_t^* \in \mathbb{R}^k$ such that $\mathbf{v}_t^* = \mathbf{B}^*\mathbf{w}_t^*$ for all $t$. Here $\mathbf{B}^*$ is the shared low-dimensional representation and $\mathbf{w}_t^*$ is the task-specific parameter for task $t$. The goal is to estimate $(\mathbf{v}_1^*, \cdots, \mathbf{v}_T^*)$.

The main novelty of our algorithmic design is a two-phase gradient descent (TPGD) algorithm. *The first phase (warm-start)* performs unregularized gradient descent on the negative log-likelihood function

$$\widehat{\mathcal{L}}\left(\mathbf{B}, \mathbf{W}\right) := \frac{1}{2N} \sum_{t=1}^{T} \sum_{i=1}^{N} \left(y_t^{(i)} - \left\langle \mathbf{x}_t^{(i)}, \mathbf{B}\mathbf{w}_t \right\rangle\right)^2, \quad (1)$$

where $\mathbf{W} = (\mathbf{w}_1, \cdots, \mathbf{w}_t)$. In this phase, the algorithm performs feature learning and ensures that, starting from random initialization, the parameters $\mathbf{B}$ and $\mathbf{W}$ converge to a small neighborhood of a global optimum within $\widetilde{\mathcal{O}}(1)$ iterations, assuming the condition number of the objective matrix is constant. This establishes that the algorithm can overcome the non-convexity barrier inherent in the objective function. *The second phase (refined optimization with correction)* performs regularized gradient descent to guarantee statical efficiency with the following update rules:

$$\begin{cases} \mathbf{B}^{(\tau+1)} = \mathbf{B}^{(\tau)} - \eta \nabla_{\mathbf{B}} \widetilde{\mathcal{L}}\left(\mathbf{B}^{(\tau)}, \mathbf{W}^{(\tau)}\right), \\ \\ \mathbf{W}^{(\tau+1)} = \mathbf{W}^{(\tau)} - \eta \nabla_{\mathbf{W}} \widetilde{\mathcal{L}}\left(\mathbf{B}^{(\tau)}, \mathbf{W}^{(\tau)}\right), \end{cases} \quad (2)$$

where $\widetilde{\mathcal{L}}$ denotes the new loss formed by augmenting $\widehat{\mathcal{L}}$ with a regularization term, as defined in Eq. (5).

Consequently, the proposed algorithm has an iteration complexity of only $\widetilde{\mathcal{O}}(1)$, and achieves a convergence rate of $\widetilde{\mathcal{O}}(\frac{dk}{NT})$ for estimation error $\frac{1}{T} \sum_{t=1}^{T} \|\widehat{\mathbf{v}}_t - \mathbf{v}_t^*\|^2$. This rate *matches* the known lower bound up to logarithmic factors, *improving* over existing likelihood-based methods by a factor of $k$. See Table 1 for a comparison of different algorithms on linear MTL.

We would emphasize that achieving the $\widetilde{\mathcal{O}}(1)$ iteration complexity and $\widetilde{\mathcal{O}}(dk)$ sample complexity for this matrix factorization problem is technically nontrivial. A closely related setting is matrix sensing, where the goal is to recover a low-rank matrix from linear measurements. Under the factorization formulation, the best known first-order algorithms achieve iteration complexity of $\mathcal{O}(\sqrt{k})$ and sample complexity of $\mathcal{O}(dk^2)$ for symmetric ground truth, and $\mathcal{O}(d^8)$ and $\mathcal{O}(dk^3)$ for the general case. It remains an open question whether statistical optimality can be attained via first-order methods. In contrast, linear MTL differs from matrix sensing notably in its noise structure. In this work, we attain the optimal estimation complexity within $\widetilde{\mathcal{O}}(1)$ iterations. The key novelty in our theoretical analysis lies in our examination of the feature learning capability in *Phase I* and the rapid convergence property in *Phase II*.

**Our Contributions.** The contributions of this paper are as follows:

(1) We propose a two-stage first-order likelihood-based algorithm for Linear MLT.

(2) The algorithm is the *first* to achieve computational efficiency with a *near-optimal* estimation error of $\widetilde{\mathcal{O}}(dk/(TN))$, *improving* over existing likelihood-based methods by a factor of $k$.

**Notations.** We denote real vectors by bold lowercase letters (e.g., $\mathbf{x}, \mathbf{y}$) and real matrices by bold uppercase letters (e.g., $\mathbf{X}, \mathbf{Y}$). A matrix $\mathbf{W}$ in $\mathbb{R}^{m \times n}$ is denoted as $\mathbf{W} = (W_{ij})_{m \times n}$. We use $\mathbf{W}^\dagger$ to denote the Moore–Penrose pseudoinverse of $\mathbf{W}$. For $n \in \mathbb{N}_+$, the set $\{1, \cdots, n\}$ is written as $[n]$. For functions $f, g : \mathbb{R} \to \mathbb{R}$, we write $f \lesssim g$ for $f = \widetilde{\mathcal{O}}(g)$; that is, there exists a constant $C > 0$ such that $f(n) \leq C \cdot \operatorname{poly} \log(n) \cdot g(n)$ for all $n$.

The standard inner product is denoted by $\langle \cdot, \cdot \rangle$. For vectors $\mathbf{u}, \mathbf{v} \in \mathbb{R}^n$, $\langle \mathbf{u}, \mathbf{v} \rangle = \sum_{i=1}^{n} u_i v_i$. For matrices $\mathbf{X}, \mathbf{Y} \in \mathbb{R}^{m \times n}$, we define $\langle \mathbf{X}, \mathbf{Y} \rangle = \operatorname{tr}(\mathbf{X}^\top \mathbf{Y})$. The $\ell_2$-norm of a vector $\mathbf{v} \in \mathbb{R}^n$ is denoted by $\|\mathbf{v}\|$. For a matrix $\mathbf{X} \in \mathbb{R}^{m \times n}$, the operator norm is defined as $\|\mathbf{X}\|_{\text{op}} := \max_{\mathbf{v} \in \mathbb{R}^n, \|\mathbf{v}\|=1} \|\mathbf{X}\mathbf{v}\|$ and the Frobenius norm is denoted by $\|\mathbf{W}\|_{\text{F}}$. The $i$-th largest singular value of $\mathbf{X}$ is denoted by $\sigma_i(\mathbf{X})$ for any $i \in [\min\{m, n\}]$, and the condition number of $\mathbf{X}$ is defined as $\kappa(\mathbf{X}) := \sigma_1(\mathbf{X})/\sigma_{\text{rank}(\mathbf{X})}(\mathbf{X})$. For a matrix $\mathbf{U} \in \mathbb{R}^{m \times n}$ and an integer $k < n$, the notation $\mathbf{U}(\cdot, [k]) \in \mathbb{R}^{m \times k}$ denotes the matrix comprising the first $k$ columns of $\mathbf{U}$. For a diagonal matrix $\boldsymbol{\Sigma} \in \mathbb{R}^{m \times n}$ and an integer $k < \min\{m, n\}$, $\boldsymbol{\Sigma}_{[k]}$ represents the top-left $k \times k$ submatrix of $\boldsymbol{\Sigma}$. Additionally, we define the distance between two matrices $\mathbf{U}, \mathbf{V} \in \mathbb{R}^{d \times k}$ as:

$$\operatorname{dist}(\mathbf{U}, \mathbf{V}) := \min_{\substack{\mathbf{R} \in \mathbb{R}^{k \times k} \\ \mathbf{R}^\top \mathbf{R} = \mathbf{I}_k}} \|\mathbf{U} - \mathbf{V}\mathbf{R}\|_{\text{F}}.$$

## 2. Related Work

**Multi-Task Learning and Transfer Learning.** The goal of MTL is to improve the accuracy of individual task models through joint learning across multiple related tasks (Argyriou et al., 2008; Maurer et al., 2012; Zhang, 2013; Liu et al., 2007; Wilson et al., 2007; Chen et al., 2010). And the goal of TL is to transfer knowledge from one task to another. For theoretical analysis of MTL and TL, Baxter (2000) studies a model under general function classes where tasks with shared representations are sampled from the same underlying environment, which was later improved in subsequent works (Pontil & Maurer, 2013; Maurer et al., 2016). Ando et al. (2005) proposes a framework to learn a shared low-dimensional predictive representation from multiple tasks. Under linear settings, Du et al. (2021); Duan & Wang (2023) provide the generalization ability of ERM for the likelihood function, and Thekumparampil et al. (2021) designs a polynomial-time alternating gradient descent algorithm that achieves similar performance as ERM but avoids solving the non-convex optimization directly. Tripuraneni et al. (2021); Duchi et al. (2022); Niu et al. (2024) propose

the solvable moment methods. Tian et al. (2025) design spectral methods via SVD with task contamination. Other studies have extensively examined loss error bounds for complex MTL representation models, such as those based on Markov decision processes (Agarwal et al., 2020; Cheng et al., 2022) and two-layer ReLU neural networks (Collins et al., 2024). (Jacob et al., 2008; Zhou et al., 2011) study models with similar rather than identical representations, while (Smith et al., 2017; Marfoq et al., 2021; Corinzia et al., 2019) investigate MTL under distributed learning setting.

**Matrix Sensing.** It's a classical problem aimed at recovering a low-rank matrix from measurement data. Due to its non-convex nature and its connection to key phenomena in deep learning theory (Li et al., 2018; 2020; Arora et al., 2019; Soltanolkotabi et al., 2023; Xiong et al., 2023; Jin et al., 2023), matrix sensing also serves as an important testbed for studying the convergence behavior of deep learning models. Specifically, Li et al. (2018; 2020); Arora et al. (2019); Soltanolkotabi et al. (2023) demonstrate that gradient descent exhibits an implicit regularization effect when training over-parameterized matrix factorization models, enabling convergence to the underlying low-rank matrix that generated the measurements. Furthermore, Xiong et al. (2023) proves that an asymmetric parameterization can achieve exponential acceleration in convergence. At the population level, the likelihood functions of the matrix sensing problem and linear MTL are identical, indicating a structural correlation between them. There exists linear MTL study leverage this connection. For instance, building on Recht et al. (2010)'s technique for solving nuclear norm minimization problem, Zhang et al. (2024) develops a method that learns the task-invariant subspace to address data scarcity in linear MTL. However, the noise structures of matrix sensing problem and linear MTL differ fundamentally: the former arises from random measurement matrices, while the latter originates from random observation vectors per task. Consequently, linear MTL and matrix sensing exhibit an essential difference in their computational complexity and sample complexity. The algorithmic design and theoretical analysis in this work are inspired by literature in matrix sensing (Tu et al., 2016; Xiong et al., 2023). However, our method exhibits substantial advantages in complexity, requiring only $\widetilde{\mathcal{O}}(1)$ computational complexity and $\widetilde{\mathcal{O}}(dk)$ sample complexity, compared to $\widetilde{\mathcal{O}}(\mathrm{poly}(d))$ and $\widetilde{\mathcal{O}}(dk^2)$ for existing approaches, respectively.

## 3. Problem Formulation

**Setup.** This work investigate a setting with $T$ tasks and high-dimensional condition $d > T$. Each task $t \in [T]$ is associated with the data space $\mathcal{X} \times \mathcal{Y}$, where $\mathcal{X} \subseteq \mathbb{R}^d$ denotes the covariate space and $\mathcal{Y} \subseteq \mathbb{R}$ denotes the response space. For each task $t \in [T]$, we have access to $N$ samples:

$(\mathbf{x}_t^{(1)}, y_t^{(1)}), \cdots, (\mathbf{x}_t^{(N)}, y_t^{(N)})$. For notational convenience, we define the covariate data matrix $\mathbf{X}_t := (\mathbf{x}_t^{(1)}, \cdots, \mathbf{x}_t^{(N)})$ and response data vector $\mathbf{y}_t := (y_t^{(1)}, \cdots, y_t^{(N)})$ for each task $t$. In this work, the common feature is characterized by a linear representation function $\phi : \mathcal{X} \to \mathcal{Z} \subseteq \mathbb{R}^k$, where $k < \min\{d, T\}$. Specifically, for each task $t \in [T]$, we define the predictive function as $\mathbf{x} \to \langle \phi(\mathbf{x}), \mathbf{w}_t \rangle := \langle \mathbf{x}_t, \mathbf{B}\mathbf{w}_t \rangle$, where $\mathbf{B} \in \mathbb{R}^{d \times k}$ and $\mathbf{w}_t \in \mathbb{R}^k$. Let $\mathbf{W}$ denote the matrix formed by aggregating all task-specific weight vectors, i.e., $\mathbf{W} = (\mathbf{w}_1, \ldots, \mathbf{w}_T) \in \mathbb{R}^{k \times T}$. Using training samples from the $T$ tasks, the representation and task-specific parameters can be learned by solving the following optimization problem:

$$\min_{\substack{\mathbf{B} \in \mathbb{R}^{d \times k} \\ \mathbf{W} \in \mathbb{R}^{k \times T}}} \left\{ \widehat{\mathcal{L}}(\mathbf{B}, \mathbf{W}) = \frac{1}{2N} \sum_{t=1}^{T} \left\| \mathbf{y}_t - \mathbf{X}_t^\top \mathbf{B}\mathbf{w}_t \right\|^2 \right\}. \quad (3)$$

Let $\widehat{\mathbf{B}}$ and $\widehat{\mathbf{W}}$ denote the representation and task-specific matrix obtained by numerically solving the optimization problem in Eq. (3). We first address whether the predictor $\widehat{\mathbf{B}}\widehat{\mathbf{W}}$ achieves a strong recover of the ground-truth $\mathbf{B}^*\mathbf{W}^* := (\mathbf{v}_1^*, \cdots, \mathbf{v}_T^*)$. Then we apply the learned representation $\widehat{\mathbf{B}}$ to a target task $T + 1$. Suppose the target task is defined by a distribution $\rho$ over $\mathcal{X} \times \mathcal{Y}$, from which we draw $M$ i.i.d. samples: $(\mathbf{x}_{T+1}^{(1)}, y_{T+1}^{(1)}), \cdots, (\mathbf{x}_{T+1}^{(M)}, y_{T+1}^{(M)})$. Using $\widehat{\phi}$, we train a linear predictor by solving the following optimization problem:

$$\min_{\mathbf{w}_{T+1} \in \mathbb{R}^k} L_\rho(\widehat{\mathbf{B}}, \mathbf{w}_{T+1}),$$

where $L_\rho(\widehat{\mathbf{B}}, \mathbf{w}_{T+1}) := \frac{1}{2}\mathbb{E}_{(\mathbf{x}_{T+1}, y_{T+1}) \sim \rho}[(y_{T+1} - \langle \mathbf{x}_{T+1}, \widehat{\mathbf{B}}\mathbf{w}_{T+1}\rangle)^2]$. Let $\widehat{\mathbf{w}}_{T+1}$ denote the numerical solution to this problem. We analyze the generalization performance of the predictor $\widehat{\mathbf{B}}\widehat{\mathbf{w}}_{T+1}$ to the target task. Specifically, we aim to ensure that the population loss $L_\rho(\widehat{\mathbf{B}}, \widehat{\mathbf{w}}_{T+1})$ is sufficiently small.

**Data Assumption.** We assume the existence of a ground-truth representation $\mathbf{B}^* \in \mathbb{R}^{d \times k}$ and task-specific parameters $\mathbf{w}_1^*, \cdots, \mathbf{w}_T^*, \mathbf{w}_{T+1}^* \in \mathbb{R}^k$, such that for every task $t \in [T + 1]$, the equality $\mathbb{E}[y_t|\mathbf{x}_t] = \langle \mathbf{x}_t, \mathbf{B}^*\mathbf{w}_t^* \rangle$ holds. More specifically, we assume that for each task $t \in [T + 1]$, the response $y_t$ given the covariate $\mathbf{x}_t$ is generated as follows:

$$y_t = \langle \mathbf{x}_t, \mathbf{B}^*\mathbf{w}_t^* \rangle + z_t, \qquad z_t \sim \mathcal{N}(0, \sigma^2), \quad (4)$$

where $\mathbf{x}_t$ and $z_t$ are independent for all $t \in [T + 1]$. For the covariate data matrices $\{\mathbf{X}_t\}_{t=1}^T$, we require them to satisfy the restricted isometry property (RIP). Before introducing specific assumptions, we first present the definition of RIP.

**Definition 3.1** (Restricted Isometry Property (RIP))**.** We say that a matrix $\mathbf{X}/\sqrt{N} \in \mathbb{R}^{d \times N}$ satisfies the $\delta$-RIP if for

every vector $\mathbf{v} \in \mathbb{R}^d$, the following inequality holds:

$$(1-\delta)\|\mathbf{v}\|^2 \leq \left\|\frac{\mathbf{X}^\top \mathbf{v}}{\sqrt{N}}\right\|^2 \leq (1+\delta)\|\mathbf{v}\|^2.$$

During the learning process for the representation function, we state the following assumption:

**Assumption 3.2.** There exists a constant $\delta \in (0,1)$ such that $\mathbf{X}_t/\sqrt{N} \in \mathbb{R}^{d \times N}$ satisfies the $\delta-$RIP for any $t \in [T]$.

The RIP is widely used in matrix sensing (Li et al., 2018; Jin et al., 2023; Xiong et al., 2023) and over-parameterized regression (Vaskevicius et al., 2019; Woodworth et al., 2020). The setting considered in Tripuraneni et al. (2021), where the covariate data $\{\mathbf{x}_i\}$ is i.i.d. and sub-Gaussian, is in fact a special case of Assumption 3.2. Based on Candès & Plan (2011), if each entry of $\mathbf{X}_t$ follows an i.i.d. $\mathcal{N}(0, 1/N)$ distribution for all $t \in [T]$, then $\mathbf{X}_t/\sqrt{N}$ satisfies the $\delta$-RIP condition when $N = \widetilde{\Omega}(d/\delta^2)$.

Below, we provide assumptions for task $T + 1$.

**Assumption 3.3.**

**[A₁]** Denote $\mathbf{H} := \mathbb{E}\left[\mathbf{x}_{T+1}\mathbf{x}_{T+1}^\top\right]$, and assume that $\mathbf{H}$ is strictly positive definite.

**[A₂]** There exists a constant $\alpha > 0$, such that for every positive semi-definite (PSD) matrix $\mathbf{M}$, the following inequality holds:

$$\mathbb{E}\left[\mathbf{x}_{T+1}\mathbf{x}_{T+1}^\top \mathbf{M}\mathbf{x}_{T+1}\mathbf{x}_{T+1}^\top\right] \preceq \alpha \langle \mathbf{H}, \mathbf{M}\rangle \cdot \mathbf{H}.$$

Assumption 3.3 is satisfied if $\mathbf{H}^{-1/2}\mathbf{x}_{T+1} \sim \mathcal{N}(\mathbf{0}, \mathbf{I_d})$, in which case $\alpha = 3$. More generally, the assumption also holds for any covariate data $\mathbf{x}_{T+1}$ whose kurtosis is bounded along every direction (Dieuleveut et al., 2017). It is a relatively weak moment condition and has been commonly used in analyses of stochastic gradient descent (SGD) (Dieuleveut et al., 2017; Wu et al., 2022; Zhao et al., 2025a). Existing theoretical guarantees for TL (Tripuraneni et al., 2021; Du et al., 2021; Niu et al., 2024) typically require the covariate $\mathbf{x}_{T+1}$ to be sub-Gaussian. In this work, we relax the assumption to a weaker moment condition specified in Assumption 3.3.

**Estimation Error.** For MTL, our focus is the estimation error $\frac{1}{T}\sum_{t=1}^{T}\|\widehat{\mathbf{v}}_t - \mathbf{v}_t\|^2$, same to Tian et al. (2025). Under the RIP condition (Candes & Tao, 2005; Recht et al., 2010) and mild moment assumption, this error bound also corresponds to the population loss $\mathbb{E}_{\{(\mathbf{X}_t,\mathbf{y}_t)\}_{t\in[T]}}[\frac{1}{T}\widehat{\mathcal{L}}(\widehat{\mathbf{B}}, \widehat{\mathbf{W}})]$. For TL, we focus on the excess risk $L_\rho(\widehat{\mathbf{B}}, \mathbf{w}_{T+1})$, same to (Du et al., 2021; Tripuraneni et al., 2021).

For clarity, we restate some key symbols: $d, k, T, N$ denote the input dimension, representation dimension, task count,

---

**Algorithm 1** Two Phase Gradient Descent (TPGD)

**Input:** Initial parameters: $\mathbf{B}^{(0)} = \widetilde{\alpha} \cdot \widetilde{\mathbf{B}}_0$ and $\mathbf{W}^{(0)} = (\widetilde{\alpha}/3) \cdot \widetilde{\mathbf{W}}_0$, where $\widetilde{\mathbf{B}}_0 \in \mathbb{R}^{d \times k}$ and $\widetilde{\mathbf{W}}_0 \in \mathbb{R}^{k \times T}$ whose entries are $i.i.d.\, \mathcal{N}(0, 1/d)$. Step-sizes: $\eta_1$ for the first phase and $\eta_2$ for the second phase. Iteration counts: $K_1$. Total sample size for each task: $N$.
**Output:** $\widehat{\mathbf{B}} = \mathbf{B}^{(K_1)} \in \mathbb{R}^{d \times k}, \widehat{\mathbf{W}} = \mathbf{W}^{(K_1)} \in \mathbb{R}^{k \times T}$.

1: **while** $\tau < K_1/2$ **do**
2:     Running *Phase I* Iteration: Gradient descent step on the loss $\widehat{L}$ (Eq. (1)).
3: **end while**
4: **while** $K_1/2 \leq \tau < K_1$ **do**
5:     Running *Phase II* Iteration: Update rule Eq. (2).
6: **end while**

---

MTL per task sample size, respectively. $\kappa, \widetilde{\alpha}, \widetilde{\delta}, K_1, K_2, M$ represent the condition number $\kappa(\mathbf{\Sigma}^*)$, initial scaling, event probability, MTL iteration count, TL iteration count, TL sample size.

In our complexity analysis, following (Du et al., 2021; Tripuraneni et al., 2021; Niu et al., 2024), we treat $\kappa(\mathbf{\Sigma}^*)$ and $\alpha$ in Assumption 3.3 as constants in Corollaries 5.3 and 5.6, and focus exclusively on the dependence of the estimation error on $d, k, T, N, K_2$.

## 4. Algorithm

To recover the underlying representation $\mathbf{B}^*$ and task-specific matrix $\mathbf{W}^*$, we propose a TPGD algorithm (Algorithm 1). Its iteration can be divided into two phase.

*Phase I (Warm-Start):* In this phase, the algorithm first performs $K_1$ iterations of gradient descent on the function $\widehat{\mathcal{L}}(\mathbf{B}, \mathbf{W})$ to obtain initial estimates $\mathbf{B}^{(K_1/2)}$ and $\mathbf{W}^{(K_1/2)}$. Our analysis shows that this phase yields a coarse estimate that lies within a constant radius $C$ of the global optimum with high probability, serving as a qualified initialization for the second phase.

*Phase II (Refined Optimization with Correction):* In the second phase, we augment the update rules with correction terms

$$\mathcal{I}_{\mathbf{B}^{(\tau)}} := \mathbf{B}^{(\tau)}\left[\left(\mathbf{B}^{(\tau)}\right)^\top \mathbf{B}^{(\tau)} - \mathbf{W}^{(\tau)}\left(\mathbf{W}^{(\tau)}\right)^\top\right],$$

$$\mathcal{I}_{\mathbf{W}^{(\tau)}} := \left[\mathbf{W}^{(\tau)}\left(\mathbf{W}^{(\tau)}\right)^\top - \left(\mathbf{B}^{(\tau)}\right)^\top \mathbf{B}^{(\tau)}\right]\mathbf{W}^{(\tau)}.$$

which are computed from the first-phase outputs $(\mathbf{B}^{(K_1/2)}, \mathbf{W}^{(K_1/2)})$. This is equivalent to performing gradient descent on a regularized objective,

$$\widetilde{\mathcal{L}}(\mathbf{B}, \mathbf{W}) = \widehat{\mathcal{L}}(\mathbf{B}, \mathbf{W}) + \frac{1}{8}\left\|\mathbf{B}^\top \mathbf{B} - \mathbf{W}\mathbf{W}^\top\right\|_F^2. \quad (5)$$

The correction terms $\mathcal{I}_\mathbf{B}$ and $\mathcal{I}_\mathbf{W}$ are derived from the gradient of this regularization term. Their primary role is to mitigate the mutual coupling and interference between $\mathbf{B}$ and $\mathbf{W}$ during joint optimization——a challenge also prevalent in matrix sensing problems (Xiong et al., 2023). As detailed in Section 5, this modification leads to a faster convergence rate in optimization and, more importantly, yields a tighter generalization error bound.

## 5. Theoretical Analysis

In this section, we analyze the convergence of Algorithm 1 for the population loss of MTL (Section 5.1), and the generalization performance of the representation predictor learned by Algorithm 1 in the target task (Section 5.2).

### 5.1. Multi-Task Learning

Consider the singular value decomposition (SVD) of $\mathbf{B}^*\mathbf{W}^*$, given by

$$\mathbf{B}^*\mathbf{W}^* = \mathbf{U}^*\mathbf{\Sigma}^*\mathbf{V}^{*\top},$$

where $\mathbf{U}^* \in \mathbb{R}^{d\times d}$ and $\mathbf{V}^* \in \mathbb{R}^{T\times T}$ are orthogonal matrices. The following theorem provides the convergence guarantee for Algorithm 1 when learning the representation function.

**Theorem 5.1.** *Under Assumption 3.2, suppose $\delta \lesssim \min\{k^{-1/2}\kappa^{-7/2}, \kappa^{-6}\}$ and select the hyper-parameters as follows:*

$$\widetilde{\alpha} \lesssim \frac{\sigma_1(\mathbf{\Sigma}^*)}{k^5\left(\max\{d+T,k\}\right)^{2+C_1\kappa/2}} \cdot \left(\frac{\widetilde{\delta}}{\kappa^2}\right)^{C_1\kappa},$$

$$\eta_1 \lesssim \frac{1}{\kappa^5\sigma_1(\mathbf{\Sigma}^*)}, \quad K_1 \gtrsim \frac{1}{\eta_1\sigma_k(\mathbf{\Sigma}^*)}, \tag{6}$$

$$\eta_2 \lesssim \frac{1}{\sigma_1(\mathbf{\Sigma}^*)}, \quad N \gtrsim \frac{\sigma^2(d+T)k\kappa^4}{\sigma_k^2(\mathbf{\Sigma}^*)},$$

*where $\widetilde{\delta} \in (0,1)$ denotes the failure probability. Then the output $\widehat{\mathbf{B}} = \mathbf{B}^{(K_1)}, \widehat{\mathbf{W}} = \mathbf{W}^{(K_1)}$ from the last iteration of Algorithm 1 satisfies*

$$\left[\mathrm{dist}\left(\begin{pmatrix}\widehat{\mathbf{B}}\\\widehat{\mathbf{W}}^\top\end{pmatrix}, \begin{pmatrix}\mathbf{F}\\\mathbf{G}\end{pmatrix}\right)\right]^2$$

$$\lesssim \left(1 - \frac{\sigma_k(\mathbf{\Sigma}^*)}{4}\eta_2\right)^{K_1/2}\sigma_k(\mathbf{\Sigma}^*) + \sigma^2 \cdot \frac{\mathrm{tr}\,(\mathbf{\Sigma}^*)\,d}{\sigma_k^2\,(\mathbf{\Sigma}^*)\,N},$$

*with probability at least $1 - \widetilde{\delta} - C_2\exp(-C_3 k)$ where $C_2, C_3 > 0$ denote fixed numerical constants, and*

$$\mathbf{F} = \mathbf{U}^*(\cdot, [k])(\mathbf{\Sigma}^*_{[k]})^{1/2} \in \mathbb{R}^{d\times k},$$

$$\mathbf{G} = \mathbf{V}^*(\cdot, [k])(\mathbf{\Sigma}^*_{[k]})^{1/2} \in \mathbb{R}^{T\times k}.$$

*Remark* 5.2. One can notice that $\mathbf{B}^*$ and $\mathbf{F}$ are equivalent up to an invertible transformation $\mathbf{P} \in \mathbb{R}^{k\times k}$, i.e., $\mathbf{B}^* = \mathbf{FP}$. Consequently, treating $\kappa(\mathbf{\Sigma}^*)$ as a constant, Theorem 5.1 implies that $[\mathrm{dist}(\widehat{\mathbf{B}}, \mathbf{B}^*\mathbf{P}^{-1})]^2 \leq \widetilde{\mathcal{O}}(\frac{\sigma^2 dk}{\sigma_k(\mathbf{\Sigma}^*)N})$. For the detailed derivation, we refer the reader to Section A.1 in the appendix. It is worth noting that directly applying the analysis of Xiong et al. (2023) to traditional gradient descent for linear low-rank MTL suffers two major challenges. (1) High computational complexity: Existing techniques guarantee convergence only when the step size is smaller than $(\mathrm{poly}(d))^{-1}$, requiring more than $\widetilde{\mathcal{O}}(\mathrm{poly}(d))$ iterations. In contrast, Algorithm 1 allows a constant step size, achieving convergence within $\widetilde{\mathcal{O}}(1)$ iterations. (2) Overly stringent RIP condition: Existing techniques require the RIP constant $\delta \lesssim k^{-1}$ for global convergence, which means that a sample size of $N \gtrsim dk^2$ per task is necessary under the Gaussian setting. Our algorithm only requires $\delta \lesssim k^{-1/2}$ in its first phase to converge to a neighborhood of the ground-truth, and further relaxes this to $\delta \lesssim 1$ in the second phase. Thus, under the Gaussian setting, sample size $N \gtrsim dk$ assumed in Theorem 5.1 is sufficient to ensure $\delta \lesssim k^{-1/2}$.

The proof of Theorem 5.1 consists of two phases: *Phase I* and *Phase II*. The analysis for *Phase I* is inspired by the works of Soltanolkotabi et al. (2023) and Xiong et al. (2023). However, unlike their approaches, which focuses on approximating $\mathbf{B}^*\mathbf{W}^*$ by $\mathbf{BW}$, we prove that $(\mathbf{B}, \mathbf{W}^\top)$ converges to a neighborhood of $(\mathbf{F}, \mathbf{G})$. This demonstrates that, starting from random initialization, Algorithm 1 learns the feature spaces of $\mathbf{F}$ and $\mathbf{G}$ already within *Phase I*, thereby establishing the feature-learning capability of this phase.

In *Phase II*, we prove the global convergence of the algorithm. Instead of adopting conventional convergence analyses from optimization theory—which typically construct a suitable convex (or strongly convex) energy function and relate the algorithm's iterative trajectory to the convexity of this function—we decompose the distance error considered in Theorem 5.1 into two components: Part I, which is driven by the gradient of $\widetilde{\mathcal{L}}$ in Eq. (5) without additive noise, and Part II, which is generated by the additive noise. Part I is shown to exhibit linear convergence. Specifically, we show that the function $\widetilde{\mathcal{L}}$ without additive noise satisfies a *Regularity Condition* (Candes et al., 2015; Tu et al., 2016), which can be viewed as a form of strong convexity applicable to functions possessing rotational degrees of freedom at optimal points. Part II contributes an error term through cumulative summation, given by $\frac{\sigma^2\,\mathrm{tr}(\mathbf{\Sigma}^*)d}{\sigma_k^2(\mathbf{\Sigma}^*)N}$. This extension is necessary because, in the problem considered, rotational degrees of freedom prevent the construction of an energy function that is convex (or strongly convex) in any neighborhood of $(\mathbf{B}^*, \mathbf{W}^*)$, except in the special case $k = 1$.

The following corollary implies that under the conditions

$T > k$ and $N$ samples per task, the estimation error of MTL is strictly lower than that of learning $T$ task parameters independently ($\widetilde{\mathcal{O}}(\frac{dk}{NT})$ v.s. $\widetilde{\mathcal{O}}(\frac{d}{N})$).

**Corollary 5.3.** *Suppose that the assumptions and hyper-parameter settings of Theorem 5.1 hold. Then, the MTL population loss achieved by Algorithm 1 satisfies*

$$\frac{1}{T} \sum_{t=1}^{T} \|\widehat{\mathbf{v}}_t - \mathbf{v}_t^*\|^2 \lesssim \sigma^2 \cdot \frac{dk}{NT},$$

*with probability at least $1 - \widetilde{\delta} - C_2 \exp(-C_3 k)$, where $\mathbf{v}_t^* := \mathbf{B}^* \mathbf{w}_t^*$ and $\widehat{\mathbf{v}}_t := \widehat{\mathbf{B}} \widehat{\mathbf{w}}_t$ for any $t \in [T]$.*

The best known likelihood-based method (Tripuraneni et al., 2021; Du et al., 2021; Thekumparampil et al., 2021; Duan & Wang, 2023; Tian et al., 2025) achieve a convergence rate of $\widetilde{\mathcal{O}}(\frac{dk^2}{NT})$ for the population loss $\frac{1}{T} \sum_{t=1}^{T} \|\widehat{\mathbf{v}}_t - \mathbf{v}_t^*\|^2$. According to Corollary 5.3, our algorithm attains a faster rate of $\widetilde{\mathcal{O}}(\frac{dk}{NT})$. This rate matches the known lower bound (Tripuraneni et al., 2021; Tian et al., 2025) up to logarithmic factors. While Corollary 5.3 is primarily established under the high-dimensional regime where $d > T$, it provides a general population loss bound of $\widetilde{\mathcal{O}}(\frac{\max\{d,T\}k}{NT})$. Therefore, in the regime where $T \geq d$, Algorithm 1 also achieves a superior loss bound of $\widetilde{\mathcal{O}}(\frac{k}{N})$, outperforming the $\widetilde{\mathcal{O}}(\frac{d}{N})$ bound associated with learning $T$ task parameters independently.

### 5.2. Transfer Learning

We consider the $T + 1$ task is trained by SGD with decay step size, which is shown in Algorithm 2 in Appendix A.2.

**Theorem 5.4.** *Under Assumption 3.2 and Assumption 3.3, suppose $\delta \lesssim \min\{k^{-1/2}\kappa^{-7/2}, \kappa^{-6}\}$ and select the hyper-parameters Eq. (6) and*

$$\eta_0 \lesssim \frac{1}{\alpha \operatorname{tr}\left(\widehat{\mathbf{B}}^\top \mathbf{H} \widehat{\mathbf{B}}\right)}, K_2 \gtrsim \max\left\{\frac{\alpha}{\sigma^2 \eta_0}, \frac{1}{2\eta_0 \sigma_k\left(\widehat{\mathbf{B}}^\top \mathbf{H} \widehat{\mathbf{B}}\right)}\right\}.$$

*Then the output $\mathbf{w}_{T+1}^{(K_2)}$ from the last iteration of stochastic gradient descent (Algorithm 2 in Appendix) satisfies*

$$L_\rho(\widehat{\mathbf{B}}, \mathbf{w}_{T+1}^{(K_2)}) - L_\rho(\mathbf{B}^*, \mathbf{w}_{T+1}^*)$$
$$\lesssim \sigma^2 \|\mathbf{w}_{T+1}^*\|^2 \left[\left(1 + \beta_1^2\right) \|\mathbf{H}\|_{\mathrm{op}} + 1 + \beta_2^2\right] \cdot \frac{\operatorname{tr}(\mathbf{\Sigma}^*) d}{\sigma_k^2(\mathbf{\Sigma}^*) N}$$
$$+ \frac{\sigma^2 k}{K_2}, \tag{7}$$

*where $\beta_1 := \|\left(\mathbf{H}^{1/2} \mathbf{B}^*\right)^\dagger\|_{\mathrm{op}}$ and $\beta_2 := \|\mathbf{B}^*\|_{\mathrm{op}} \|\mathbf{H}\|_{\mathrm{op}}$.*

*Remark 5.5.* Under the constraint of a maximum sample size $M$ in TL phase, the online nature of Algorithm 2 implies that $M = K_2$. Given orthogonal matrix $\widetilde{\mathbf{U}} \in \mathbb{R}^{d \times k}$, suppose $\widetilde{\mathbf{U}}^\top \mathbf{x}_{T+1}$ satisfies the RIP with $\delta \lesssim 1$. Then applying

the ERM methods of Du et al. (2021); Tripuraneni et al. (2021) for linear regression on the target task $T + 1$ yields a convergence rate analogous to that in Theorem 5.4, under the condition that the transfer sample size $M \gtrsim k$. In contrast, Theorem 5.4 is established under a much weaker assumption, requiring only that covariate $\mathbf{x}_{T+1}$ satisfies mild moment conditions.

In this section, we optimize $\mathbf{w}_{T+1}$ while fixing the representation $\widehat{\mathbf{B}}$ fixed. Compared with the typical dynamics of SGD with decaying step size in linear models (Wu et al., 2022), the variance component of SGD in this TL setting contains an additional term arising from both additive noise and the irreducible approximation error $\operatorname{dist}(\widehat{\mathbf{B}}, \mathbf{B}^*)$. This error further introduces stochastic noise that is correlated with the covariate $\mathbf{x}_{T+1}$. To address the statistical dependence caused by this correlation, we employ a block-wise variance decomposition technique, which provides a fine-grained analysis of the variance error, along with a recursive analysis that yields an iteration-wise estimate of the variance.

**Corollary 5.6.** *Suppose that the assumptions and hyper-parameter settings of Theorem 5.4 hold, and assume $\|\mathbf{v}_t^*\| = \mathcal{O}(1)$. Then the output $\mathbf{w}_{T+1}^{(K_2)}$ from the last iteration of stochastic gradient descent (Algorithm 2 in Appendix) satisfies*

$$L_\rho(\widehat{\mathbf{B}}, \mathbf{w}_{T+1}^{(K_2)}) - L_\rho(\mathbf{B}^*, \mathbf{w}_{T+1}^*)$$
$$\lesssim \sigma^2 \left[\left(1 + \beta_1^2\right) \|\mathbf{H}\|_{\mathrm{op}} + 1 + \beta_2^2\right] \cdot \frac{dk^2}{NT} + \frac{\sigma^2 k}{K_2}, \tag{8}$$

*where $\beta_1 := \|\left(\mathbf{H}^{1/2} \mathbf{B}^*\right)^\dagger\|_{\mathrm{op}}$ and $\beta_2 := \|\mathbf{B}^*\|_{\mathrm{op}} \|\mathbf{H}\|_{\mathrm{op}}$.*

The convergence rate of our algorithm in Eq. (8) matches the best known rate $\widetilde{\mathcal{O}}(\frac{dk^2}{NT} + \frac{k}{M})$ up to logarithmic factors for likelihood-based low-dimensional linear representation TL methods. Under the additional assumption $\mathbb{E}[\|\mathbf{w}_{T+1}\|^2] \lesssim 1/k$ introduced by Du et al. (2021) (Theorem 4.1), the convergence rate of our algorithm achieves $\widetilde{\mathcal{O}}(\frac{dk}{NT} + \frac{k}{M})$.

## 6. Numerical Simulations

This section provides numerical simulations of TPGD, focusing on its convergence rate advantage with respect to both the total number of iterations and the sample size.

We compare the estimation error curves of Algorithm 1 against three comparative LBM algorithms for MTL, which have tractable update rule and require no per-iteration matrix decomposition (e.g., SVD or QR). The comparison also includes the theoretical estimation error curve derived in Section 5. All algorithms are applied to the same MTL problem defined in Section 3. The experiments consist of two groups, presented from top to bottom, under fixed input

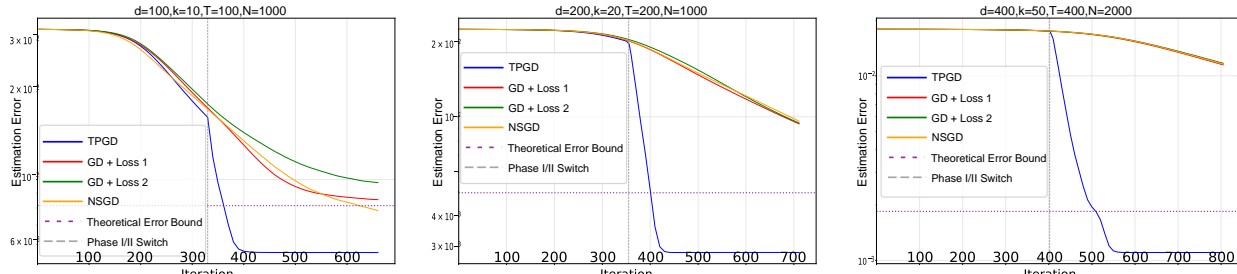

*Figure 1.* The convergence rates between TPGD and the comparative algorithms with fixed per-task sample size $N$. The vertical axis represents the last-iterate estimation error, while the horizontal axis denotes the number of training iterations.

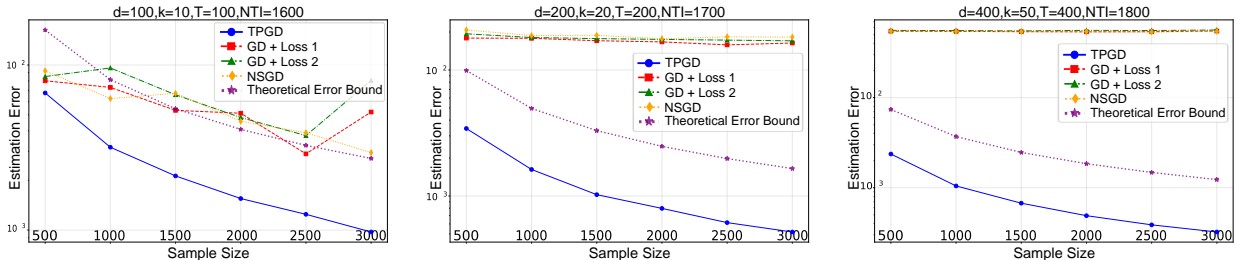

*Figure 2.* The convergence rates between TPGD and the comparative algorithms with fixed total number of iterations $NTI$. The vertical axis represents the last-iterate estimation error, while the horizontal axis denotes the sample size.

dimension $d$, representation dimension $k$, and task count $T$. The first group (Figure 1) examines the convergence rate with respect to the total number of iterations for fixed per-task sample sizes. The second group (Figure 2) evaluates the convergence rate with respect to the per-task sample size for fixed total numbers of iterations. The theoretical curve is given by $\widetilde{\mathcal{O}}\left(\frac{dk}{NT}\right)$. Hence, when per-task sample size $N$ is fixed, this curve appears as a horizontal line.

In the experiments, the curve labeled "TPGD" represents the last-iterate estimation error of Algorithm 1. The "GD + Loss 1" curve corresponds to the tail-averaging estimation error of constant-step-size gradient descent (GD) applied to the *Phase I* loss (Eq. (3)). The "GD + Loss 2" curve denotes the tail-averaging estimation error of a constant-step-size GD method applied to the loss function from Tripuraneni et al. (2021), which is defined as

$$\frac{1}{2N}\sum_{t=1}^{T}\left\|\mathbf{y}_t - \mathbf{X}_t^{\top}\mathbf{B}\mathbf{w}_t\right\|^2 + \frac{1}{2}\left\|\mathbf{B}^{\top}\mathbf{B} - \mathbf{W}\mathbf{W}^{\top}\right\|_{\mathrm{F}}^2. \tag{9}$$

The "NSGD" curve represents the tail-averaging estimation error of the Noise-Scheduled SGD method (Fang et al., 2019) on the loss given in Eq. (9).

All comparative algorithms are initialized in the same manner as Algorithm 1, employing the recommended or default hyper-parameters from their original publications. The hyper-parameters for Algorithm 1 are set following the guidelines of Theorem 5.1, with the theoretical values scaled

by a constant factor. Specifically, the *Phase I* step size $\eta_1$ is scaled by a constant factor 1.1, while the *Phase II* step size $\eta_2$ is taken as 0.1 times the theoretical value. For "GD + Loss 1", which lacks specific recommendations, we set its step size to match the *Phase I* step size of TPGD.

As shown in Figure 1, Algorithm 1 exhibits the fastest convergence in reducing the estimation error when the per-task sample size $N$ is fixed. Figure 2 further demonstrates that Algorithm 1 also achieves the lowest estimation error among all algorithms compared. In summary, these results indicate that our algorithm holds a notable efficiency advantage over existing tractable LBM algorithms for MTL, which have no per-iteration steps (e.g., SVD or QR).

In Section C, we provide additional experiments to validate the efficiency of TPGD. Specifically, the ablation study (Figure 3) confirms that both phases are indispensable, while the iteration comparison (Table 2) shows that TPGD achieves faster stabilization and a lower final loss than SVD/QR-based methods.

## 7. Conclusion

This work provides an efficiently likelihood-based first-order algorithm for MTL with a shared linear representation. Our results justify that principled likelihood optimization, despite its inherent non-convexity, can be achieved with both computational efficiency and statistical near-optimality in the MTL setting.

In Appendix A.3, we extend the analysis to the regime where task noise variances are heterogeneous. There, we study a curriculum learning strategy wherein tasks are first grouped and then learned sequentially in order of increasing noise variance. We provide a theoretical justification for the advantage of this incremental scheme over training all tasks at the same time.

## Acknowledgements

C. Fang was supported by the NSF China (No.s 62376008) and the NSF China (No.s 92470117). This work was also supported in part by the Beijing Major Science and Technology Project under Contract no. Z251100008125007.

## Impact Statement

This paper presents work whose goal is to advance the field of Machine Learning. There are many potential societal consequences of our work, none which we feel must be specifically highlighted here.

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

# A. Appendix

Recall the SVD of $\mathbf{B}^*\mathbf{W}^*$ is:

$$\mathbf{B}^*\mathbf{W}^* = \mathbf{U}^*\mathbf{\Sigma}^*\left(\mathbf{V}^*\right)^\top,$$

where $\mathbf{U}^* \in \mathbb{R}^{d\times d}$ and $\mathbf{V}^* \in \mathbb{R}^{T\times T}$. For simplicity, we denote the $t$-th row of $\mathbf{V}^*$ by $\mathbf{v}_t^* = (\mathbf{V}^*(t,\cdot))^\top$ for any $t \in [T]$. Define the transformed matrices $\widetilde{\mathbf{B}} = (\mathbf{U}^*)^\top\mathbf{B}$ and $\widetilde{\mathbf{W}} = \mathbf{W}\mathbf{V}^*$. Then, according to the update rule in *Phase I* of Algorithm 1, the iterations from $\widetilde{\mathbf{B}}^{(\tau)}$ to $\widetilde{\mathbf{B}}^{(\tau+1)}$ and from $\widetilde{\mathbf{W}}^{(\tau)}$ to $\widetilde{\mathbf{W}}^{(\tau+1)}$ can be written as follows:

$$
\begin{aligned}
\widetilde{\mathbf{B}}^{(\tau+1)} =& \widetilde{\mathbf{B}}^{(\tau)} - \frac{\eta_1}{N}(\mathbf{U}^*)^\top\left(\mathbf{X}_1\left(\mathbf{X}_1^\top\mathbf{B}^{(\tau)}\mathbf{w}_1^{(\tau)} - \mathbf{y}_1\right), \cdots, \mathbf{X}_T\left(\mathbf{X}_T^\top\mathbf{B}^{(\tau)}\mathbf{w}_T^{(\tau)} - \mathbf{y}_T\right)\right)\left(\mathbf{W}^{(\tau)}\right)^\top \\
=& \widetilde{\mathbf{B}}^{(\tau)} - \eta_1\sum_{t=1}^T\left[\frac{1}{N}\sum_{i=1}^N\left(\left\langle\mathbf{x}_t^{(i)}, \mathbf{B}^{(\tau)}\mathbf{w}_t^{(\tau)}\right\rangle - y_t^{(i)}\right)(\mathbf{U}^*)^\top\mathbf{x}_t^{(i)}(\mathbf{v}_t^*)^\top\right]\left(\widetilde{\mathbf{W}}^{(\tau)}\right)^\top \\
\overset{(a_1)}{=}& \widetilde{\mathbf{B}}^{(\tau)} - \eta_1\sum_{t=1}^T\left[\frac{1}{N}\sum_{i=1}^N\left(\left\langle\mathbf{M}_t^{(i)}(\mathbf{x}), \widetilde{\mathbf{B}}^{(\tau)}\widetilde{\mathbf{W}}^{(\tau)} - \mathbf{\Sigma}^*\right\rangle - z_t^{(i)}\right)\mathbf{M}_t^{(i)}(\mathbf{x})\right]\left(\widetilde{\mathbf{W}}^{(\tau)}\right)^\top, \quad (10)
\end{aligned}
$$

$$
\begin{aligned}
\widetilde{\mathbf{W}}^{(\tau+1)} =& \widetilde{\mathbf{W}}^\tau - \frac{\eta_1}{N}\left(\mathbf{B}^{(\tau)}\right)^\top\left(\mathbf{X}_1\left(\mathbf{X}_1^\top\mathbf{B}^{(\tau)}\mathbf{w}_1^{(\tau)} - \mathbf{y}_1\right), \cdots, \mathbf{X}_T\left(\mathbf{X}_T^\top\mathbf{B}^{(\tau)}\mathbf{w}_T^{(\tau)} - \mathbf{y}_T\right)\right)\mathbf{V}^* \\
\overset{(a_2)}{=}& \widetilde{\mathbf{W}}^\tau - \eta_1\left(\widetilde{\mathbf{B}}^{(\tau)}\right)^\top\sum_{t=1}^T\left[\frac{1}{N}\sum_{i=1}^N\left(\left\langle\mathbf{M}_t^{(i)}(\mathbf{x}), \widetilde{\mathbf{B}}^{(\tau)}\widetilde{\mathbf{W}}^{(\tau)} - \mathbf{\Sigma}^*\right\rangle - z_t^{(i)}\right)\mathbf{M}_t^{(i)}(\mathbf{x})\right], \quad (11)
\end{aligned}
$$

where matrix $\mathbf{M}_t^{(i)}(\mathbf{x}) \in \mathbb{R}^{d\times T}$ denotes $(\mathbf{U}^*)^\top\mathbf{x}_t^{(i)}(\mathbf{v}_t^*)^\top$, and $(a_1)$ and $(a_2)$ are derived from the fact that

$$
\begin{aligned}
\left\langle\mathbf{x}_t^{(i)}, \mathbf{B}^{(\tau)}\mathbf{w}_t^{(\tau)}\right\rangle - y_t^{(i)} &= \left\langle\mathbf{x}_t^{(i)}, \mathbf{B}^{(\tau)}\mathbf{w}_t^{(\tau)} - \mathbf{B}^*\mathbf{w}_t^*\right\rangle - z_t^{(i)} \\
&= \left\langle\mathbf{x}_t^{(i)}, \mathbf{U}^*\widetilde{\mathbf{B}}^{(\tau)}\widetilde{\mathbf{W}}^{(\tau)}\mathbf{v}_t^* - \mathbf{U}^*\mathbf{\Sigma}^*\mathbf{v}_t^*\right\rangle - z_t^{(i)} \\
&= \left\langle(\mathbf{U}^*)^\top\mathbf{x}_t^{(i)}(\mathbf{v}_t^*)^\top, \widetilde{\mathbf{B}}^{(\tau)}\widetilde{\mathbf{W}}^{(\tau)} - \mathbf{\Sigma}^*\right\rangle - z_t^{(i)},
\end{aligned}
$$

for any $t \in [T]$ and $i \in [N]$. Since $[\mathbf{A_1}]$ of Assumption 3.2 holds, we have

$$(1-\delta)\|\mathbf{M}\mathbf{v}_t^*\|^2 \leq \frac{1}{N}\sum_{i=1}^N\left\langle\mathbf{M}_t^{(i)}(\mathbf{x}), \mathbf{M}\right\rangle^2 = \frac{1}{N}\sum_{i=1}^N\left\langle\mathbf{x}_t^{(i)}, \mathbf{U}^*\mathbf{M}\mathbf{v}_t^*\right\rangle^2 \leq (1+\delta)\|\mathbf{M}\mathbf{v}_t^*\|^2,$$

for any $t \in [T]$ and $\mathbf{M} \in \mathbb{R}^{d\times T}$. Then, utilizing Lemma D.1, we obtain that

$$\left|\frac{1}{N}\sum_{i=1}^N\left\langle\mathbf{M}_t^{(i)}(\mathbf{x}), \mathbf{M}_1\right\rangle\left\langle\mathbf{M}_t^{(i)}(\mathbf{x}), \mathbf{M}_2\right\rangle - \left\langle\mathbf{M}_1\mathbf{v}_t^*, \mathbf{M}_2\mathbf{v}_t^*\right\rangle\right| \leq \delta\|\mathbf{M}_1\mathbf{v}_t^*\|\|\mathbf{M}_2\mathbf{v}_t^*\|$$

for any $t \in [T]$ and $\mathbf{M}_1, \mathbf{M}_2 \in \mathbb{R}^{d\times T}$.

## A.1. Proof of Theorem 5.1

The analysis in **Phase I** is inspired by the works of (Soltanolkotabi et al., 2023; Xiong et al., 2023). However, while their focus lies on approximating $\mathbf{B}^*\mathbf{W}^*$ by $\mathbf{B}\mathbf{W}$, we instead prove that $(\mathbf{B}, \mathbf{W}^\top)$ converges to a neighborhood of $(\mathbf{F}, \mathbf{G})$, where $\mathbf{F} = \mathbf{U}^*(\cdot, [k])(\mathbf{\Sigma}_{[k]}^*)^{1/2} \in \mathbb{R}^{d\times k}$ and $\mathbf{G} = \mathbf{V}^*(\cdot, [k])(\mathbf{\Sigma}_{[k]}^*)^{1/2} \in \mathbb{R}^{T\times k}$. Specifically, we establish

$$\text{dist}\left(\begin{pmatrix}\mathbf{B}\\\mathbf{W}^\top\end{pmatrix}, \begin{pmatrix}\mathbf{F}\\\mathbf{G}\end{pmatrix}\right) \leq \frac{\sigma_k^{1/2}(\mathbf{\Sigma}^*)}{8}.$$

Our proof primarily analyzes the dynamics of the projected parameter matrices $\widetilde{\mathbf{B}}$ and $\widetilde{\mathbf{W}}^\top$ (defined in Eq. (10) and Eq. (11)), which lie in the subspace spanned by $\mathbf{U}^*$ and $\mathbf{V}^*$. The stochastic error induced by label noise $z$ is bounded via Lemma B.1. A further block-wise analysis of the iterative updates for $\widetilde{\mathbf{B}}$ and $\widetilde{\mathbf{W}}$ reveals that the following quantities remain consistently small throughout this phase:

- $\left\|\widetilde{\mathbf{B}}_{[k],[k]} - \widetilde{\mathbf{W}}^\top_{[k],[k]}\right\|$, the operator norm of the difference between the top $k \times k$ submatrices of $\widetilde{\mathbf{B}}$ and $\widetilde{\mathbf{W}}^\top$.

- $\left\|\widetilde{\mathbf{B}}_{[k+1:d],[k]}\right\|$ and $\left\|\widetilde{\mathbf{W}}^\top_{[k],[k+1:T]}\right\|$, the operator norms of the corresponding trailing submatrices of $\widetilde{\mathbf{B}}$ and $\widetilde{\mathbf{W}}^\top$.

Combining these error estimates with Lemma B.2 completes the proof for **Phase I**.

**Phase II** of Algorithm 2 employs gradient descent with a regularization term. Here, we aim to prove that the distance between $(\mathbf{B}, \mathbf{W}^\top)$ and $(\mathbf{F}, \mathbf{G})$ converges throughout the algorithm's updates. In conventional convergence analyses of optimization algorithms, one often constructs a suitable convex (or strongly convex) energy function and relates the iterative trajectory of the algorithm to the convexity of this function to establish convergence. However, in the problem we consider, any optimal point possesses rotational degrees of freedom, which prevents the construction of such a convex (or strongly convex) energy function. Therefore, we adopt an approach based on designing an appropriate loss function and verifying that it satisfies the *Regularity Condition* which is introduced in prior works (Tu et al., 2016; Candes et al., 2015). This condition provides a sufficient criterion for convergence and relies only on first-order information along specific trajectories.

We have shown that the output $\left(\mathbf{B}^{(K_1/2)}, (\mathbf{W}^{K_1/2})^\top\right)$ from **Phase I** lies within a neighborhood of $(\mathbf{F}, \mathbf{G})$. In **Phase II**, we prove that the noise error arising from the covariate vectors $\{\mathbf{x}_t^{(i)}\}_{i=1}^N$ in each task $t$ can be bounded to an appropriate magnitude. This enables us to further establish that the component of the loss gradient formed by these covariate vectors satisfies a certain *Regularity Condition*. Under this condition, the distance between $\left(\mathbf{B}^{(\tau)}, (\mathbf{W}^{(\tau)})^\top\right)$ and $(\mathbf{F}, \mathbf{G})$ decays linearly for $\tau \in [K_1/2 : K_1]$ when label noise is disregarded. Consequently, we demonstrate that this distance comprises two distinct components: a deterministic error term that decreases linearly, and a stochastic error term induced by label noise, the latter of which can be controlled by the sample size $N$.

### A.1.1. PROOF OF PHASE I

For simplicity, define the operator $\mathcal{A} : \mathbb{R}^{d \times T} \to \mathbb{R}^{N \times T}$ and its adjoint $\mathcal{A}^* : \mathbb{R}^{N \times T} \to \mathbb{R}^{d \times T}$ by

$$\mathcal{A}(\mathbf{M}) = \left(\mathcal{A}(\mathbf{M})_{it}\right)_{N \times T} = \frac{1}{\sqrt{N}}\left(\langle \mathbf{M}_t^{(i)}(\mathbf{x}), \mathbf{M}\rangle\right)_{N \times T}, \quad \forall \mathbf{M} \in \mathbb{R}^{d \times T},$$

and

$$\mathcal{A}^*(\mathbf{B}) = \frac{1}{\sqrt{N}}\sum_{t=1}^T \sum_{i=1}^N \mathbf{B}_{it}\mathbf{M}_t^{(i)}(\mathbf{x}), \quad \forall \mathbf{B} \in \mathbb{R}^{N \times T}.$$

Furthermore, we denote the noise term as

$$\mathbf{E} = \frac{\eta_1}{N}\sum_{i=1}^N \sum_{t=1}^T z_t^{(i)} \mathbf{M}_t^{(i)}(\mathbf{x}).$$

Then we rewrite Eqs. (10) and (11) as

$$\begin{cases} \widetilde{\mathbf{B}}^{(\tau+1)} = \widetilde{\mathbf{B}}^{(\tau)} - \eta_1 \mathcal{A}^* \mathcal{A}\left(\widetilde{\mathbf{B}}^{(\tau)}\widetilde{\mathbf{W}}^{(\tau)} - \mathbf{\Sigma}^*\right)\left(\widetilde{\mathbf{W}}^{(\tau)}\right)^\top - \eta_1 \cdot \mathbf{E}\left(\widetilde{\mathbf{W}}^{(\tau)}\right)^\top, \\[2mm] \widetilde{\mathbf{W}}^{(\tau+1)} = \widetilde{\mathbf{W}}^{(\tau)} - \eta_1 \left(\widetilde{\mathbf{B}}^{(\tau)}\right)^\top \mathcal{A}^* \mathcal{A}\left(\widetilde{\mathbf{B}}^{(\tau)}\widetilde{\mathbf{W}}^{(\tau)} - \mathbf{\Sigma}^*\right) - \eta_1 \cdot \left(\widetilde{\mathbf{B}}^{(\tau)}\right)^\top \mathbf{E}. \end{cases} \tag{12}$$

**Theorem A.1.** *Let the initial scale $\alpha$, step-size $\eta_1$ and sample size $N$ satisfy*

$$\alpha \lesssim \min\left\{ \frac{1}{\kappa^2(\mathbf{\Sigma}^*)\sigma_k^{1/2}(\mathbf{\Sigma}^*)}, \frac{\sigma_k^{5/3}(\mathbf{\Sigma}^*)}{k\sigma_1^{3/2}(\mathbf{\Sigma}^*)\kappa^2(\mathbf{\Sigma}^*)}, \frac{\sigma_1^{1/2}(\mathbf{\Sigma}^*)}{k^5 \max\{d,T\}^2} \cdot \left(\frac{\epsilon\left(\sqrt{k} - \sqrt{k-1}\right)}{\kappa^2(\mathbf{\Sigma}^*)\sqrt{\max\{d,T\}}}\right)^{C_1\kappa(\mathbf{\Sigma}^*)} \right\},$$

$$\eta_1 \lesssim \frac{1}{\kappa^5(\mathbf{\Sigma}^*)\sigma_1(\mathbf{\Sigma}^*)}, \qquad N \gtrsim \frac{\sigma^2\left(\max\{d,T\} + d\right)k\kappa^4(\mathbf{\Sigma}^*)}{\sigma_k^2(\mathbf{\Sigma}^*)},$$

*and suppose Assumption 3.2 holds with $\delta$ such that*

$$\delta \lesssim \min\left\{\frac{1}{\kappa^6(\mathbf{\Sigma}^*)}, \frac{1}{\kappa^3(\mathbf{\Sigma}^*)\sqrt{k}}\right\}. \tag{13}$$

*Then the last-iterate output $\left(\mathbf{B}^{(K_1/2)}, \mathbf{W}^{(K_1/2)}\right)$ of the first phase of Algorithm 2 satisfies*

$$\text{dist}\left(\mathbf{Z}^{(K_1/2)}, \mathbf{J}\right) \leq \frac{\sigma_k(\mathbf{J})}{8}, \tag{14}$$

*where*

$$\mathbf{Z}^{(K_1/2)} = \begin{pmatrix} \mathbf{B}^{(K_1/2)} \\ (\mathbf{W}^{(K_1/2)})^\top \end{pmatrix}, \qquad \mathbf{J} = \begin{pmatrix} \mathbf{U}^*(\cdot, [k]) \left(\mathbf{\Sigma}^*_{[k]}\right)^{1/2} \\ \mathbf{V}^*(\cdot, [k]) \left(\mathbf{\Sigma}^*_{[k]}\right)^{1/2} \end{pmatrix},$$

*with probability at least $1 - \widetilde{\delta} - C_2 \exp(-C_3 k) - C_4 \epsilon$ for failure probability $\widetilde{\delta} \in (0, 1)$ and fixed numerical constants $C_2, C_3, C_4 > 0$ when $K_1 \gtrsim \frac{\kappa(\mathbf{\Sigma}^*)}{\eta_1 \sigma_k(\mathbf{\Sigma}^*)}$.*

*Proof.* Utilizing Lemma B.1, and Lemma B.3, and Lemma B.5 for the first phase of Algorithm 1 yields

$$\left\| \widehat{\mathbf{B}}^{(K_1/2)} \widehat{\mathbf{W}}^{(K_1/2)} - \mathbf{\Sigma}^* \right\| \lesssim \frac{\sigma_k(\mathbf{\Sigma}^*)}{\sqrt{k}\kappa(\mathbf{\Sigma}^*)}$$

$$\left\| \widetilde{\mathbf{B}}^{(K_1/2)}_{[k],[k]} - \widetilde{\mathbf{W}}^{(K_1/2)}_{[k],[k]} \right\| \lesssim \alpha + \frac{\delta \sigma_1^{3/2}(\mathbf{\Sigma}^*)}{\sigma_k(\mathbf{\Sigma}^*)} + \frac{\sigma \left(\sigma_1(\mathbf{\Sigma}^*)(\max\{d, T\} + d)\right)^{\frac{1}{2}}}{\sigma_k(\mathbf{\Sigma}^*)\sqrt{N}}, \tag{15}$$

$$\max\left\{ \left\| \widetilde{\mathbf{B}}^{(K_1/2)}_{[k+1:d],[k]} \right\|, \left\| \widetilde{\mathbf{W}}^{(K_1/2)}_{[k],[k+1:T]} \right\| \right\} \lesssim \alpha + \frac{\delta \sigma_1^{3/2}(\mathbf{\Sigma}^*)}{\sigma_k(\mathbf{\Sigma}^*)} + \frac{\sigma \left(\sigma_1(\mathbf{\Sigma}^*)(\max\{d, T\} + d)\right)^{\frac{1}{2}}}{\sigma_k(\mathbf{\Sigma}^*)\sqrt{N}},$$

with probability at least $1 - \widetilde{\delta} - C_2 \exp(-C_3 k) - C_4 \epsilon$. Combining Lemma B.2 with Eq. (15) and the hyper-parameters setting of Theorem 5.1, we have

$$\text{dist}\left( \begin{pmatrix} \widetilde{\mathbf{B}}^{(K_1/2)} \\ \left(\widetilde{\mathbf{W}}^{(K_1/2)}\right)^\top \end{pmatrix}, \begin{pmatrix} \begin{pmatrix} \left(\mathbf{\Sigma}^*_{[k]}\right)^{1/2} \\ \mathbf{0}_{(d-k)\times k} \end{pmatrix} \\ \begin{pmatrix} \left(\mathbf{\Sigma}^*_{[k]}\right)^{1/2} \\ \mathbf{0}_{(T-k)\times k} \end{pmatrix} \end{pmatrix} \right) \leq \frac{\sigma_k^{1/2}(\mathbf{\Sigma}^*)}{8},$$

which implies that

$$\text{dist}\left(\mathbf{Z}^{(K_1/2)}, \mathbf{J}\right) \leq \frac{\sigma_k(\mathbf{J})}{8}.$$

$\square$

### A.1.2. PRELIMINARY

Before proceeding to the proof of **Phase II**, we first introduce some preliminary knowledge. For a general function $f : \mathbb{R}^d \to \mathbb{R}_+$, ensuring that a numerical method can find its global minimum typically requires $f$ to be convex or strongly convex. For a loss function on matrix $F : \mathbb{R}^{d \times k} \to \mathbb{R}_+$, there often exists rotational invariance around its global optima. Consequently, unless in the special case $k = 1$, $F$ is generally neither convex nor strongly convex in any neighborhood of a global optimum. To handle loss functions with such rotational freedom, Candes et al. (2015) and Tu et al. (2016) introduce a regularity condition (RC)—a property analogous to strong convexity—and showed that it is satisfied for problems such as quadratic equations solving, matrix factorization, and offline matrix sensing. Next, we formally define the RC as follows:

**Definition A.2** (Regularity Condition). Let $\mathbf{J} \in \mathbb{R}^{d \times k}$ be a global optimum of $F(\cdot)$. Define the neighborhood $B(\hat{\delta})$ as

$$B(\hat{\delta}) := \left\{ \mathbf{Z} \in \mathbb{R}^{d \times k} : \text{dist}(\mathbf{Z}, \mathbf{J}) \leq \hat{\delta} \right\}.$$

We say $F(\cdot)$ satisfies the regularity condition $\text{RC}(\alpha, \beta, \hat{\delta})$ if for all matrices $\mathbf{Z} \in B(\hat{\delta})$ the following inequality holds

$$\langle \nabla F(\mathbf{Z}), \mathbf{Z} - \mathbf{J}\mathbf{R} \rangle \geq \frac{1}{\alpha} \|\mathbf{Z} - \mathbf{J}\mathbf{R}\|_{\text{F}}^2 + \frac{1}{\beta} \|\nabla F(\mathbf{Z})\|_{\text{F}}^2,$$

for some orthogonal matrix $\mathbf{R} \in \mathbb{R}^{k \times k}$.

Specifically, Tu et al. (2016) define a reference function $L(\mathbf{Z}) := \frac{1}{4} \left\| \mathbf{Z}\mathbf{Z}^\top - \mathbf{J}\mathbf{J}^\top \right\|_F^2$ with gradient $\nabla L(\mathbf{Z}) = \left( \mathbf{Z}\mathbf{Z}^\top - \mathbf{J}\mathbf{J}^\top \right) \mathbf{Z}$, and show that $L(\cdot)$ satisfies $RC(4/\sigma_k^2(\mathbf{J}), 9\sigma_1^2(\mathbf{J}), \sigma_k(\mathbf{J})/4)$. Moreover, they prove that there exists an orthogonal matrix $\mathbf{R}$ the gradient $\nabla L(\cdot)$ satisfies the following inequality:

$$\langle \nabla L(\mathbf{Z}), \mathbf{Z} - \mathbf{J}\mathbf{R} \rangle \geq \frac{\sigma_k^2(\mathbf{J})}{4} \left\| \mathbf{Z} - \mathbf{J}\mathbf{R} \right\|_F^2 + \frac{1}{4} \left\| \mathbf{Z}\mathbf{Z}^\top - \mathbf{J}\mathbf{J}^\top \right\|_F^2 + \frac{1}{20} \left\| (\mathbf{Z} - \mathbf{J}\mathbf{R}) \mathbf{Z}^\top \right\|_F^2, \tag{16}$$

for any matrix $\mathbf{Z}$ obeying $\|\mathbf{Z} - \mathbf{J}\mathbf{R}\| \leq \sigma_k(\mathbf{J})/4$.

### A.1.3. PROOF OF PHASE II

We define the affine maps $\mathcal{B} : \mathbb{R}^{d \times T} \to \mathbb{R}^{T \times N}$ as follows:

$$\mathcal{B}(\mathbf{M}) = \frac{1}{\sqrt{N}} \begin{pmatrix} \left\langle \mathbf{x}_1^{(1)}, \mathbf{M}(\cdot, 1) \right\rangle & \cdots & \left\langle \mathbf{x}_1^{(N)}, \mathbf{M}(\cdot, 1) \right\rangle \\ \vdots & \ddots & \vdots \\ \left\langle \mathbf{x}_T^{(1)}, \mathbf{M}(\cdot, T) \right\rangle & \cdots & \left\langle \mathbf{x}_T^{(N)}, \mathbf{M}(\cdot, T) \right\rangle \end{pmatrix}, \qquad \forall \mathbf{M} \in \mathbb{R}^{d \times T}.$$

Notice that matrix $\mathbf{B}^*\mathbf{W}^*$ admits a singular value decomposition of the form: $\mathbf{B}^*\mathbf{W}^* = \mathbf{U}^*\mathbf{\Sigma}^*(\mathbf{V}^*)^\top$. Define $\mathbf{F} = \mathbf{U}^*(\cdot, [k]) \left( \mathbf{\Sigma}_{[k]}^* \right)^{1/2} \in \mathbb{R}^{d \times k}$, $\mathbf{G} = \mathbf{V}^*(\cdot, [k]) \left( \mathbf{\Sigma}_{[k]}^* \right)^{1/2} \in \mathbb{R}^{T \times k}$. Under this notation, the iterative variables $\mathbf{B}^{(\tau)}$ and $\mathbf{W}^{(\tau)}$ in the second stage of Algorithm 1 can be viewed as estimating $\mathbf{F}$ and $\mathbf{G}^\top$, respectively. To simplify exposition we aggregate the pairs of matrices $(\mathbf{B}, \mathbf{W})$, $(\mathbf{B}^{(\tau)}, \mathbf{W}^{(\tau)})$, $(\mathbf{F}, \mathbf{G})$, and $(\mathbf{F}, -\mathbf{G})$ into larger "lifted" matrices as follows:

$$\mathbf{Z} := \begin{pmatrix} \mathbf{B} \\ \mathbf{W}^\top \end{pmatrix}, \qquad \mathbf{Z}^{(\tau)} := \begin{pmatrix} \mathbf{B}^{(\tau)} \\ (\mathbf{W}^{(\tau)})^\top \end{pmatrix}, \qquad \mathbf{J} := \begin{pmatrix} \mathbf{F} \\ \mathbf{G} \end{pmatrix}, \qquad \widetilde{\mathbf{J}} := \begin{pmatrix} \mathbf{F} \\ -\mathbf{G} \end{pmatrix}. \tag{17}$$

Moreover, define orthogonal matrix

$$\mathbf{R} = \min_{\substack{\mathbf{R} \in \mathbb{R}^{k \times k} \\ \mathbf{R}\mathbf{R}^\top = \mathbf{I}_k}} \left\| \mathbf{Z} - \mathbf{J}\mathbf{R} \right\|_F^2, \tag{18}$$

and objective functions $f : \mathbb{R}^{d \times k} \times \mathbb{R}^{k \times T} \to \mathbb{R}_+$ and $g : \mathbb{R}^{d \times k} \times \mathbb{R}^{k \times T} \to \mathbb{R}_+$

$$f(\mathbf{B}, \mathbf{W}) := \frac{1}{2N} \sum_{t=1}^{T} \left\| \mathbf{y}_t - \mathbf{X}_t^\top \mathbf{B}\mathbf{w}_t \right\|^2, \quad g(\mathbf{B}, \mathbf{W}) := f(\mathbf{B}, \mathbf{W}) + \frac{1}{8} \left\| \mathbf{B}^\top \mathbf{B} - \mathbf{W}\mathbf{W}^\top \right\|_F^2.$$

Note that the update iteration of Algorithm 1 in phase II is based on the gradients $\nabla_{\mathbf{B}} g(\mathbf{B}, \mathbf{W})$ and $\nabla_{\mathbf{W}} g(\mathbf{B}, \mathbf{W})$ given by

$$\nabla_{\mathbf{B}} g(\mathbf{B}, \mathbf{W}) = \nabla_{\mathbf{B}} f(\mathbf{B}, \mathbf{W}) + \frac{1}{2} \mathbf{B} \left( \mathbf{B}^\top \mathbf{B} - \mathbf{W}\mathbf{W}^\top \right),$$

$$\nabla_{\mathbf{W}} g(\mathbf{B}, \mathbf{W}) = \nabla_{\mathbf{W}} f(\mathbf{B}, \mathbf{W}) + \frac{1}{2} \left( \mathbf{W}\mathbf{W}^\top - \mathbf{B}^\top \mathbf{B} \right) \mathbf{W},$$

where $\nabla_{\mathbf{B}} f(\mathbf{B}, \mathbf{W})$ and $\nabla_{\mathbf{W}} f(\mathbf{B}, \mathbf{W})$ have the following form:

$$\nabla_{\mathbf{B}} f(\mathbf{B}, \mathbf{W}) = \underbrace{\frac{1}{\sqrt{N}} \left( \mathbf{X}_1 \left[ \mathcal{B} \left( \mathbf{B}\mathbf{W} - \mathbf{B}^*\mathbf{W}^* \right) (1, \cdot) \right]^\top \quad \cdots \quad \mathbf{X}_T \left[ \mathcal{B} \left( \mathbf{B}\mathbf{W} - \mathbf{B}^*\mathbf{W}^* \right) (T, \cdot) \right]^\top \right) \mathbf{W}^\top}_{\nabla_{\mathbf{B}} \widetilde{f}(\mathbf{B}, \mathbf{W})}$$

$$+ \underbrace{\frac{1}{N} \left( \mathbf{X}_1 \mathbf{z}_1 \quad \cdots \quad \mathbf{X}_T \mathbf{z}_T \right) \mathbf{W}^\top}_{\mathbf{N}},$$

$$\nabla_{\mathbf{W}} f(\mathbf{B}, \mathbf{W}) = \mathbf{B}^\top \underbrace{\frac{1}{\sqrt{N}} \left( \mathbf{X}_1 \left[ \mathcal{B} \left( \mathbf{B}\mathbf{W} - \mathbf{B}^*\mathbf{W}^* \right) (1, \cdot) \right]^\top \quad \cdots \quad \mathbf{X}_T \left[ \mathcal{B} \left( \mathbf{B}\mathbf{W} - \mathbf{B}^*\mathbf{W}^* \right) (T, \cdot) \right]^\top \right)}_{\nabla_{\mathbf{W}} \widetilde{f}(\mathbf{B}, \mathbf{W})}$$

$$+ \mathbf{B}^\top \underbrace{\frac{1}{N} \left( \mathbf{X}_1 \mathbf{z}_1 \quad \cdots \quad \mathbf{X}_T \mathbf{z}_T \right)}_{\mathbf{N}},$$

where $\mathbf{z}_t := \left( z_t^{(1)}, \cdots, z_t^{(N)} \right)^\top$ for any $t \in [T]$. Here, we decompose the gradient $\nabla_{\mathbf{B}} f(\mathbf{B}, \mathbf{W})$ into the gradient of the bias-free loss function $\widetilde{f}$ (i.e., output data satisfies $y_t = \langle \mathbf{x}_t, \mathbf{B}^* \mathbf{w}_t^* \rangle$ for any $t \in [T]$) with respect to $\mathbf{B}$ and a corresponding bias noise matrix $\mathbf{N} \mathbf{W}^\top$. Similarly, we express $\nabla_{\mathbf{W}} f(\mathbf{B}, \mathbf{W})$ as the gradient of $\widetilde{f}$ with respect to $\mathbf{W}$ plus an associated bias noise term $\mathbf{B}^\top \mathbf{N}$.

To prove Theorem 5.1, we examine the optimization landscape of the function $g(\mathbf{Z}) := g(\mathbf{B}, \mathbf{W})$ in the lifted space $\mathbb{R}^{(d+T) \times k}$. Therefore, we introduce several key block matrix operators and definitions. Let $\mathrm{Sym} : \mathbb{R}^{d \times T} \to \mathbb{R}^{(d+T) \times (d+T)}$ be defined as

$$\mathrm{Sym}(\mathbf{M}) := \begin{pmatrix} \mathbf{0}_{d \times d} & \mathbf{M} \\ \mathbf{M}^\top & \mathbf{0}_{T \times T} \end{pmatrix}.$$

We note for future use that with this notation we have $\mathrm{Sym}(\mathbf{B}^* \mathbf{W}^*) = \frac{1}{2} \left( \mathbf{J} \mathbf{J}^\top - \widetilde{\mathbf{J}} \widetilde{\mathbf{J}}^\top \right)$. Given a block matrix $\mathbf{M} \in \mathbb{R}^{(d+T) \times (d+T)}$ partitioned as

$$\mathbf{M} = \begin{pmatrix} \mathbf{M}_{11} & \mathbf{M}_{12} \\ \mathbf{M}_{21} & \mathbf{M}_{22} \end{pmatrix}, \qquad \text{with} \quad \mathbf{M}_{11} \in \mathbb{R}^{d \times d}, \quad \mathbf{M}_{12} \in \mathbb{R}^{d \times T}, \quad \mathbf{M}_{21} \in \mathbb{R}^{T \times d}, \quad \mathbf{M}_{22} \in \mathbb{R}^{T \times T}.$$

We define the linear operators $\mathcal{P}_{\mathrm{diag}}$ and $\mathcal{P}_{\mathrm{off}}$ from $\mathbb{R}^{(d+T) \times (d+T)} \to \mathbb{R}^{(d+T) \times (d+T)}$ as follows

$$\mathcal{P}_{\mathrm{diag}}(\mathbf{M}) := \begin{pmatrix} \mathbf{M}_{11} & \mathbf{0}_{d \times T} \\ \mathbf{0}_{T \times d} & \mathbf{M}_{22} \end{pmatrix}, \qquad \mathcal{P}_{\mathrm{off}}(\mathbf{M}) := \begin{pmatrix} \mathbf{0}_{d \times d} & \mathbf{M}_{12} \\ \mathbf{M}_{21} & \mathbf{0}_{T \times T} \end{pmatrix}.$$

Our final piece of notation is a matrix-valued map which works over lifted matrices, which we call $\mathcal{C}$. The map $\mathcal{C} : \mathbb{R}^{d \times T} :\to \mathbb{R}^{d \times T}$ is defined as

$$\mathcal{C}(\mathbf{M}) := \frac{1}{\sqrt{N}} \left( \mathbf{X}_1 \left[ \mathcal{B}(\mathbf{M})(1, \cdot) \right]^\top \quad \cdots \quad \mathbf{X}_T \left[ \mathcal{B}(\mathbf{M})(T, \cdot) \right]^\top \right).$$

One can easily verify that this update has the following compact representation in terms of the lifted space

$$\nabla g(\mathbf{Z}) = \begin{pmatrix} \nabla_{\mathbf{B}} \widetilde{f}(\mathbf{B}, \mathbf{W}) \\ \left[ \nabla_{\mathbf{W}} \widetilde{f}(\mathbf{B}, \mathbf{W}) \right]^\top \end{pmatrix} + \frac{1}{2} \left( \mathcal{P}_{\mathrm{diag}} - \mathcal{P}_{\mathrm{off}} \right) \left( \mathbf{Z} \mathbf{Z}^\top \right) \mathbf{Z} + \begin{pmatrix} \mathbf{N} \mathbf{W}^\top \\ \mathbf{N}^\top \mathbf{B} \end{pmatrix}$$

$$= \underbrace{\begin{pmatrix} \mathbf{0} & \mathcal{C}(\mathbf{B} \mathbf{W} - \mathbf{B}^* \mathbf{W}^*) \\ \left[ \mathcal{C}(\mathbf{B} \mathbf{W} - \mathbf{B}^* \mathbf{W}^*) \right]^\top & \mathbf{0} \end{pmatrix} \mathbf{Z} + \frac{1}{2} \left( \mathcal{P}_{\mathrm{diag}} - \mathcal{P}_{\mathrm{off}} \right) \left( \mathbf{Z} \mathbf{Z}^\top \right) \mathbf{Z}}_{\nabla \widetilde{f}(\mathbf{Z})} + \begin{pmatrix} \mathbf{N} \mathbf{W}^\top \\ \mathbf{N}^\top \mathbf{B} \end{pmatrix}.$$

We then utilize the following Lemma A.3 and Lemma A.4 to prove that $\widetilde{f}(\mathbf{Z})$ satisfies RC.

**Lemma A.3.** *Suppose Assumption 3.2 holds. Then, for any $\mathbf{Z} \in \mathbb{R}^{(d+T) \times k}$ and $\mathbf{R} \in \mathbb{R}^{k \times k}$, the function $\widetilde{f}$ satisfies the following relation with L:*

$$\left\langle \nabla \widetilde{f}(\mathbf{Z}), \mathbf{Z} - \mathbf{J} \mathbf{R} \right\rangle \geq -\frac{\delta}{2} \left\| \mathbf{Z} \mathbf{Z}^\top - \mathbf{J} \mathbf{J}^\top \right\|_{\mathrm{F}} \left\| (\mathbf{Z} - \mathbf{J} \mathbf{R}) \mathbf{Z}^\top \right\|_{\mathrm{F}}$$
$$+ \frac{1}{2} \left\langle \nabla L(\mathbf{Z}), \mathbf{Z} - \mathbf{J} \mathbf{R} \right\rangle + \frac{1}{4 \left\| \mathbf{B}^* \mathbf{W}^* \right\|} \left\| \widetilde{\mathbf{J}} \widetilde{\mathbf{J}}^\top \mathbf{Z} \right\|_{\mathrm{F}}^2. \tag{19}$$

*Proof.* For simplicity, we define $\Delta := \mathbf{Z}\mathbf{Z}^\top - Sym\left(\mathbf{B}^*\mathbf{W}^*\right)$ and derive that

$$
\begin{aligned}
\nabla\widetilde{f}(\mathbf{Z}) = &\left[\begin{pmatrix} \mathbf{0} & \mathcal{C}\left(\mathbf{B}\mathbf{W} - \mathbf{B}^*\mathbf{W}^*\right) \\ \left[\mathcal{C}\left(\mathbf{B}\mathbf{W} - \mathbf{B}^*\mathbf{W}^*\right)\right]^\top & \mathbf{0} \end{pmatrix} - \mathcal{P}_{\text{off}}(\Delta)\right]\mathbf{Z} + \mathcal{P}_{\text{off}}\left(\mathbf{Z}\mathbf{Z}^\top\right)\mathbf{Z} \\
& - \mathcal{P}_{\text{off}}\left(\text{Sym}\left(\mathbf{B}^*\mathbf{W}^*\right)\right)\mathbf{Z} + \frac{1}{2}\mathcal{P}_{\text{diag}}\left(\mathbf{Z}\mathbf{Z}^\top\right)\mathbf{Z} - \frac{1}{2}\mathcal{P}_{\text{off}}\left(\mathbf{Z}\mathbf{Z}^\top\right)\mathbf{Z} \\
= &\left[\begin{pmatrix} \mathbf{0} & \mathcal{C}\left(\mathbf{B}\mathbf{W} - \mathbf{B}^*\mathbf{W}^*\right) \\ \left[\mathcal{C}\left(\mathbf{B}\mathbf{W} - \mathbf{B}^*\mathbf{W}^*\right)\right]^\top & \mathbf{0} \end{pmatrix} - \mathcal{P}_{\text{off}}(\Delta)\right]\mathbf{Z} + \frac{1}{2}\left(\mathcal{P}_{\text{diag}} + \mathcal{P}_{\text{off}}\right)\left(\mathbf{Z}\mathbf{Z}^\top\right)\mathbf{Z} \\
& - \text{Sym}\left(\mathbf{B}^*\mathbf{W}^*\right)\mathbf{Z} \\
= &\left[\begin{pmatrix} \mathbf{0} & \mathcal{C}\left(\mathbf{B}\mathbf{W} - \mathbf{B}^*\mathbf{W}^*\right) \\ \left[\mathcal{C}\left(\mathbf{B}\mathbf{W} - \mathbf{B}^*\mathbf{W}^*\right)\right]^\top & \mathbf{0} \end{pmatrix} - \mathcal{P}_{\text{off}}(\Delta)\right]\mathbf{Z} + \frac{1}{2}\left(\mathbf{Z}\mathbf{Z}^\top - 2\,\text{Sym}\left(\mathbf{B}^*\mathbf{W}^*\right)\right)\mathbf{Z}.
\end{aligned}
$$

Taking inner products of both sides of Eq. (20) yields that

$$
\left\langle\nabla\widetilde{f}(\mathbf{Z}), \mathbf{Z} - \mathbf{J}\mathbf{R}\right\rangle = \underbrace{\left\langle\left[\begin{pmatrix} \mathbf{0} & \mathcal{C}\left(\mathbf{B}\mathbf{W} - \mathbf{B}^*\mathbf{W}^*\right) \\ \left[\mathcal{C}\left(\mathbf{B}\mathbf{W} - \mathbf{B}^*\mathbf{W}^*\right)\right]^\top & \mathbf{0} \end{pmatrix} - \mathcal{P}_{\text{off}}(\Delta)\right]\mathbf{Z}, \mathbf{Z} - \mathbf{J}\mathbf{R}\right\rangle}_{\mathcal{I}} \tag{20}
$$
$$
+ \frac{1}{2}\underbrace{\left\langle\left(\mathbf{Z}\mathbf{Z}^\top - 2\,\text{Sym}\left(\mathbf{B}^*\mathbf{W}^*\right)\right)\mathbf{Z}, \mathbf{Z} - \mathbf{J}\mathbf{R}\right\rangle}_{\mathcal{II}}.
$$

Term $\mathcal{I}$ can be controlled by the RIP condition. One can notice that

$$
\begin{aligned}
\mathcal{I} = &\left\langle\mathcal{B}\left(\mathbf{B}\mathbf{W} - \mathbf{F}\mathbf{G}^\top\right), \mathcal{B}\left((\mathbf{B} - \mathbf{F}\mathbf{R})\mathbf{W}\right)\right\rangle + \left\langle\mathcal{B}\left(\mathbf{B}\mathbf{W} - \mathbf{F}\mathbf{G}^\top\right), \mathcal{B}\left(\mathbf{B}\left(\mathbf{W} - \mathbf{R}^\top\mathbf{G}^\top\right)\right)\right\rangle \\
& - \left\langle\mathcal{P}_{\text{off}}(\Delta)\mathbf{Z}, \mathbf{Z} - \mathbf{J}\mathbf{R}\right\rangle \\
= &\left\langle\mathcal{B}\left(\mathbf{B}\mathbf{W} - \mathbf{F}\mathbf{G}^\top\right), \mathcal{B}\left((\mathbf{B} - \mathbf{F}\mathbf{R})\mathbf{W}\right)\right\rangle + \left\langle\mathcal{B}\left(\mathbf{B}\mathbf{W} - \mathbf{F}\mathbf{G}^\top\right), \mathcal{B}\left(\mathbf{B}\left(\mathbf{W} - \mathbf{R}^\top\mathbf{G}^\top\right)\right)\right\rangle \\
& - \left\langle\mathbf{B}\mathbf{W} - \mathbf{F}\mathbf{G}^\top, (\mathbf{B} - \mathbf{F}\mathbf{R})\mathbf{W}\right\rangle - \left\langle\mathbf{B}\mathbf{W} - \mathbf{F}\mathbf{G}^\top, \mathbf{B}\left(\mathbf{W} - \mathbf{R}^\top\mathbf{G}^\top\right)\right\rangle \\
\overset{(a)}{\geq} & - \delta\left(\left\|\mathbf{B}\mathbf{W} - \mathbf{F}\mathbf{G}^\top\right\|_{\mathrm{F}}\left\|(\mathbf{B} - \mathbf{F}\mathbf{R})\mathbf{W}\right\|_{\mathrm{F}} + \left\|\mathbf{B}\mathbf{W} - \mathbf{F}\mathbf{G}^\top\right\|_{\mathrm{F}}\left\|\mathbf{B}\left(\mathbf{W} - \mathbf{R}^\top\mathbf{G}^\top\right)\right\|_{\mathrm{F}}\right) \\
\geq & - \delta\left\|\mathbf{Z}\mathbf{Z}^\top - \mathbf{J}\mathbf{J}^\top\right\|_{\mathrm{F}}\left\|(\mathbf{Z} - \mathbf{J}\mathbf{R})\mathbf{Z}^\top\right\|_{\mathrm{F}},
\end{aligned}
\tag{21}
$$

where (a) follows from Lemma D.1.

We now relate the second term to the gradient of $L$. According to the structure of $\mathbf{J}$ and $\widetilde{\mathbf{J}}$, we obtain

$$
\begin{aligned}
\mathcal{II} \overset{(b)}{=} &\left\langle\left(\mathbf{Z}\mathbf{Z}^\top - \mathbf{J}\mathbf{J}^\top\right)\mathbf{Z}, \mathbf{Z} - \mathbf{J}\mathbf{R}\right\rangle + \left\langle\widetilde{\mathbf{J}}\widetilde{\mathbf{J}}^\top\mathbf{Z}, \mathbf{Z} - \mathbf{J}\mathbf{R}\right\rangle \\
\overset{(c)}{=} &\left\langle\left(\mathbf{Z}\mathbf{Z}^\top - \mathbf{J}\mathbf{J}^\top\right)\mathbf{Z}, \mathbf{Z} - \mathbf{J}\mathbf{R}\right\rangle + \left\|\widetilde{\mathbf{J}}^\top\mathbf{Z}\right\|_{\mathrm{F}}^2 \\
\geq &\left\langle\left(\mathbf{Z}\mathbf{Z}^\top - \mathbf{J}\mathbf{J}^\top\right)\mathbf{Z}, \mathbf{Z} - \mathbf{J}\mathbf{R}\right\rangle + \frac{1}{\left\|\widetilde{\mathbf{J}}\right\|^2}\left\|\widetilde{\mathbf{J}}\widetilde{\mathbf{J}}^\top\mathbf{Z}\right\|_{\mathrm{F}}^2 \\
= &\left\langle\nabla L(\mathbf{Z}), \mathbf{Z} - \mathbf{J}\mathbf{R}\right\rangle + \frac{1}{2\left\|\mathbf{B}^*\mathbf{W}^*\right\|}\left\|\widetilde{\mathbf{J}}\widetilde{\mathbf{J}}^\top\mathbf{Z}\right\|_{\mathrm{F}}^2,
\end{aligned}
\tag{22}
$$

where (b) is derived from the fact that $2\,\text{Sym}(\mathbf{B}^*\mathbf{W}^*) = \mathbf{J}\mathbf{J}^\top - \widetilde{\mathbf{J}}\widetilde{\mathbf{J}}^\top$, (c) follows from $\widetilde{\mathbf{J}}^\top\mathbf{J} = \mathbf{0}_{\mathbf{k}\times\mathbf{k}}$. We complete the proof by applying Eqs. (21) and (22) to Eq. (20). $\qquad\square$

**Lemma A.4.** *Suppose Assumption 3.2 holds. Then, for any* $\mathbf{Z} \in \mathbb{R}^{(d+T)\times k}$ *satisfying* $\text{dist}(\mathbf{Z}, \mathbf{J}) \leq \frac{1}{4}\|\mathbf{J}\|$, *we have that*

$$
(2 + 20\delta)\left\|\mathbf{Z}\mathbf{Z}^\top - \mathbf{J}\mathbf{J}^\top\right\|_{\mathrm{F}}^2 + \frac{1}{\left\|\mathbf{B}^*\mathbf{W}^*\right\|}\left\|\widetilde{\mathbf{J}}\widetilde{\mathbf{J}}^\top\mathbf{Z}\right\|_{\mathrm{F}}^2 \geq \frac{1}{\left\|\mathbf{B}^*\mathbf{W}^*\right\|}\left\|\nabla\widetilde{f}(\mathbf{Z})\right\|_{\mathrm{F}}^2. \tag{23}
$$

*Proof.* Defining $\Delta := \mathbf{Z}\mathbf{Z}^\top - \mathrm{Sym}(\mathbf{B}^*\mathbf{W}^*)$, we obtain the inequality:

$$
\frac{1}{\|\mathbf{B}^*\mathbf{W}^*\|} \left\|\nabla \widetilde{f}(\mathbf{Z})\right\|_F^2 \overset{(a)}{\leq} \underbrace{\frac{4}{\|\mathbf{Z}\|^2} \left\|\left[\begin{pmatrix} \mathbf{0} & \mathcal{C}\left(\mathbf{B}\mathbf{W} - \mathbf{B}^*\mathbf{W}^*\right) \\ \left[\mathcal{C}\left(\mathbf{B}\mathbf{W} - \mathbf{B}^*\mathbf{W}^*\right)\right]^\top & \mathbf{0} \end{pmatrix} - \mathcal{P}_{\mathrm{off}}(\Delta)\right]\mathbf{Z}\right\|_F^2}_{\mathcal{I}}
$$
$$
+ \underbrace{\frac{1}{2\|\mathbf{B}^*\mathbf{W}^*\|} \left\|\left(\mathbf{Z}\mathbf{Z}^\top - 2\,\mathrm{Sym}\left(\mathbf{B}^*\mathbf{W}^*\right)\right)\mathbf{Z}\right\|_F^2}_{\mathcal{II}},
$$
(24)

where (a) follows from the Cauchy–Schwarz inequality and the assumption that $\mathrm{dist}(\mathbf{Z}, \mathbf{J}) \leq \frac{1}{4}\|\mathbf{J}\|$. Term $\mathcal{I}$ can be bounded via the RIP condition. We observe that:

$$
\mathcal{I} \overset{(a)}{\leq} 4\left\|\begin{pmatrix} \mathbf{0} & \mathcal{C}\left(\mathbf{B}\mathbf{W} - \mathbf{B}^*\mathbf{W}^*\right) \\ \left[\mathcal{C}\left(\mathbf{B}\mathbf{W} - \mathbf{B}^*\mathbf{W}^*\right)\right]^\top & \mathbf{0} \end{pmatrix} - \mathcal{P}_{\mathrm{off}}(\Delta)\right\|_F^2
$$
$$
= 4\|\mathcal{P}_{\mathrm{off}}(\Delta)\|_F^2 - 16\left\|\mathcal{B}\left(\mathbf{B}\mathbf{W} - \mathbf{F}\mathbf{G}^\top\right)\right\|_F^2 + 4\left\|\begin{pmatrix} \mathbf{0} & \mathcal{C}\left(\mathbf{B}\mathbf{W} - \mathbf{B}^*\mathbf{W}^*\right) \\ \left[\mathcal{C}\left(\mathbf{B}\mathbf{W} - \mathbf{B}^*\mathbf{W}^*\right)\right]^\top & \mathbf{0} \end{pmatrix}\right\|_F^2
$$
(25)
$$
\overset{(b)}{\leq} (8\delta - 4)\|\mathcal{P}_{\mathrm{off}}(\Delta)\|_F^2 + 8\|\mathcal{C}\left(\mathbf{B}\mathbf{W} - \mathbf{B}^*\mathbf{W}^*\right)\|_F^2,
$$

where (a) uses the inequality $\|\mathbf{A}\mathbf{Z}\|_F^2 \leq \|\mathbf{Z}\|^2\|\mathbf{A}\|_F^2$ for any matrix $\mathbf{A} \in \mathbb{R}^{(d+T)\times(d+T)}$, and (b) is derived from the RIP condition. We now control $\|\mathcal{C}\left(\mathbf{B}\mathbf{W} - \mathbf{B}^*\mathbf{W}^*\right)\|_F$ above. Using the variational form of the Frobenius norm,

$$
\|\mathcal{C}\left(\mathbf{B}\mathbf{W} - \mathbf{B}^*\mathbf{W}^*\right)\|_F = \sup_{\mathbf{M} \in \mathbb{R}^{d\times T}, \|\mathbf{M}\|_F \leq 1} \langle \mathcal{B}\left(\mathbf{B}\mathbf{W} - \mathbf{B}^*\mathbf{W}^*\right), \mathcal{B}(\mathbf{M})\rangle
$$
$$
\leq (1 + \delta)\|\mathbf{B}\mathbf{W} - \mathbf{B}^*\mathbf{W}^*\|_F.
$$
(26)

Applying Eq. (26) to Eq. (25), we obtain

$$
\mathcal{I} \leq 20\delta\|\mathcal{P}_{\mathrm{off}}(\Delta)\|_F^2.
$$
(27)

We now provide an upperbound for the second term.

$$
\mathcal{II} \leq \frac{1}{\|\mathbf{B}^*\mathbf{W}^*\|}\left(\left\|\left(\mathbf{Z}\mathbf{Z}^\top - \mathbf{J}\mathbf{J}^\top\right)\mathbf{Z}\right\|_F^2 + \left\|\widetilde{\mathbf{J}}\widetilde{\mathbf{J}}^\top\mathbf{Z}\right\|_F^2\right)
$$
$$
\overset{(c)}{\leq} \frac{2}{\|\mathbf{Z}\|^2}\left\|\left(\mathbf{Z}\mathbf{Z}^\top - \mathbf{J}\mathbf{J}^\top\right)\mathbf{Z}\right\|_F^2 + \frac{1}{\|\mathbf{B}^*\mathbf{W}^*\|}\left\|\widetilde{\mathbf{J}}\widetilde{\mathbf{J}}^\top\mathbf{Z}\right\|_F^2
$$
(28)
$$
\leq 2\left\|\mathbf{Z}\mathbf{Z}^\top - \mathbf{J}\mathbf{J}^\top\right\|_F^2 + \frac{1}{\|\mathbf{B}^*\mathbf{W}^*\|}\left\|\widetilde{\mathbf{J}}\widetilde{\mathbf{J}}^\top\mathbf{Z}\right\|_F^2,
$$

where (c) is derived from the assumption that $\mathrm{dist}(\mathbf{Z}, \mathbf{J}) \leq \frac{1}{4}\|\mathbf{J}\|$. Furthermore, since

$$
\mathcal{P}_{\mathrm{off}}(\Delta) = \frac{1}{2}\left(\mathbf{Z}\mathbf{Z}^\top - \mathbf{J}\mathbf{J}^\top\right) - \frac{1}{2}\begin{pmatrix} \mathbf{I}_{d\times d} & \mathbf{0}_{d\times T} \\ \mathbf{0}_{T\times d} & -\mathbf{I}_{T\times T} \end{pmatrix}\left(\mathbf{Z}\mathbf{Z}^\top - \mathbf{J}\mathbf{J}^\top\right)\begin{pmatrix} \mathbf{I}_{d\times d} & \mathbf{0}_{d\times T} \\ \mathbf{0}_{T\times d} & -\mathbf{I}_{T\times T} \end{pmatrix},
$$

which implicates that $\|\mathcal{P}_{\mathrm{off}}(\Delta)\|_F^2 \leq \|\mathbf{Z}\mathbf{Z}^\top - \mathbf{J}\mathbf{J}^\top\|_F^2$. Therefore, applying Eqs. (27) and (28) to Eq. (24), we complete the proof. $\square$

We have now established the necessary groundwork for the final convergence proof of Phase II. This leads to the main theorem for this phase.

**Theorem A.5.** *At the phase II of Algorithm 1, assuming the solution* $(\mathbf{B}^{(K_1/2)}, \mathbf{W}^{(K_1/2)})$ *obeys*

$$
\mathrm{dist}\left(\mathbf{Z}^{(K_1/2)}, \mathbf{J}\right) \leq \frac{\sigma_k(\mathbf{J})}{8},
$$
(29)

*then we have*

$$\left[\text{dist}\left(\mathbf{Z}^{(K_1)}, \mathbf{J}\right)\right]^2 \leq \left(1 - \frac{\sigma_k^2(\mathbf{J})}{8}\eta_2\right)^{K_1/2} \text{dist}\left(\mathbf{Z}^{(K_1/2)}, \mathbf{J}\right)$$
$$+ \mathcal{O}\left(\sigma^2 \log^3\left(\delta^{-1}\right) \log^2(T) \cdot \frac{\text{tr}\left(\mathbf{\Sigma}^*\right)\left(\max\{d, T\} + d\right)}{\sigma_k^2\left(\mathbf{\Sigma}^*\right) N}\right).$$

*Proof.* We begin by demonstrating that the function $\widetilde{f}$ satisfies the RC A.2. Based on Lemma A.3, we use Eq. (19) and Cauchy-Schwarz inequality to obtain that,

$$\left\langle \nabla \widetilde{f}(\mathbf{Z}), \mathbf{Z} - \mathbf{JR} \right\rangle \geq \frac{1}{2}\left\langle \nabla L(\mathbf{Z}), \mathbf{Z} - \mathbf{JR}\right\rangle + \frac{1}{4\|\mathbf{B}^*\mathbf{W}^*\|}\left\|\widetilde{\mathbf{J}}\widetilde{\mathbf{J}}^\top \mathbf{Z}\right\|_F^2$$
$$- \frac{\delta}{4}\left(\left\|\mathbf{ZZ}^\top - \mathbf{JJ}^\top\right\|_F^2 + \left\|(\mathbf{Z} - \mathbf{JR})\mathbf{Z}^\top\right\|_F^2\right). \tag{30}$$

Recall the definition of $\mathbf{R}$ in Eq. (18). By assumption $\text{dist}\left(\mathbf{Z}^{(\tau)}, \mathbf{J}\right) \leq \frac{\sigma_k(\mathbf{J})}{4}$, so we can apply Eq. (16) to Eq. (30), which yields that

$$\left\langle \nabla \widetilde{f}\left(\mathbf{Z}^{(\tau)}\right), \mathbf{Z}^{(\tau)} - \mathbf{JR}^{(\tau)}\right\rangle \geq \frac{\sigma_k^2(\mathbf{J})}{8}\left\|\mathbf{Z}^{(\tau)} - \mathbf{JR}^{(\tau)}\right\|_F^2 + \frac{1-2\delta}{8}\left\|\mathbf{Z}^{(\tau)}\left(\mathbf{Z}^{(\tau)}\right)^\top - \mathbf{JJ}^\top\right\|_F^2$$
$$+ \frac{1-10\delta}{40}\left\|\left(\mathbf{Z}^\tau - \mathbf{JR}^{(\tau)}\right)\mathbf{Z}^\top\right\|_F^2$$
$$+ \frac{1}{4\|\mathbf{B}^*\mathbf{W}^*\|}\left\|\widetilde{\mathbf{J}}\widetilde{\mathbf{J}}^\top \mathbf{Z}^{(\tau)}\right\|_F^2. \tag{31}$$

Applying the result of Lemma A.4 to Eq. (31), we can further obtain that

$$\left\langle \nabla \widetilde{f}\left(\mathbf{Z}^{(\tau)}\right), \mathbf{Z}^{(\tau)} - \mathbf{JR}^{(\tau)}\right\rangle \geq \frac{\sigma_k^2(\mathbf{J})}{8}\left\|\mathbf{Z}^{(\tau)} - \mathbf{JR}^{(\tau)}\right\|_F^2 + \frac{1-2\delta}{16(1+10\delta)\|\mathbf{B}^*\mathbf{W}^*\|}\left\|\nabla \widetilde{f}\left(\mathbf{Z}^{(\tau)}\right)\right\|_F^2$$
$$\overset{(a)}{\geq} \frac{\sigma_k^2(\mathbf{J})}{8}\left\|\mathbf{Z}^{(\tau)} - \mathbf{JR}^{(\tau)}\right\|_F^2 + \frac{1}{64\|\mathbf{B}^*\mathbf{W}^*\|}\left\|\nabla \widetilde{f}\left(\mathbf{Z}^{(\tau)}\right)\right\|_F^2, \tag{32}$$

where (a) is derived from that $\delta \leq \frac{1}{10}$. Therefore, considering the iteration of Algorithm 1 at the phase II, we have

$$\left\|\mathbf{Z}^{(\tau+1)} - \mathbf{JR}^{(\tau+1)}\right\|_F^2 \overset{(b)}{\leq} \left\|\mathbf{Z}^{(\tau+1)} - \mathbf{JR}^{(\tau)}\right\|_F^2$$
$$= \left\|\mathbf{Z}^{(\tau)} - \mathbf{JR}^{(\tau)}\right\|_F^2 - 2\eta_2\left\langle \nabla g\left(\mathbf{Z}^{(\tau)}\right), \mathbf{Z}^{(\tau)} - \mathbf{JR}^{(\tau)}\right\rangle + \eta_2^2\left\|\nabla g\left(\mathbf{Z}^{(\tau)}\right)\right\|_F^2$$
$$\overset{(c)}{\leq} \left\|\mathbf{Z}^{(\tau)} - \mathbf{JR}^{(\tau)}\right\|_F^2 - 2\eta_2\left\langle \nabla \widetilde{f}\left(\mathbf{Z}^{(\tau)}\right), \mathbf{Z}^{(\tau)} - \mathbf{JR}^{(\tau)}\right\rangle + 2\eta_2^2\left\|\nabla \widetilde{f}\left(\mathbf{Z}^{(\tau)}\right)\right\|_F^2$$
$$- 2\eta_2\left\langle \begin{pmatrix}\mathbf{N}\left(\mathbf{W}^{(\tau)}\right)^\top \\ \mathbf{N}^\top \mathbf{B}^{(\tau)}\end{pmatrix}, \mathbf{Z}^{(\tau)} - \mathbf{JR}^{(\tau)}\right\rangle$$
$$+ 2\eta_2^2\left\|\begin{pmatrix}\mathbf{N}\left(\mathbf{W}^{(\tau)}\right)^\top \\ \mathbf{N}^\top \mathbf{B}^{(\tau)}\end{pmatrix}\right\|_F^2$$
$$\overset{(d)}{\leq} \left(1 - \frac{\sigma_k^2(\mathbf{J})}{8}\eta_2\right)\left\|\mathbf{Z}^{(\tau)} - \mathbf{JR}^{(\tau)}\right\|_F^2 - \left(\frac{\eta_2}{32\|\mathbf{B}^*\mathbf{W}^*\|} - 2\eta_2^2\right)\left\|\nabla \widetilde{f}\left(\mathbf{Z}^{(\tau)}\right)\right\|_F^2$$
$$+ \left(\frac{8\eta_2}{\sigma_k^2(\mathbf{J})} + 2\eta_2^2\right)\left\|\begin{pmatrix}\mathbf{N}\left(\mathbf{W}^{(\tau)}\right)^\top \\ \mathbf{N}^\top \mathbf{B}^{(\tau)}\end{pmatrix}\right\|_F^2, \tag{33}$$

for any $\mathbf{Z}^{(\tau)}$ satisfies $\text{dist}\left(\mathbf{Z}^{(\tau)}, \mathbf{J}\right) \leq \frac{\sigma_k(\mathbf{J})}{4}$, where (b) follows from the optimality condition of $\mathbf{R}^{(\tau+1)}$ given $\mathbf{Z}^{(\tau+1)}$; (c) is a direct application of the Cauchy–Schwarz inequality; and (d) is derived from combining Eq. (32) with the

Cauchy–Schwarz inequality. Under the hyper-parameter setting $\eta_2 \leq \min\left\{\frac{1}{64\|\mathbf{B}^*\mathbf{W}^*\|}, \frac{4}{\sigma_k^2(\mathbf{J})}\right\}$, Eq. (33) implies the following: if $\text{dist}\left(\mathbf{Z}^{(K_1/2)}, \mathbf{J}\right) \leq \frac{\sigma_k(\mathbf{J})}{8}$, then the inequality $\text{dist}\left(\mathbf{Z}^{(\tau)}, \mathbf{J}\right) \leq \frac{\sigma_k(\mathbf{J})}{4}$ holds for all $\tau \geq K_1/2$. Moreover, it ensures that the distance $\text{dist}\left(\mathbf{Z}^{(K_1/2+\tau)}, \mathbf{J}\right)$ after $K_1/2$ additional iterations satisfies

$$
\begin{aligned}
\left[\text{dist}\left(\mathbf{Z}^{(K_1)}, \mathbf{J}\right)\right]^2 &\leq \left(1 - \frac{\sigma_k^2(\mathbf{J})}{8}\eta_2\right)^{K_1/2} \text{dist}\left(\mathbf{Z}^{(K_1/2)}, \mathbf{J}\right) + \mathcal{O}\left(\frac{\text{tr}\left(\mathbf{\Sigma}^*\right)}{\sigma_k^2\left(\mathbf{\Sigma}^*\right)} \cdot \left\|\begin{pmatrix}\mathbf{0}_{\mathbf{d}\times\mathbf{d}} & \mathbf{N} \\ \mathbf{N}^\top & \mathbf{0}_{\mathbf{T}\times\mathbf{T}}\end{pmatrix}\right\|^2\right) \\
&\overset{(e)}{\leq} \left(1 - \frac{\sigma_k^2(\mathbf{J})}{8}\eta_2\right)^{K_1/2} \text{dist}\left(\mathbf{Z}^{(K_1/2)}, \mathbf{J}\right) \\
&\quad + \mathcal{O}\left(\sigma^2 \log^3\left(\delta^{-1}\right)\log^2(T) \cdot \frac{\text{tr}\left(\mathbf{\Sigma}^*\right)\left(\max\{d, T\} + d\right)}{\sigma_k^2\left(\mathbf{\Sigma}^*\right)N}\right),
\end{aligned}
\tag{34}
$$

where (e) uses the bound

$$
\|\mathbf{N}\| \leq \mathcal{O}\left(\frac{\sigma \log^{\frac{3}{2}}\left(\delta^{-1}\right)\log(T)}{\sqrt{N}} \cdot \left(\max\{d, T\} + d\right)^{\frac{1}{2}}\right),
$$

which is obtained from Lemma B.1. □

### A.1.4. PROOF OF THEOREM 5.1

Noting that Eq. (14) in Theorem A.1 satisfies the condition of Theorem A.5, we complete the proof by combining Theorem A.5 with Theorem A.1 directly.

*Proof of Corollary 5.3.* Notice that

$$
\begin{aligned}
\frac{1}{T}\sum_{t=1}^{T}\|\widehat{\mathbf{v}}_t - \mathbf{v}_t^*\|^2 &= \frac{1}{T}\left\|\widehat{\mathbf{B}}\widehat{\mathbf{W}} - \mathbf{FR}(\mathbf{GR})^\top\right\|_F^2 \\
&\leq \frac{2}{T}\left(\left\|(\widehat{\mathbf{B}} - \mathbf{FR})\widehat{\mathbf{W}}\right\|_F^2 + \left\|\mathbf{FR}\left(\widehat{\mathbf{W}} - (\mathbf{GR})^\top\right)\right\|_F^2\right) \\
&\leq \frac{2\sigma_1(\mathbf{\Sigma}^*)}{T}\left[\text{dist}\left(\begin{pmatrix}\widehat{\mathbf{B}} \\ \widehat{\mathbf{W}}^\top\end{pmatrix}, \begin{pmatrix}\mathbf{F} \\ \mathbf{G}\end{pmatrix}\right)\right]^2,
\end{aligned}
\tag{35}
$$

where

$$
\mathbf{F} = \mathbf{U}^*(\cdot, [k])(\mathbf{\Sigma}_{[k]}^*)^{1/2} \in \mathbb{R}^{d\times k}, \quad \mathbf{G} = \mathbf{V}^*(\cdot, [k])(\mathbf{\Sigma}_{[k]}^*)^{1/2} \in \mathbb{R}^{T\times k}.
$$

Therefore, combining Eq. (35) with Theorem 5.1, we complete the proof. □

## A.2. Proof of Theorem 5.4

For learning the future task parameter $\mathbf{w}_{T+1}^*$, we employ a stochastic gradient descent (SGD) procedure with a geometrically decaying step size schedule (Algorithm 2). The step size of our algorithm remains constant for the first $K_2' + h$ iterations, where $h$ denotes the length of the middle phase and $K_2' := \lfloor K_2 - h/\log(K_2 - h)\rfloor$. Subsequently, the step size is halved every $K_2'$ steps. Specifically, the step size schedule is defined as follows:

$$
\eta_\tau = \begin{cases} \eta, & 0 \leq \tau \leq K_2' + h, \\ \eta/2^l, & K_2' + h < \tau \leq K_2, \ l = \lfloor (\tau - h)/K_2' \rfloor, \end{cases}
$$

The combination of warm-up and learning rate decay has become a prevalent technique in deep learning optimization (Goyal et al., 2017). Among decay strategies, geometric schedules have shown superior empirical efficiency compared to polynomial decay, as they effectively balance rapid learning in early stages with stable refinement in late stages (Ge et al., 2019). Building on these advantages, we design our step size schedule to include an initial constant phase followed by a geometric decay stage.

---

**Algorithm 2** Stochastic Gradient Descent (SGD)

---

**Input:** Initial weight: $\mathbf{w}_{T+1}^{(0)}$. The estimator generated by Algorithm 1: $\widehat{\mathbf{B}}$. Initial step-size: $\eta_0$. Total sample size: $K_2$. Middle phase length: $h$. Decaying phase length: $K_2' := \lfloor (K_2 - h)/\log(K_2 - h) \rfloor$.

**Output:** $\mathbf{w}_{T+1}^{(K_2-1)}$.

1: **while** $\tau < K_2 - 1$ **do**
2:    **if** $\tau + 1 > h$ and $(\tau + 1 - h) \mod K_2' = 0$ **then**
3:       $\eta_0 \leftarrow \eta_0/2$.
4:    **end if**
5:    Sample a fresh data $(\mathbf{x}_{T+1}^{(\tau+1)}, y_{T+1}^{(\tau+1)})$.
6:

$$\mathbf{w}_{T+1}^{(\tau+1)} \leftarrow \mathbf{w}_{T+1}^{(\tau)} - \eta_0 \left( \left( \mathbf{x}_{T+1}^{(\tau+1)} \right)^\top \widehat{\mathbf{B}} \mathbf{w}_{T+1}^{(\tau)} - y_{T+1}^{(\tau+1)} \right) \widehat{\mathbf{B}}^\top \mathbf{x}_{T+1}^{(\tau+1)}.$$

7: **end while**

---

During the training phase for future task $T + 1$, Algorithm 2 performs online iterative updates using the gradient of the following empirical loss function:

$$\widehat{L}(\mathbf{w}_{T+1}) := \frac{1}{2} \left( \mathbf{x}_{T+1}^\top \widehat{\mathbf{B}} \mathbf{w}_{T+1} - y_{T+1} \right)^2. \tag{36}$$

The loss considered in this section resembles that of a linear model. However, according to the expression in Eq. (36), this loss additionally contains an irreducible error between $\widehat{\mathbf{B}}$ and $\mathbf{B}^*$. Consequently, compared with the typical dynamic of SGD with decaying step size in ordinary linear models (Wu et al., 2022), the dynamic of Algorithm 2 includes a component generated by additive noise that further involves stochastic noise correlated with the covariate vector $\mathbf{x}_{T+1}$. We resolve this correlation challenge by employing a recursive analysis technique. Moreover, in contrast to existing theoretical studies on transfer learning, our analysis requires only weaker moment assumptions.

According to the assumption for the output data $y_{T+1}$, we have

$$y_{T+1} = \mathbf{x}_{T+1}^\top \mathbf{B}^* \mathbf{w}_{T+1}^* + z_{T+1} = \mathbf{x}_{T+1}^\top \mathbf{F} \mathbf{P} \mathbf{w}_{T+1}^* + z_{T+1},$$

where $\mathbf{P} \in \mathbb{R}^{k \times k}$ is an invertible matrix, and $\mathbf{F} \in \mathbb{R}^{d \times k}$ has been defined in section A.1.3. With a slight abuse of notation, we let $\mathbf{w}_{T+1}^*$ represent $\mathbf{P} \mathbf{w}_{T+1}^*$ hereafter. Therefore, rewrite Eq. (36) as follows,

$$\begin{aligned} \widehat{L}(\mathbf{w}_{T+1}) =& \frac{1}{2} \left[ \mathbf{x}_{T+1}^\top \left( \widehat{\mathbf{B}} \mathbf{w}_{T+1} - \mathbf{F} \mathbf{w}_{T+1}^* \right) - z_{T+1} \right]^2 \\ =& \frac{1}{2} \left[ \mathbf{x}_{T+1}^\top \widehat{\mathbf{B}} \left( \mathbf{w}_{T+1} - \mathbf{R} \mathbf{w}_{T+1}^* \right) + \mathbf{x}_{T+1}^\top \left( \widehat{\mathbf{B}} \mathbf{R} - \mathbf{F} \right) \mathbf{w}_{T+1}^* - z_{T+1} \right]^2, \end{aligned}$$

where orthogonal matrix $\mathbf{R} \in \mathbb{R}^{k \times k}$ satisfies

$$\mathbf{R} := \underset{\substack{\mathbf{Q} \in \mathbb{R}^{k \times k} \\ \mathbf{Q} \mathbf{Q}^\top = \mathbf{I}_k}}{\operatorname{argmin}} \left\| \widehat{\mathbf{B}} \mathbf{Q} - \mathbf{F} \right\|_{\mathrm{F}}^2.$$

The update rule of Algorithm 2 can be written as

$$\mathbf{w}_{T+1}^{(\tau+1)} = \mathbf{w}_{T+1}^{(\tau)} - \eta_\tau \left[ \left( \mathbf{x}_{T+1}^{(\tau+1)} \right)^\top \widehat{\mathbf{B}} \left( \mathbf{w}_{T+1}^{(\tau)} - \mathbf{R} \mathbf{w}_{T+1}^* \right) + \zeta_{T+1}^{(\tau+1)} \right] \widehat{\mathbf{B}}^\top \mathbf{x}_{T+1}^{(\tau+1)}, \tag{37}$$

for any $\tau + 1 \in [K_2]$, where $\zeta_{T+1}^{(\tau+1)} = \left( \mathbf{x}_{T+1}^{(\tau+1)} \right)^\top \left( \widehat{\mathbf{B}} \mathbf{R} - \mathbf{F} \right) \mathbf{w}_{T+1}^* - z_{T+1}^{(\tau+1)}$. For simplicity, we define $\mathbf{E} := \widehat{\mathbf{B}} \mathbf{R} - \mathbf{F}$ and $\widehat{\mathbf{x}}_{T+1} := \widehat{\mathbf{B}}^\top \mathbf{x}_{T+1}$. Now consider the centered SGD iterates $\widehat{\mathbf{w}}_{T+1}^{(\tau+1)} := \mathbf{w}_{T+1}^{(\tau+1)} - \mathbf{R} \mathbf{w}_{T+1}^*$ with $\mathbf{w}_{T+1}^{(\tau+1)}$ given by Eq. (37), then the centered iterates are updated by

$$\widehat{\mathbf{w}}_{T+1}^{(\tau+1)} = \widehat{\mathbf{w}}_{T+1}^{(\tau)} - \eta_\tau \left[ \left( \widehat{\mathbf{x}}_{T+1}^{(\tau+1)} \right)^\top \widehat{\mathbf{w}}_{T+1}^{(\tau)} + \zeta_{T+1}^{(\tau+1)} \right] \widehat{\mathbf{x}}_{T+1}^{(\tau+1)}, \tag{38}$$

for any $\tau + 1 \in [K_2]$. According to Eq. (38), the centered SGD iterates can be interpreted——with a slight abuse of probability spaces——as the sum of two random processes:

$$\widehat{\mathbf{w}}_{T+1}^{(\tau+1)} = \widehat{\mathbf{w}}_{T+1,\text{bias}}^{(\tau+1)} + \widehat{\mathbf{w}}_{T+1,\text{var}}^{(\tau+1)},$$

where

$$\begin{cases} \widehat{\mathbf{w}}_{T+1,\text{var}}^{(\tau+1)} = \left(\mathbf{I}_k - \eta_\tau \widehat{\mathbf{x}}_{T+1}^{(\tau+1)} \left(\widehat{\mathbf{x}}_{T+1}^{(\tau+1)}\right)^\top\right) \widehat{\mathbf{w}}_{T+1,\text{var}}^{(\tau)} - \eta_\tau \zeta_{T+1}^{(\tau+1)} \widehat{\mathbf{x}}_{T+1}^{(\tau+1)}, \\ \widehat{\mathbf{w}}_{T+1,\text{var}}^{(0)} = \mathbf{0}, \end{cases} \tag{39}$$

and

$$\begin{cases} \widehat{\mathbf{w}}_{T+1,\text{bias}}^{(\tau+1)} = \left(\mathbf{I}_k - \eta_\tau \widehat{\mathbf{x}}_{T+1}^{(\tau+1)} \left(\widehat{\mathbf{x}}_{T+1}^{(\tau+1)}\right)^\top\right) \widehat{\mathbf{w}}_{T+1,\text{bias}}^{(\tau)}, \\ \widehat{\mathbf{w}}_{T+1,\text{bias}}^{(0)} = \mathbf{w}_{T+1}^{(0)} - \mathbf{R}\mathbf{w}_{T+1}^*. \end{cases} \tag{40}$$

**Notations.** Before proceeding with the subsequent analysis, we summarize the linear operators acting on symmetric matrices that will be employed in the proof.

$$\mathcal{I} := \mathbf{I}_k \otimes \mathbf{I}_k, \quad \mathcal{H} := \mathbb{E}\left[\left(\widehat{\mathbf{x}}_{T+1}\widehat{\mathbf{x}}_{T+1}^\top\right) \otimes \left(\widehat{\mathbf{x}}_{T+1}\widehat{\mathbf{x}}_{T+1}^\top\right)\right], \quad \widetilde{\mathcal{H}} := \widehat{\mathbf{B}}^\top \mathbf{H}\widehat{\mathbf{B}} \otimes \widehat{\mathbf{B}}^\top \mathbf{H}\widehat{\mathbf{B}},$$

$$\mathcal{G}^{(\tau)} := \widehat{\mathbf{B}}^\top \mathbf{H}\widehat{\mathbf{B}} \otimes \mathbf{I}_k + \mathbf{I}_k \otimes \widehat{\mathbf{B}}^\top \mathbf{H}\widehat{\mathbf{B}} - \eta_\tau \mathcal{H}, \quad \widetilde{\mathcal{G}}^{(\tau)} := \widehat{\mathbf{B}}^\top \mathbf{H}\widehat{\mathbf{B}} \otimes \mathbf{I}_k + \mathbf{I}_k \otimes \widehat{\mathbf{B}}^\top \mathbf{H}\widehat{\mathbf{B}} - \eta_\tau \widetilde{\mathcal{H}}.$$

We use the notation $\mathcal{O} \circ \mathbf{M}$ to denotes the operator $\mathcal{O}$ acting on a symmetric matrix $\mathbf{M}$. One can verify the following rules for these operators acting on a symmetric matrix $\mathbf{M}$,

$$\mathcal{I} \circ \mathbf{M} = \mathbf{M}, \quad \mathcal{H} \circ \mathbf{M} = \mathbb{E}\left[\left(\widehat{\mathbf{x}}_{T+1}^\top \mathbf{M}\widehat{\mathbf{x}}_{T+1}\right) \widehat{\mathbf{x}}_{T+1}\widehat{\mathbf{x}}_{T+1}^\top\right], \quad \widetilde{\mathcal{H}} \circ \mathbf{M} = \widehat{\mathbf{B}}^\top \mathbf{H}\widehat{\mathbf{B}}\mathbf{M}\widehat{\mathbf{B}}^\top \mathbf{H}\widehat{\mathbf{B}},$$

$$\left(\mathcal{I} - \eta_\tau \mathcal{G}^{(\tau)}\right) \circ \mathbf{M} = \mathbb{E}\left[\left(\mathbf{I}_k - \eta_\tau \widehat{\mathbf{x}}_{T+1}\widehat{\mathbf{x}}_{T+1}^\top\right) \mathbf{M} \left(\mathbf{I} - \eta_\tau \widehat{\mathbf{x}}_{T+1}\widehat{\mathbf{x}}_{T+1}^\top\right)\right],$$

$$\left(\mathcal{I} - \eta_\tau \widetilde{\mathcal{G}}^{(\tau)}\right) \circ \mathbf{M} = \left(\mathbf{I}_k - \eta_\tau \widehat{\mathbf{B}}^\top \mathbf{H}\widehat{\mathbf{B}}\right) \mathbf{M} \left(\mathbf{I}_k - \eta_\tau \widehat{\mathbf{B}}^\top \mathbf{H}\widehat{\mathbf{B}}\right).$$

Moreover, let $L := \log(K_2 - h)$.

To establish the convergence result for the future task, we begin by analyzing the recursive properties of the matrices $\mathbf{V}^{(\tau)}$ and $\mathbf{B}^{(\tau)}$, defined as follows:

$$\mathbf{V}^{(\tau)} := \mathbb{E}\left[\widehat{\mathbf{w}}_{T+1,\text{var}}^{(\tau)} \otimes \widehat{\mathbf{w}}_{T+1,\text{var}}^{(\tau)}\right], \qquad \mathbf{B}^{(\tau)} := \mathbb{E}\left[\widehat{\mathbf{w}}_{T+1,\text{bias}}^{(\tau)} \otimes \widehat{\mathbf{w}}_{T+1,\text{bias}}^{(\tau)}\right].$$

This approach is motivated by the inequality

$$\begin{aligned} L_\rho(\widehat{\mathbf{B}}, \mathbf{w}_{T+1}^{(K_2-1)}) - L_\rho(\mathbf{B}^*, \mathbf{w}_{T+1}^*) =& \mathbb{E}\left[\widehat{L}(\mathbf{w}_{T+1}^{(K_2-1)})\right] - \sigma^2 \\ \leq& \left\langle \widehat{\mathbf{B}}^\top \mathbf{H}\widehat{\mathbf{B}}, \mathbb{E}\left[\widehat{\mathbf{w}}_{T+1}^{(K_2-1)} \otimes \widehat{\mathbf{w}}_{T+1}^{(K_2-1)}\right]\right\rangle + \left\|\left(\widehat{\mathbf{B}}\mathbf{R} - \mathbf{F}\right)\mathbf{w}_{T+1}^*\right\|_{\mathbf{H}}^2 \\ \overset{(a)}{\leq}& 2\left(\left\langle \widehat{\mathbf{B}}^\top \mathbf{H}\widehat{\mathbf{B}}, \mathbf{V}^{(K_2-1)}\right\rangle + \left\langle \widehat{\mathbf{B}}^\top \mathbf{H}\widehat{\mathbf{B}}, \mathbf{B}^{(K_2-1)}\right\rangle\right) \\ & + \left\|\left(\widehat{\mathbf{B}}\mathbf{R} - \mathbf{F}\right)\mathbf{w}_{T+1}^*\right\|_{\mathbf{H}}^2, \end{aligned} \tag{41}$$

where (a) is derived from: for two vectors $\mathbf{v}_1$ and $\mathbf{v}_2$, $(\mathbf{v}_1 + \mathbf{v}_2) \otimes (\mathbf{v}_1 + \mathbf{v}_2) \preceq 2(\mathbf{v}_1 \otimes \mathbf{v}_1 + \mathbf{v}_2 \otimes \mathbf{v}_2)$.

### A.2.1. VARIANCE UPPER BOUND

According to the definition of $\mathbf{V}^{(\tau)}$, we have the following recursive iteration:

$$\mathbf{V}^{(\tau+1)} \preceq \left(\mathcal{I} - \eta_\tau \mathcal{G}^{(\tau)}\right) \circ \mathbf{V}^{(\tau)} \underbrace{-\eta_\tau \mathbb{E}\left[\widehat{\mathbf{w}}_{T+1,\text{var}}^{(\tau)}\right] \left(\mathbf{E}\mathbf{w}_{T+1}^*\right)^\top \mathbf{H}\widehat{\mathbf{B}}}_{\mathcal{I}_{\mathbf{V}^{(\tau+1)}}} \underbrace{-\eta_\tau \widehat{\mathbf{B}}^\top \mathbf{H}\mathbf{E}\mathbf{w}_{T+1}^* \left(\mathbb{E}\left[\widehat{\mathbf{w}}_{T+1,\text{var}}^{(\tau)}\right]\right)^\top}_{\mathcal{I}_{\mathbf{V}^{(\tau+1)}}^\top}$$

$$+ \eta_\tau^2 \mathbb{E}\left[ \zeta_{T+1}^{(\tau+1)} \widehat{\mathbf{x}}_{T+1}^{(\tau+1)} \left(\widehat{\mathbf{x}}_{T+1}^{(\tau+1)}\right)^\top \widehat{\mathbf{w}}_{T+1,\text{var}}^{(\tau)} \left(\widehat{\mathbf{x}}_{T+1}^{(\tau+1)}\right)^\top \right]$$

$$\underbrace{\qquad\qquad\qquad\qquad\qquad\qquad\qquad\qquad\qquad\qquad}_{\mathcal{II}_{\mathbf{V}^{(\tau+1)}}}$$

$$+ \eta_\tau^2 \mathbb{E}\left[ \zeta_{T+1}^{(\tau+1)} \widehat{\mathbf{x}}_{T+1}^{(\tau+1)} \left(\widehat{\mathbf{w}}_{T+1,\text{var}}^{(\tau)}\right)^\top \widehat{\mathbf{x}}_{T+1}^{(\tau+1)} \left(\widehat{\mathbf{x}}_{T+1}^{(\tau+1)}\right)^\top \right]$$

$$\underbrace{\qquad\qquad\qquad\qquad\qquad\qquad\qquad\qquad\qquad\qquad}_{\mathcal{II}_{\mathbf{V}^{(\tau+1)}}^\top}$$

$$+ \eta_\tau^2 \left( \alpha \left\| \mathbf{E}\mathbf{w}_{T+1}^* \right\|_{\mathbf{H}}^2 + \sigma^2 \right) \widehat{\mathbf{B}}^\top \mathbf{H}\widehat{\mathbf{B}}. \tag{42}$$

The inequality above is derived from that

$$\mathbb{E}\left[ \left(\zeta_{T+1}^{(\tau+1)}\right)^2 \widehat{\mathbf{x}}_{T+1}^{(\tau+1)} \left(\widehat{\mathbf{x}}_{T+1}^{(\tau+1)}\right)^\top \right] \overset{(a)}{\preceq} \left( \alpha \left\| \mathbf{E}\mathbf{w}_{T+1}^* \right\|_{\mathbf{H}}^2 + \sigma^2 \right) \widehat{\mathbf{B}}^\top \mathbf{H}\widehat{\mathbf{B}},$$

where (a) follows from Assumption 3.3.

**Lemma A.6.** *Suppose Assumption 3.3 holds. Then, for any $\tau + 1 \in [K_2]$, the matrix $\mathbf{V}^{(\tau+1)}$ admits the following upper bounds:*

$$\mathbf{V}^{(\tau+1)} \preceq \left( \mathcal{I} - \eta_\tau \mathcal{G}^{(\tau)} \right) \circ \mathbf{V}^{(\tau)} + \eta_\tau L^2 \left\| \left( \mathbf{H}^{1/2}\widehat{\mathbf{B}} \right)^\dagger \mathbf{H}^{1/2} \mathbf{E}\mathbf{w}_{T+1}^* \right\|^2 \cdot \mathbf{I}_k$$

$$+ \eta_\tau \left\| \widehat{\mathbf{B}}^\top \mathbf{H}\widehat{\mathbf{B}} \right\|_{\text{op}} \left\| \mathbf{H} \right\|_{\text{op}} \left\| \mathbf{E}\mathbf{w}_{T+1}^* \right\|^2 \cdot \mathbf{I}_k$$

$$+ \alpha L^2 \eta_\tau^2 \left\| \widehat{\mathbf{B}}^\top \mathbf{H}\widehat{\mathbf{B}} \right\|_{\text{op}} \left\| \left( \mathbf{H}^{1/2}\widehat{\mathbf{B}} \right)^\dagger \mathbf{H}^{1/2} \mathbf{E}\mathbf{w}_{T+1}^* \right\|^2 \cdot \widehat{\mathbf{B}}^\top \mathbf{H}\widehat{\mathbf{B}}$$

$$+ \eta_\tau^2 \left( 2\alpha \left\| \mathbf{H} \right\|_{\text{op}} \left\| \mathbf{E}\mathbf{w}_{T+1}^* \right\|^2 + \sigma^2 \right) \widehat{\mathbf{B}}^\top \mathbf{H}\widehat{\mathbf{B}}, \tag{43}$$

*and*

$$\mathbf{V}^{(\tau+1)} \preceq \left( \mathcal{I} - \eta_\tau \mathcal{G}^{(\tau)} \right) \circ \mathbf{V}^{(\tau)} + \eta_\tau \left( \mathcal{III}^{(\tau)} + \mathcal{IV} \right)$$

$$+ \alpha L^2 \eta_\tau^2 \left\| \widehat{\mathbf{B}}^\top \mathbf{H}\widehat{\mathbf{B}} \right\|_{\text{op}} \left\| \left( \mathbf{H}^{1/2}\widehat{\mathbf{B}} \right)^\dagger \mathbf{H}^{1/2} \mathbf{E}\mathbf{w}_{T+1}^* \right\|^2 \cdot \widehat{\mathbf{B}}^\top \mathbf{H}\widehat{\mathbf{B}}$$

$$+ \eta_\tau^2 \left( 2\alpha \left\| \mathbf{H} \right\|_{\text{op}} \left\| \mathbf{E}\mathbf{w}_{T+1}^* \right\|^2 + \sigma^2 \right) \widehat{\mathbf{B}}^\top \mathbf{H}\widehat{\mathbf{B}}, \tag{44}$$

*where* $\mathcal{III}^{(\tau)} := \left\{ \left[ \sum_{j=0}^\tau \eta_j \prod_{i=j+1}^\tau \left( \mathbf{I}_k - \eta_i \widehat{\mathbf{B}}^\top \mathbf{H}\widehat{\mathbf{B}} \right) \right] \widehat{\mathbf{B}}^\top \mathbf{H}\widehat{\mathbf{B}} \left( \mathbf{H}^{1/2}\widehat{\mathbf{B}} \right)^\dagger \mathbf{H}^{1/2} \mathbf{E}\mathbf{w}_{T+1}^* \right\}^{\otimes 2}$, *and* $\mathcal{IV} := \left( \widehat{\mathbf{B}}^\top \mathbf{H}\mathbf{E}\mathbf{w}_{T+1}^* \right)^{\otimes 2}.$

*Proof.* According to Eq. (39), the following recursive equation holds for $\mathbb{E}\left[ \widehat{\mathbf{w}}_{T+1,\text{var}}^{(\tau)} \right]$,

$$\mathbb{E}\left[ \widehat{\mathbf{w}}_{T+1,\text{var}}^{(\tau)} \right] = \left( \mathbf{I}_k - \eta_{\tau-1} \widehat{\mathbf{B}}^\top \mathbf{H}\widehat{\mathbf{B}} \right) \mathbb{E}\left[ \widehat{\mathbf{w}}_{T+1,\text{var}}^{(\tau-1)} \right] - \eta_{\tau-1} \widehat{\mathbf{B}}^\top \mathbf{H}\mathbf{E}\mathbf{w}_{T+1}^*$$

$$= \left[ \sum_{j=0}^\tau \eta_j \prod_{i=j+1}^\tau \left( \mathbf{I}_k - \eta_i \widehat{\mathbf{B}}^\top \mathbf{H}\widehat{\mathbf{B}} \right) \right] \widehat{\mathbf{B}}^\top \mathbf{H}\mathbf{E}\mathbf{w}_{T+1}^*$$

$$= \left[ \sum_{j=0}^\tau \eta_j \prod_{i=j+1}^\tau \left( \mathbf{I}_k - \eta_i \widehat{\mathbf{B}}^\top \mathbf{H}\widehat{\mathbf{B}} \right) \right] \widehat{\mathbf{B}}^\top \mathbf{H}\widehat{\mathbf{B}} \left( \mathbf{H}^{1/2}\widehat{\mathbf{B}} \right)^\dagger \mathbf{H}^{1/2} \mathbf{E}\mathbf{w}_{T+1}^*. \tag{45}$$

Applying Eq. (45) to the term $\mathcal{I}_{\mathbf{V}^{(\tau+1)}}$ of Eq. (42), we can obtain that

$$\mathcal{I}_{\mathbf{V}^{(\tau+1)}} \preceq \frac{\eta_\tau}{2} \left\{ \left[ \sum_{j=0}^\tau \eta_j \prod_{i=j+1}^\tau \left( \mathbf{I}_k - \eta_i \widehat{\mathbf{B}}^\top \mathbf{H}\widehat{\mathbf{B}} \right) \right] \widehat{\mathbf{B}}^\top \mathbf{H}\widehat{\mathbf{B}} \left( \mathbf{H}^{1/2}\widehat{\mathbf{B}} \right)^\dagger \mathbf{H}^{1/2} \mathbf{E}\mathbf{w}_{T+1}^* \right\}^{\otimes 2}$$

$$+ \frac{\eta_\tau}{2} \left( \widehat{\mathbf{B}}^\top \mathbf{H} \mathbf{E} \mathbf{w}_{T+1}^* \right)^{\otimes 2}$$

$$\preceq \frac{\eta_\tau}{2} \left\| \left( \mathbf{H}^{1/2} \widehat{\mathbf{B}} \right)^\dagger \mathbf{H}^{1/2} \mathbf{E} \mathbf{w}_{T+1}^* \right\|^2 \cdot \left[ \sum_{j=0}^\tau \eta_j \prod_{i=j+1}^\tau \left( \mathbf{I}_k - \eta_i \widehat{\mathbf{B}}^\top \mathbf{H} \widehat{\mathbf{B}} \right) \right]^2 \left( \widehat{\mathbf{B}}^\top \mathbf{H} \widehat{\mathbf{B}} \right)^2$$

$$+ \frac{\eta_\tau}{2} \left\| \widehat{\mathbf{B}}^\top \mathbf{H} \mathbf{E} \mathbf{w}_{T+1}^* \right\|^2 \cdot \mathbf{I}_k.$$

In order to estimate term $\left[ \sum_{j=0}^\tau \eta_j \prod_{i=j+1}^\tau \left( \mathbf{I}_k - \eta_i \widehat{\mathbf{B}}^\top \mathbf{H} \widehat{\mathbf{B}} \right) \right] \widehat{\mathbf{B}}^\top \mathbf{H} \widehat{\mathbf{B}}$, we introduce the following auxiliary function $f_\tau : \mathbb{R}_+ \to \mathbb{R}$ for any $\tau \in [K_2 - 1] \bigcup 0$,

$$f_\tau(x) := \left[ \sum_{j=0}^\tau \eta_j \prod_{i=j+1}^\tau (1 - \eta_i x) \right] x, \qquad \forall x \in \mathbb{R}_+.$$

One can notice that

$$f_\tau(x) \leq f_{K_2 - 1}(x)$$

$$= \eta_0 \sum_{j=0}^{K_2' + h - 1} (1 - \eta_0 x)^{K_2' + h - 1 - j} \prod_{i=1}^{L-1} \left( 1 - \frac{\eta_0}{2^i} x \right)^{K_2'} x$$

$$+ \sum_{l=1}^{L-1} \frac{\eta_0}{2^l} \sum_{j=1}^{K_2'} \left( 1 - \frac{\eta_0}{2^l} x \right)^{K_2' - j} \prod_{i=l+1}^{L-1} \left( 1 - \frac{\eta_0}{2^i} x \right)^{K_2'} x$$

$$= \left( 1 - (1 - \eta_0 x)^{K_2' + h} \right) \prod_{i=1}^{L-1} \left( 1 - \frac{\eta_0}{2^i} x \right)^{K_2'}$$

$$+ \sum_{l=1}^{L-1} \left( 1 - \left( 1 - \frac{\eta_0}{2^l} x \right)^{K_2'} \right) \prod_{i=l+1}^{L-1} \left( 1 - \frac{\eta_0}{2^i} x \right)^{K_2'}$$

$$\leq L, \tag{46}$$

for all $x$ satisfying $\eta_0 x < 1$. Therefore, the following upper bounds hold for $\mathcal{I}_{\mathbf{V}^{(\tau+1)}}$,

$$\mathcal{I}_{\mathbf{V}^{(\tau+1)}} \preceq \frac{\eta_\tau L^2}{2} \left\| \left( \mathbf{H}^{1/2} \widehat{\mathbf{B}} \right)^\dagger \mathbf{H}^{1/2} \mathbf{E} \mathbf{w}_{T+1}^* \right\|^2 \cdot \mathbf{I}_k + \frac{\eta_\tau}{2} \left\| \widehat{\mathbf{B}}^\top \mathbf{H} \mathbf{E} \mathbf{w}_{T+1}^* \right\|^2 \cdot \mathbf{I}_k, \tag{47}$$

and

$$\mathcal{I}_{\mathbf{V}^{(\tau+1)}} \preceq \frac{\eta_\tau}{2} \mathcal{III}^{(\tau)} + \frac{\eta_\tau}{2} \left( \widehat{\mathbf{B}}^\top \mathbf{H} \mathbf{E} \mathbf{w}_{T+1}^* \right)^{\otimes 2}. \tag{48}$$

Furthermore, by applying Eq. (45) to the term $\mathcal{II}_{\mathbf{V}^{(\tau+1)}}$ in Eq. (42), we obtain

$$\mathcal{II}_{\mathbf{V}^{(\tau+1)}} = \eta_\tau^2 \mathbb{E} \left[ \widehat{\mathbf{x}}_{T+1}^{(\tau+1)} \left( \widehat{\mathbf{x}}_{T+1}^{(\tau+1)} \right)^\top \mathbb{E} \left[ \widehat{\mathbf{w}}_{T+1,\mathrm{var}}^{(\tau)} \right] \left( \mathbf{E} \mathbf{w}_{T+1}^* \right)^\top \mathbf{x}_{T+1}^{(\tau+1)} \left( \widehat{\mathbf{x}}_{T+1}^{(\tau+1)} \right)^\top \right]$$

$$\overset{(a)}{\preceq} \frac{\alpha \eta_\tau^2}{2} \left\| \mathbb{E} \left[ \widehat{\mathbf{w}}_{T+1,\mathrm{var}}^{(\tau)} \right] \right\|_{\widehat{\mathbf{B}}^\top \mathbf{H} \widehat{\mathbf{B}}}^2 \cdot \widehat{\mathbf{B}}^\top \mathbf{H} \widehat{\mathbf{B}} + \frac{\alpha \eta_\tau^2}{2} \left\| \mathbf{E} \mathbf{w}_{T+1}^* \right\|_{\mathbf{H}}^2 \cdot \widehat{\mathbf{B}}^\top \mathbf{H} \widehat{\mathbf{B}}$$

$$\overset{(b)}{\preceq} \frac{\alpha L^2 \eta_\tau^2}{2} \left\| \left( \mathbf{H}^{1/2} \widehat{\mathbf{B}} \right)^\dagger \mathbf{H}^{1/2} \mathbf{E} \mathbf{w}_{T+1}^* \right\|^2 \left\| \widehat{\mathbf{B}}^\top \mathbf{H} \widehat{\mathbf{B}} \right\|_{\mathrm{op}} \cdot \widehat{\mathbf{B}}^\top \mathbf{H} \widehat{\mathbf{B}}$$

$$+ \frac{\alpha \eta_\tau^2}{2} \left\| \mathbf{E} \mathbf{w}_{T+1}^* \right\|^2 \left\| \mathbf{H} \right\|_{\mathrm{op}} \cdot \widehat{\mathbf{B}}^\top \mathbf{H} \widehat{\mathbf{B}}, \tag{49}$$

where (a) follows from [**A$_2$**] of Assumption 3.3, and (b) combines Eq. (45) with the upper bound on $f$ given in Eq. (46). Substituting Eqs. (47), and (48), and (49) into Eq. (42) completes the proof. $\qquad \square$

Next, provide an uniform upper bound for $\mathbf{V}^{(\tau+1)}$ over $[K_2]$

**Lemma A.7.** *Suppose Assumption 3.3 holds and*

$$\left\|\mathbf{E}\mathbf{w}_{T+1}^*\right\|^2 \leq \frac{\sigma^2 \min\left\{\eta_0 \sigma_{\min}^2\left(\mathbf{B}^*\right)\sigma_{\min}(\mathbf{H}), 1\right\}}{8(1+\alpha)(1+\beta)\|\mathbf{H}\|_{\mathrm{op}}\left[L^2\left\|\left(\mathbf{H}^{1/2}\mathbf{B}^*\right)^\dagger\right\|_{\mathrm{op}}^2 + L^2 + 1\right]}, \tag{50}$$

$$\left\|\left(\mathbf{H}^{1/2}\mathbf{B}^*\mathbf{R}^\top\right)^\dagger - \left(\mathbf{H}^{1/2}\widehat{\mathbf{B}}\right)^\dagger\right\| \leq \mathcal{O}(1), \quad \left\|\widehat{\mathbf{B}}\mathbf{R} - \mathbf{B}^*\right\| \leq \frac{1}{2}\sigma_k\left(\mathbf{B}^*\right) \tag{51}$$

*where $\beta := \|\mathbf{B}^*\|_{\mathrm{op}}^2 \|\mathbf{H}\|_{\mathrm{op}}^2$. According to Eq. (43), by choosing $\eta_0 < 1/\left(2\alpha\operatorname{tr}\left(\widehat{\mathbf{B}}^\top\mathbf{H}\widehat{\mathbf{B}}\right)\right)$. Then for every $\tau \in [K_2 - 1]\bigcup 0$, we have*

$$\mathbf{V}^{(\tau)} \preceq \frac{\eta_0\sigma^2}{1 - \eta_0\alpha\operatorname{tr}\left(\widehat{\mathbf{B}}^\top\mathbf{H}\widehat{\mathbf{B}}\right)}\mathbf{I}_k. \tag{52}$$

*Proof.* Proceed with the proof by mathematical induction. For $t = 0$, we already have $\mathbf{V}^{(0)} = \mathbf{0}$. Now, assume $\mathbf{V}^{(\tau)} \preceq \frac{\eta_0\sigma^2}{1-\eta_0\alpha\operatorname{tr}(\widehat{\mathbf{B}}^\top\mathbf{H}\widehat{\mathbf{B}})}\mathbf{I}_k$. Using the recurrence relation of $\mathbf{V}^{(\tau)}$ given in Eq. (43), we show that Eq. (52) holds:

$$\mathbf{V}^{(\tau+1)} \preceq \left(\mathcal{I} - \eta_\tau\mathcal{G}^{(\tau)}\right)\circ\mathbf{V}^{(\tau)} + \eta_\tau\left(L^2\left\|\left(\mathbf{H}^{1/2}\widehat{\mathbf{B}}\right)^\dagger\mathbf{H}^{1/2}\right\|_{\mathrm{op}}^2 + \left\|\widehat{\mathbf{B}}^\top\mathbf{H}\widehat{\mathbf{B}}\right\|_{\mathrm{op}}\|\mathbf{H}\|_{\mathrm{op}}\right)\left\|\mathbf{E}\mathbf{w}_{T+1}^*\right\|^2\cdot\mathbf{I}_k$$

$$+ \alpha\eta_\tau^2\left(L^2\left\|\widehat{\mathbf{B}}^\top\mathbf{H}\widehat{\mathbf{B}}\right\|_{\mathrm{op}}\left\|\left(\mathbf{H}^{1/2}\widehat{\mathbf{B}}\right)^\dagger\mathbf{H}^{1/2}\right\|_{\mathrm{op}}^2 + 2\|\mathbf{H}\|_{\mathrm{op}}\right)\left\|\mathbf{E}\mathbf{w}_{T+1}^*\right\|^2\cdot\widehat{\mathbf{B}}^\top\mathbf{H}\widehat{\mathbf{B}}$$

$$+ \eta_\tau^2\sigma^2\widehat{\mathbf{B}}^\top\mathbf{H}\widehat{\mathbf{B}}$$

$$\overset{(a)}{\preceq}\left(\mathcal{I} - \eta_\tau\widehat{\mathbf{B}}^\top\mathbf{H}\widehat{\mathbf{B}}\otimes\mathbf{I}_k - \eta_\tau\mathbf{I}_k\otimes\widehat{\mathbf{B}}^\top\mathbf{H}\widehat{\mathbf{B}}\right)\circ\mathbf{V}^{(\tau)} + \eta_\tau^2\mathcal{H}\circ\mathbf{V}^{(\tau)}$$

$$+ \eta_\tau\eta_0\cdot\frac{\sigma^2}{1-\eta_0\alpha\operatorname{tr}\left(\widehat{\mathbf{B}}^\top\mathbf{H}\widehat{\mathbf{B}}\right)}\widehat{\mathbf{B}}^\top\mathbf{H}\widehat{\mathbf{B}} + \eta_\tau^2\sigma^2\widehat{\mathbf{B}}^\top\mathbf{H}\widehat{\mathbf{B}}$$

$$\preceq\frac{\eta_0\sigma^2}{1-\eta_0\alpha\operatorname{tr}\left(\widehat{\mathbf{B}}^\top\mathbf{H}\widehat{\mathbf{B}}\right)}\left(\mathbf{I}_k - 2\eta_\tau\widehat{\mathbf{B}}^\top\mathbf{H}\widehat{\mathbf{B}}\right) + \frac{\eta_\tau^2\eta_0\sigma^2}{1-\eta_0\alpha\operatorname{tr}\left(\widehat{\mathbf{B}}^\top\mathbf{H}\widehat{\mathbf{B}}\right)}\cdot\alpha\operatorname{tr}\left(\widehat{\mathbf{B}}^\top\mathbf{H}\widehat{\mathbf{B}}\right)\widehat{\mathbf{B}}^\top\mathbf{H}\widehat{\mathbf{B}}$$

$$+ \eta_\tau\eta_0\cdot\frac{\sigma^2}{1-\eta_0\alpha\operatorname{tr}\left(\widehat{\mathbf{B}}^\top\mathbf{H}\widehat{\mathbf{B}}\right)}\widehat{\mathbf{B}}^\top\mathbf{H}\widehat{\mathbf{B}} + \eta_\tau^2\sigma^2\widehat{\mathbf{B}}^\top\mathbf{H}\widehat{\mathbf{B}}$$

$$=\frac{\eta_0\sigma^2}{1-\eta_0\alpha\operatorname{tr}\left(\widehat{\mathbf{B}}^\top\mathbf{H}\widehat{\mathbf{B}}\right)}\mathbf{I}_k - \left(\eta_\tau\eta_0 - \eta_\tau^2\right)\cdot\frac{\sigma^2}{1-\eta_0\alpha\operatorname{tr}\left(\widehat{\mathbf{B}}^\top\mathbf{H}\widehat{\mathbf{B}}\right)}\widehat{\mathbf{B}}^\top\mathbf{H}\widehat{\mathbf{B}}$$

$$\preceq\frac{\eta_0\sigma^2}{1-\eta_0\alpha\operatorname{tr}\left(\widehat{\mathbf{B}}^\top\mathbf{H}\widehat{\mathbf{B}}\right)}\mathbf{I}_k,$$

where (a) is derived from combining the assumption (50) with $\eta_0 < 1/\left(2\alpha\operatorname{tr}\left(\widehat{\mathbf{B}}^\top\mathbf{H}\widehat{\mathbf{B}}\right)\right)$. This completes the induction. $\square$

**Theorem A.8.** *Assuming that Assumption 3.3 holds and that Eqs. (50) and (51) are satisfied, we further recall the definition $K_2' = (K_2 - h)/\log(K_2 - h)$. Under the conditions $h \geq 0$, $K_2' \geq 1$, and $\eta_0 < 1/\left(2\alpha\operatorname{tr}\left(\widehat{\mathbf{B}}^\top\mathbf{H}\widehat{\mathbf{B}}\right)\right)$, we obtain the following result,*

$$\left\langle\widehat{\mathbf{B}}^\top\mathbf{H}\widehat{\mathbf{B}}, \mathbf{V}^{(K_2-1)}\right\rangle \leq \log^3(K_2)\left[\left(\left\|\left(\mathbf{H}^{1/2}\mathbf{B}^*\right)^\dagger\right\|_{\mathrm{op}}^2 + 1\right)\|\mathbf{H}\|_{\mathrm{op}} + 2\|\mathbf{B}^*\|_{\mathrm{op}}^2\|\mathbf{H}\|_{\mathrm{op}}^2\right]\left\|\mathbf{E}\mathbf{w}_{T+1}^*\right\|^2$$

$$+ \frac{16\sigma^2 \log(K_2)k}{K_2}. \tag{53}$$

*Proof.* Based on Eq. (44), we obtain

$$
\begin{aligned}
\mathbf{V}^{(\tau+1)} &\overset{(a)}{\preceq} \left(\mathcal{I} - \eta_\tau \widetilde{\mathcal{G}}^{(\tau)}\right) \circ \mathbf{V}^{(\tau)} + \eta_\tau \left(\mathcal{III}^{(\tau)} + \mathcal{IV}\right) + \eta_\tau^2 \mathcal{H} \circ \mathbf{V}^{(\tau)} + \eta_\tau^2 \sigma^2 \widehat{\mathbf{B}}^\top \mathbf{H} \widehat{\mathbf{B}} \\
&\qquad + \frac{\eta_\tau^2 \sigma^2}{1 - \eta_0 \alpha \operatorname{tr}\left(\widehat{\mathbf{B}}^\top \mathbf{H} \widehat{\mathbf{B}}\right)} \widehat{\mathbf{B}}^\top \mathbf{H} \widehat{\mathbf{B}} \\
&\overset{(b)}{\preceq} \left(\mathcal{I} - \eta_\tau \widetilde{\mathcal{G}}^{(\tau)}\right) \circ \mathbf{V}^{(\tau)} + \eta_\tau \left(\mathcal{III}^{(\tau)} + \mathcal{IV}\right) + \frac{\eta_\tau^2 \eta_0 \sigma^2}{1 - \eta_0 \alpha \operatorname{tr}\left(\widehat{\mathbf{B}}^\top \mathbf{H} \widehat{\mathbf{B}}\right)} \cdot \alpha \operatorname{tr}\left(\widehat{\mathbf{B}}^\top \mathbf{H} \widehat{\mathbf{B}}\right) \widehat{\mathbf{B}}^\top \mathbf{H} \widehat{\mathbf{B}} \\
&\qquad + \eta_\tau^2 \sigma^2 \widehat{\mathbf{B}}^\top \mathbf{H} \widehat{\mathbf{B}} \\
&= \left(\mathcal{I} - \eta_\tau \widetilde{\mathcal{G}}^{(\tau)}\right) \circ \mathbf{V}^{(\tau)} + \eta_\tau \left(\mathcal{III}^{(\tau)} + \mathcal{IV}\right) + \frac{\eta_\tau^2 \sigma^2}{1 - \eta_0 \alpha \operatorname{tr}\left(\widehat{\mathbf{B}}^\top \mathbf{H} \widehat{\mathbf{B}}\right)} \widehat{\mathbf{B}}^\top \mathbf{H} \widehat{\mathbf{B}},
\end{aligned}
$$

where (a) follows from the assumption for $\left\|\mathbf{E}\mathbf{w}_{T+1}^*\right\|^2$ and (b) is obtained from Lemma A.7. Solving the recursion yields

$$
\begin{aligned}
\mathbf{V}^{(K_2-1)} &\preceq \sum_{j=0}^{K_2-1} \eta_j \prod_{i=j+1}^{K_2-1} \left(\mathcal{I} - \eta_i \widetilde{\mathcal{G}}^{(i)}\right) \circ \left(\mathcal{III}^{(j)} + \mathcal{IV}\right) \\
&\qquad + \frac{\sigma^2}{1 - \eta_0 \alpha \operatorname{tr}\left(\widehat{\mathbf{B}}^\top \mathbf{H} \widehat{\mathbf{B}}\right)} \sum_{j=0}^{K_2-1} \eta_j^2 \prod_{i=j+1}^{K_2-1} \left(\mathcal{I} - \eta_i \widetilde{\mathcal{G}}^{(i)}\right) \circ \widehat{\mathbf{B}}^\top \mathbf{H} \widehat{\mathbf{B}} \\
&= \sum_{j=0}^{K_2-1} \eta_j \left[\prod_{i=j+1}^{K_2-1} \left(\mathbf{I}_k - \eta_i \widehat{\mathbf{B}}^\top \mathbf{H} \widehat{\mathbf{B}}\right)\right] \mathcal{III}^{(j)} \left[\prod_{i=j+1}^{K_2-1} \left(\mathbf{I}_k - \eta_i \widehat{\mathbf{B}}^\top \mathbf{H} \widehat{\mathbf{B}}\right)\right] \\
&\qquad + \sum_{j=0}^{K_2-1} \eta_j \left[\prod_{i=j+1}^{K_2-1} \left(\mathbf{I}_k - \eta_i \widehat{\mathbf{B}}^\top \mathbf{H} \widehat{\mathbf{B}}\right)\right] \mathcal{IV} \left[\prod_{i=j+1}^{K_2-1} \left(\mathbf{I}_k - \eta_i \widehat{\mathbf{B}}^\top \mathbf{H} \widehat{\mathbf{B}}\right)\right] \\
&\qquad + \frac{\sigma^2}{1 - \eta_0 \alpha \operatorname{tr}\left(\widehat{\mathbf{B}}^\top \mathbf{H} \widehat{\mathbf{B}}\right)} \sum_{j=0}^{K_2-1} \eta_j^2 \prod_{i=j+1}^{K_2-1} \left(\mathbf{I}_k - \eta_i \widehat{\mathbf{B}}^\top \mathbf{H} \widehat{\mathbf{B}}\right)^2 \widehat{\mathbf{B}}^\top \mathbf{H} \widehat{\mathbf{B}} \\
&\preceq \underbrace{\sum_{j=0}^{K_2-1} \eta_j \left[\prod_{i=j+1}^{K_2-1} \left(\mathbf{I}_k - \eta_i \widehat{\mathbf{B}}^\top \mathbf{H} \widehat{\mathbf{B}}\right)\right] \mathcal{III}^{(j)} \left[\prod_{i=j+1}^{K_2-1} \left(\mathbf{I}_k - \eta_i \widehat{\mathbf{B}}^\top \mathbf{H} \widehat{\mathbf{B}}\right)\right]}_{\mathcal{V}} \\
&\qquad + \underbrace{\sum_{j=0}^{K_2-1} \eta_j \left[\prod_{i=j+1}^{K_2-1} \left(\mathbf{I}_k - \eta_i \widehat{\mathbf{B}}^\top \mathbf{H} \widehat{\mathbf{B}}\right)\right] \mathcal{IV} \left[\prod_{i=j+1}^{K_2-1} \left(\mathbf{I}_k - \eta_i \widehat{\mathbf{B}}^\top \mathbf{H} \widehat{\mathbf{B}}\right)\right]}_{\mathcal{VI}} \\
&\qquad + \frac{\sigma^2}{1 - \eta_0 \alpha \operatorname{tr}\left(\widehat{\mathbf{B}}^\top \mathbf{H} \widehat{\mathbf{B}}\right)} \underbrace{\sum_{j=0}^{K_2-1} \eta_j^2 \prod_{i=j+1}^{K_2-1} \left(\mathbf{I}_k - \eta_i \widehat{\mathbf{B}}^\top \mathbf{H} \widehat{\mathbf{B}}\right) \widehat{\mathbf{B}}^\top \mathbf{H} \widehat{\mathbf{B}}}_{\mathcal{VII}}. \tag{54}
\end{aligned}
$$

The overall goal is to estimate $\left\langle \widehat{\mathbf{B}}^\top \mathbf{H} \widehat{\mathbf{B}}, \mathbf{V}^{(K_2-1)} \right\rangle$. As an intermediate step, we first estimate $\left\langle \widehat{\mathbf{B}}^\top \mathbf{H} \widehat{\mathbf{B}}, \mathcal{V} \right\rangle$ and

$\left\langle \widehat{\mathbf{B}}^\top \mathbf{H} \widehat{\mathbf{B}}, \mathcal{VI} \right\rangle$. To proceed, consider the SVD of $\widehat{\mathbf{B}}^\top \mathbf{H} \widehat{\mathbf{B}}$, which is $\widehat{\mathbf{B}}^\top \mathbf{H} \widehat{\mathbf{B}} = \widehat{\mathbf{U}}^\top \widehat{\boldsymbol{\Sigma}} \widehat{\mathbf{U}}$. We then obtain

$$\left\langle \widehat{\mathbf{B}}^\top \mathbf{H} \widehat{\mathbf{B}}, \mathcal{V} \right\rangle = \left\langle \widehat{\boldsymbol{\Sigma}}, \sum_{j=0}^{K_2-1} \eta_j \left\{ \left[ \prod_{i=j+1}^{K_2-1} \left( \mathbf{I}_k - \eta_i \widehat{\boldsymbol{\Sigma}} \right) \right] \left[ \sum_{l=0}^{j} \eta_l \prod_{m=l+1}^{j} \left( \mathbf{I}_k - \eta_m \widehat{\boldsymbol{\Sigma}} \right) \right] \widehat{\boldsymbol{\Sigma}} \widehat{\mathbf{U}} \widehat{\mathbf{w}}_{T+1}^* \right\}^{\otimes 2} \right\rangle, \quad (55)$$

where $\widehat{\mathbf{w}}_{T+1}^* := \left( \mathbf{H}^{1/2} \widehat{\mathbf{B}} \right)^\dagger \mathbf{H}^{1/2} \mathbf{E} \mathbf{w}_{T+1}^*$. Observing that the matrix $\widehat{\mathbf{U}} \widehat{\mathbf{w}}_{T+1}^* \left( \widehat{\mathbf{w}}_{T+1}^* \right)^\top \widehat{\mathbf{U}}^\top$ is rank-1, we can estimate Eq. (55) element-wise along the diagonal from 1 to $k$. Combining the auxiliary function $f_\tau$ from Lemma A.6 with its upper bound in Eq. (46), we obtain:

$$\left\langle \widehat{\mathbf{B}}^\top \mathbf{H} \widehat{\mathbf{B}}, \mathcal{V} \right\rangle \le L^3 \left\| \left( \mathbf{H}^{1/2} \widehat{\mathbf{B}} \right)^\dagger \mathbf{H}^{1/2} \mathbf{E} \mathbf{w}_{T+1}^* \right\|^2. \quad (56)$$

Similarly, we also have

$$\left\langle \widehat{\mathbf{B}}^\top \mathbf{H} \widehat{\mathbf{B}}, \mathcal{VI} \right\rangle = \left\langle \widehat{\boldsymbol{\Sigma}}, \sum_{j=0}^{K_2-1} \eta_j \left[ \prod_{i=j+1}^{K_2-1} \left( \mathbf{I}_k - \eta_i \widehat{\boldsymbol{\Sigma}} \right) \right] \widehat{\mathbf{U}} \, \mathcal{I} \mathcal{V} \, \widehat{\mathbf{U}}^\top \left[ \prod_{i=j+1}^{K_2-1} \left( \mathbf{I}_k - \eta_i \widehat{\boldsymbol{\Sigma}} \right) \right] \right\rangle$$

$$\le L^3 \left\| \widehat{\mathbf{B}}^\top \mathbf{H} \mathbf{E} \mathbf{w}_{T+1}^* \right\|^2. \quad (57)$$

Finally, to estimate $\left\langle \widehat{\mathbf{B}}^\top \mathbf{H} \widehat{\mathbf{B}}, \mathcal{VII} \right\rangle$, we introduce an auxiliary function $g : \mathbb{R}_+ \to \mathbb{R}$ defined as

$$g(x) := x \cdot \left( 1 - (1-x)^{K_2'+h} \right) \cdot \prod_{i=1}^{L-1} \left( 1 - \frac{x}{2^i} \right)^{K_2'} + \sum_{l=1}^{L-1} \frac{x}{2^l} \cdot \left( 1 - \left( 1 - \frac{x}{2^l} \right)^{K_2'} \right) \cdot \prod_{i=l+1}^{L-1} \left( 1 - \frac{x}{2^i} \right)^{K_2'}.$$

One can notice that

$$\mathcal{VII} = \eta_0^2 \sum_{j=0}^{K_2'+h-1} \left( \mathbf{I}_k - \eta_0 \widehat{\mathbf{B}}^\top \mathbf{H} \widehat{\mathbf{B}} \right) \prod_{i=1}^{L-1} \left( \mathbf{I}_k - \frac{\eta_0}{2^i} \widehat{\mathbf{B}}^\top \mathbf{H} \widehat{\mathbf{B}} \right)^{K_2'} \widehat{\mathbf{B}}^\top \mathbf{H} \widehat{\mathbf{B}}$$

$$+ \sum_{l=1}^{L-1} \left( \frac{\eta_0}{2^l} \right)^2 \sum_{j=1}^{K_2'} \left( \mathbf{I}_k - \frac{\eta_0}{2^l} \widehat{\mathbf{B}}^\top \mathbf{H} \widehat{\mathbf{B}} \right)^{K_2'-j} \prod_{i=l+1}^{L-1} \left( \mathbf{I}_k - \frac{\eta_0}{2^i} \widehat{\mathbf{B}}^\top \mathbf{H} \widehat{\mathbf{B}} \right)^{K_2'} \widehat{\mathbf{B}}^\top \mathbf{H} \widehat{\mathbf{B}}$$

$$= \eta_0 \left( \mathbf{I}_k - \left( \mathbf{I}_k - \eta_0 \widehat{\mathbf{B}}^\top \mathbf{H} \widehat{\mathbf{B}} \right)^{K_2'+h} \right) \prod_{i=1}^{L-1} \left( \mathbf{I}_k - \frac{\eta_0}{2^i} \widehat{\mathbf{B}}^\top \mathbf{H} \widehat{\mathbf{B}} \right)^{K_2'}$$

$$+ \sum_{l=1}^{L-1} \frac{\eta_0}{2^l} \left( \mathbf{I}_k - \left( \mathbf{I}_k - \frac{\eta_0}{2^l} \widehat{\mathbf{B}}^\top \mathbf{H} \widehat{\mathbf{B}} \right)^{K_2'} \right) \prod_{i=l+1}^{L-1} \left( \mathbf{I}_k - \frac{\eta_0}{2^i} \widehat{\mathbf{B}}^\top \mathbf{H} \widehat{\mathbf{B}} \right)^{K_2'}. \quad (58)$$

Applying $g(\cdot)$ to $\eta_0 \widehat{\mathbf{B}}^\top \mathbf{H} \widehat{\mathbf{B}}$ in each diagonal entry, and using Lemma C.3 in Wu et al. (2022), we have

$$g \left( \eta_0 \widehat{\mathbf{B}}^\top \mathbf{H} \widehat{\mathbf{B}} \right) \preceq \frac{8}{K_2'} \mathbf{I}_k.$$

Now combining Eq. (54) with Eqs. (56), (57) and (58), we obtain

$$\left\langle \widehat{\mathbf{B}}^\top \mathbf{H} \widehat{\mathbf{B}}, \mathbf{V}^{(K_2-1)} \right\rangle \le L^3 \left( \left\| \left( \mathbf{H}^{1/2} \widehat{\mathbf{B}} \right)^\dagger \mathbf{H}^{1/2} \mathbf{E} \mathbf{w}_{T+1}^* \right\|^2 + \left\| \widehat{\mathbf{B}}^\top \mathbf{H} \mathbf{E} \mathbf{w}_{T+1}^* \right\|^2 \right)$$

$$+ \frac{\sigma^2}{1 - \eta_0 \alpha \operatorname{tr} \left( \widehat{\mathbf{B}}^\top \mathbf{H} \widehat{\mathbf{B}} \right)} \operatorname{tr} \left( g \left( \eta_0 \widehat{\mathbf{B}}^\top \mathbf{H} \widehat{\mathbf{B}} \right) \right)$$

$$\le L^3 \left[ \left( \left\| \left( \mathbf{H}^{1/2} \mathbf{B}^* \right)^\dagger \right\|_{\mathrm{op}}^2 + 1 \right) \|\mathbf{H}\|_{\mathrm{op}} + 2 \|\mathbf{B}^*\|_{\mathrm{op}}^2 \|\mathbf{H}\|_{\mathrm{op}}^2 \right] \|\mathbf{E} \mathbf{w}_{T+1}^*\|^2 + \frac{16\sigma^2 k}{K_2'}.$$

$\square$

### A.2.2. BIAS UPPER BOUND

To estimate the bias term $\left\langle \widehat{\mathbf{B}}^\top \mathbf{H} \widehat{\mathbf{B}}, \mathbf{B}^{(K_2-1)} \right\rangle$, we can directly apply the result from Lemma C.10 in Wu et al. (2022).

**Lemma A.9.** *[Lemma C.10 in Wu et al. (2022)] Suppose Assumption 3.3 holds. Assume $\eta_0 < \frac{1}{\left(3\alpha \operatorname{tr}\left(\widehat{\mathbf{B}}^\top \mathbf{H} \widehat{\mathbf{B}}\right) \log(K_2'+h)\right)}$. We have*

$$\left\langle \widehat{\mathbf{B}}^\top \mathbf{H} \widehat{\mathbf{B}}, \mathbf{B}^{(K_2-1)} \right\rangle \leq \frac{12e \log(K_2)}{\eta_0 K_2} \operatorname{tr}\left(\left(\mathbf{I}_k - \eta_0 \widehat{\mathbf{B}}^\top \mathbf{H} \widehat{\mathbf{B}}\right)^{2(K_2'+h)} \mathbf{B}^{(0)}\right)$$
$$+ \frac{108 e\alpha \log^3(K_2) k}{\eta_0 K_2^2} \cdot \left\| \mathbf{w}_{T+1}^{(0)} - \mathbf{R} \mathbf{w}_{T+1}^* \right\|^2.$$

### A.2.3. PROOF OF THEOREM 5.4

Notice that $\widehat{\mathbf{B}}$ satisfies the condition of Theorem A.8 by utilizing Theorem 5.1 directly. Then the proof is completed by combining Eq. (41) with Theorem A.8 and Lemma A.9, and the estimation for $\left\| \widehat{\mathbf{B}} \mathbf{R} - \mathbf{F} \right\|_F^2$ established in Theorem 5.1.

We further consider sequential curriculum learning, where parameters are learned sequentially across task groups. Curriculum learning manifests the principle of "starting small" (Wang et al., 2021), in which a model is initially trained on easier subsets of data before being gradually exposed to more difficult examples. This approach has demonstrated substantial improvements in training efficiency (Bengio et al., 2009; Hacohen & Weinshall, 2019; Narvekar et al., 2020). In our MTL setup, a curriculum consists of multiple tasks. By analogy, a course (e.g., advanced mathematics) contains multiple chapters, each can be treated as a task. We formulate task with lower additive noise variance as the "easier" curriculum, while task with higher additive noise variance constitutes the "harder" curriculum. Consider $D$ curricula categories, where the variance $\left(\sigma^{(j)}\right)^2 (j \in [D])$ of the additive noise is monotonically increasing, and the number of tasks in each category satisfies $T_j \geq d$. Our result (Corollary A.11) establishes that the curriculum strategy achieves a estimation error of $\widetilde{\mathcal{O}}\left(\frac{k\left(\sum_{j=1}^D \left(\sigma^{(j)}\right)^2 T_j\right)}{N\left(\sum_{j=1}^D T_j\right)}\right)$, which is substantially *lower* than the rate of $\widetilde{\mathcal{O}}\left(\frac{\left(\sigma^{(D)}\right)^2 k}{N}\right)$ obtained by using all data indiscriminately when the noise variances are widely dispersed. As a result, we provide theoretical justification for curriculum learning in the context Linear MTL.

## A.3. Curriculum Learning

The original concept of curriculum learning was formally introduced by Bengio et al. (2009), with its core principle being to "train by progressing from simpler to more complex data." This idea has since been expanded to a wide range of applications, including computer vision (CV) (Guo et al., 2018; Jiang et al., 2014), natural language processing (NLP) (Platanios et al., 2019; Tay et al., 2019), and various reinforcement learning (RL) tasks (Florensa et al., 2017; Narvekar et al., 2017; Ren et al., 2018). Theoretically, Weinshall et al. (2018) defined the difficulty of a training sample (or curriculum) as its empirical loss relative to the optimal classifier. They demonstrated that, in the context of linear regression, curriculum learning can potentially lead to significantly accelerated learning.

In this section, we consider scenarios where task noise variances diverge. We partition the tasks into D D groups based on their noise levels and consider learning each group sequentially. This resembles curriculum learning, where tasks with lower variance are treated as an "easier" curriculum. For simplicity, we assume the number of tasks is large. Specifically, we study under the following assumption:

**Assumption A.10.**

**[A₁]** For each task $t \in [T_j]$ in curriculum level $j$, the response $y_{t,j} \in \mathbb{R}$ givens the covariate $\mathbf{x}_{t,j} \in \mathbb{R}^d$ is generated as follows:

$$y_{t,j} = \left\langle \mathbf{x}_{t,j}, \mathbf{B}^* \mathbf{w}_{t,j}^* \right\rangle + z_{t,j}, \qquad z_{t,j} \sim \mathcal{N}\left(0, (\sigma^{(j)})^2\right).$$

Moreover, $\sigma^{(1)} \leq \sigma^{(2)} \leq \cdots \leq \sigma^{(D)}$.

**[A₂]** For each curriculum level $j \in [D]$, the number of tasks $T_j$ satisfies $T_j \geq d$.

We study the multi-task data $\{(\mathbf{x}_{t,j}, y_{t,j})\}_{t=1}^{T_j}$ for $j \in [D]$, generated under the **Data Assumption** introduced in Section 3. The noise levels in the data increase monotonically with the task index $j$. This progression in noise creates a sequence of curriculum with ascending difficulty. Suppose the number of tasks exceeds the data dimension in each curriculum ($T_j \geq d$ for all $j \in [D]$), the following Corollary A.11 specializes Theorem 5.1 to the setting of a $D$-level curriculum. It provides a theoretical guarantee for the convergence of Algorithm 1 in curriculum learning, revealing a concrete benefit of a curriculum-like training approach for MTL.

**Corollary A.11.** *Suppose $T_j \geq d$ for all $j \in [D]$ and Assumption A.10 holds. Consider that each curriculum $j$ provides $N$ samples $\{(\mathbf{x}_{t,j}^{(i)}, y_{t,j}^{(i)})\}_{i=1}^N$ for every task $t \in [T_j]$. Through proper initialization of Algorithm 3 at each curriculum stage, we obtain:*

**[A]** *Assume that Assumption 3.2 holds under the following conditions: for the level 1 curriculum, it holds with sample size $N$ and $\delta_1 \lesssim \min\{k^{-1/2}\kappa^{-7/2}(\mathbf{B}^*\mathbf{W}_1^*), \kappa^{-6}(\mathbf{B}^*\mathbf{W}_1^*)\}$; for each higher-level curriculum $j \in [2:D]$, it holds with sample size at least $N_j$ and $\delta_j \lesssim 1$. We select the hyper-parameters for the level 1 curriculum as*

$$\alpha_1 \lesssim \frac{\sigma_1(\mathbf{B}^*\mathbf{W}_1^*)}{k^5 \left(\max\{d+T, k\}\right)^{2 + \frac{C_1\kappa(\mathbf{B}^*\mathbf{W}_1^*)}{2}}} \cdot \left(\frac{\widetilde{\delta}}{\kappa^2(\mathbf{B}^*\mathbf{W}_1^*)}\right)^{C_1\kappa(\mathbf{B}^*\mathbf{W}_1^*)},$$

$$\eta_{1,1} \lesssim \frac{1}{\kappa^5(\mathbf{B}^*\mathbf{W}_1^*)\sigma_1(\mathbf{B}^*\mathbf{W}_1^*)}, \quad K_1 \gtrsim \frac{1}{\eta_{1,1}\sigma_k(\mathbf{B}^*\mathbf{W}_1^*)}, \quad \eta_{2,1} \lesssim \frac{1}{\sigma_1(\mathbf{B}^*\mathbf{W}_1^*)},$$

$$N \gtrsim \max \left\{ \frac{\left(\sigma^{(1)}\right)^2 \kappa^4(\mathbf{B}^*\mathbf{W}_1^*)T_1 k}{\sigma_k^2(\mathbf{B}^*\mathbf{W}_1^*)}, \frac{\left(\sigma^{(1)}\right)^2 \kappa(\mathbf{B}^*\mathbf{W}_1^*)\kappa^2(\mathbf{B}^*)T_1 k}{\sigma_k(\mathbf{B}^*\mathbf{W}_1^*)\sigma_k^2(\mathbf{B}^*)} \cdot \left\{\frac{\sigma_1^2(\mathbf{W}_j^*)}{\sigma_k(\mathbf{B}^*\mathbf{W}_j^*)}\right\}_{j=2}^D, \{N_j\}_{j=2}^D \right\},$$

*and for each higher level $j \in [2:D]$ we set*

$$\eta_{2,j} \lesssim \frac{1}{\sigma_1(\mathbf{B}^*\mathbf{W}_j^*)}, \quad K_j \gtrsim \eta_{2,j}\sigma_k(\mathbf{B}^*\mathbf{W}_j^*), \quad N_j \gtrsim \frac{\left(\sigma^{(j)}\right)^2 \kappa^2(\mathbf{B}^*)T_j k}{\sigma_k(\mathbf{B}^*\mathbf{W}_j^*)\sigma_k^2(\mathbf{B}^*)}. \tag{59}$$

*When learning parameters $\mathbf{B}^*$ and $\mathbf{W}_j^*$ jointly under an easy-to-difficult curriculum (from $j = 1$ to $D$), Algorithm 3 achieves a estimation error that satisfies:*

$$\frac{1}{\sum_{j=1}^D T_j} \sum_{j=1}^D \sum_{t=1}^{T_j} \left\|\widehat{\mathbf{v}}_{t,j} - \mathbf{v}_{t,j}^*\right\|^2 \lesssim \frac{k \left(\sum_{j=1}^D \left(\sigma^{(j)}\right)^2 T_j\right)}{N \left(\sum_{j=1}^D T_j\right)},$$

*where*

$$\begin{cases} \widehat{\mathbf{v}}_{t,j} = \widehat{\mathbf{B}}\widehat{\mathbf{w}}_{t,j}, \\ \mathbf{v}_{t,j}^* = \mathbf{B}^*\mathbf{w}_{t,j}^*, \end{cases} \quad \begin{cases} \widehat{\mathbf{W}}_j = \left(\widehat{\mathbf{w}}_{1,j}, \cdots, \widehat{\mathbf{w}}_{T_j,j}\right), \\ \mathbf{W}_j^* = \left(\mathbf{w}_{1,j}^*, \cdots, \mathbf{w}_{T_j,j}^*\right), \end{cases}$$

*for any $j \in [D]$ and $t \in [T_j]$.*

**[B]** *In contrast, when learning parameters $\mathbf{B}^*$ and $\{\mathbf{W}_j^*\}_{j=1}^D$ jointly under all data from all curricula, our algorithm achieves a population loss that satisfies:*

$$\frac{1}{\sum_{j=1}^D T_j} \sum_{j=1}^D \sum_{t=1}^{T_j} \left\|\widehat{\mathbf{v}}_{t,j} - \mathbf{v}_{t,j}^*\right\|^2 \lesssim \frac{\left(\sigma^{(D)}\right)^2 k}{N}.$$

*This rate is slower than that of curriculum learning due to $\left(\sigma^{(D)}\right)^2 \left(\sum_{j=1}^D T_j\right) \geq \sum_{j=1}^D \left(\sigma^{(j)}\right)^2 T_j$.*

*Proof.* Without loss of generality, according to the result of Theorem 5.1, we assume

$$\left\|\widehat{\mathbf{B}} - \mathbf{B}^*\right\|_{\mathrm{F}}^2 + \left\|\widehat{\mathbf{W}}_1 - \mathbf{W}_1^*\right\|_{\mathrm{F}}^2 \lesssim \frac{\mathrm{tr}(\mathbf{B}^*\mathbf{W}_1^*)T_1}{\sigma_k^2(\mathbf{B}^*\mathbf{W}_1^*)N}. \tag{60}$$

---

**Algorithm 3** Curriculum Learning TPGD

---

1: For level 1 curriculum, use the following input: Initial parameters: $\mathbf{B}^{(0)} \in \mathbb{R}^{d \times k}$ and $\mathbf{W}_1^{(0)} \in \mathbb{R}^{k \times T_1}$ satisfy the same initialization as Algorithm 1 with scaling parameter $\alpha_1$. Step-sizes: $\eta_{1,1}$ for the first phase and $\eta_{2,1}$ for the second phase. Iteration counts: $K_1$. Total sample size for each task $t \in [T_1]$: $N$.

2: Run Algorithm 1 for level 1 curriculum.

3: Output $\widehat{\mathbf{B}} = \mathbf{B}^{(K_1)} \in \mathbb{R}^{d \times k}$, and $\widehat{\mathbf{W}}_1 = \mathbf{W}_1^{(K_1)} \in \mathbb{R}^{k \times T_1}$.

4: **for** level $j$ from 2 to $D$ **do**

5:    Use the following input: Total sample size for each task $t \in [T_1]$: $N$. Initial parameters: $\mathbf{B}^{(0)} = \widehat{\mathbf{B}}$, and $\mathbf{W}_j^{(0)} = \widetilde{\mathbf{W}}_j$, where

$$\widetilde{\mathbf{W}}_j := \operatorname*{argmin}_{\mathbf{W}_j \in \mathbb{R}^{k \times T_j}} \frac{1}{2N_j} \sum_{t=1}^{T_j} \left\| \mathbf{y}_{t,j}[N_j] - [\mathbf{X}_{t,j}(\cdot, [N_j])]^\top \widehat{\mathbf{B}} \mathbf{w}_{t,j} \right\|^2,$$

   $N_j \leq N$, and $\mathbf{W}_j = (\mathbf{w}_{1,j}, \cdots, \mathbf{w}_{T_j,j})$. Step-size: $\eta_{2,j}$. Iteration counts: $K_j$.

6:    Run the phase II of Algorithm 1 with sample size $N - N_j$.

7:    Output $\widehat{\mathbf{W}}_j = \mathbf{W}_j^{(K_j)} \in \mathbb{R}^{k \times T_j}$.

8: **end for**

---

For simplicity, we denote $\mathbf{y}_{t,j}[N_j]$ and $\mathbf{X}_{t,j}(\cdot, [N_j])$ as $\mathbf{y}_{t,j}$ and $\mathbf{X}_{t,j}$, respectively, for any $j \in [D]$ and $t \in [T_j]$. According to the optimality of $\mathbf{W}_j^{(0)}$ for any $j \in [2:D]$, we have

$$
\begin{aligned}
0 \leq & \frac{1}{N_j} \sum_{t=1}^{T_j} \left\langle \mathbf{X}_{t,j}^\top \widehat{\mathbf{B}} \left( \mathbf{w}_{t,j}^* - \mathbf{w}_{t,j}^{(0)} \right), \mathbf{X}_{t,j}^\top \widehat{\mathbf{B}} \left( \mathbf{w}_{t,j}^{(0)} - \mathbf{w}_{t,j}^* \right) \right\rangle \\
& + \frac{1}{N_j} \sum_{t=1}^{T_j} \left\langle \mathbf{X}_{t,j}^\top \widehat{\mathbf{B}} \left( \mathbf{w}_{t,j}^{(0)} - \mathbf{w}_{t,j}^* \right), \mathbf{X}_{t,j}^\top (\widehat{\mathbf{B}} - \mathbf{B}^*) \mathbf{w}_{t,j}^* - \mathbf{z}_{t,j} \right\rangle \\
& + \frac{1}{N_j} \left\langle \mathbf{X}_{t,j}^\top (\widehat{\mathbf{B}} - \mathbf{B}^*) \left( \mathbf{w}_{t,j}^{(0)} - \mathbf{w}_{t,j}^* \right), \mathbf{X}_{t,j}^\top (\widehat{\mathbf{B}} - \mathbf{B}^*) \mathbf{w}_{t,j}^* - \mathbf{z}_{t,j} \right\rangle.
\end{aligned}
\tag{61}
$$

Combining Cauchy-Schwarz inequality with Eq. (61) yields that

$$
\begin{aligned}
& \frac{1}{N_j} \sum_{t=1}^{T_j} \left\| \mathbf{X}_{t,j}^\top \widehat{\mathbf{B}} \left( \mathbf{w}_{t,j}^* - \mathbf{w}_{t,j}^{(0)} \right) \right\|^2 \\
\leq & \frac{1}{N_j} \left\langle \sum_{t=1}^{T_j} \mathbf{X}_{t,j} \mathbf{z}_{t,j} \mathbf{e}_t^\top, \widehat{\mathbf{B}} \left( \mathbf{W}_j^* - \mathbf{W}_j^{(0)} \right) \right\rangle + \frac{1}{N_j} \sum_{t=1}^{T_j} \left\| \mathbf{X}_{t,j}^\top \widehat{\mathbf{B}} \left( \mathbf{w}_{t,j}^{(0)} - \mathbf{w}_{t,j}^* \right) \right\| \cdot \left\| \mathbf{X}_{t,j}^\top (\widehat{\mathbf{B}} - \mathbf{B}^*) \mathbf{w}_{t,j}^* \right\| \\
& + \frac{1}{N_j} \left\langle \sum_{t=1}^{T_j} \mathbf{X}_{t,j} \mathbf{z}_{t,j} \mathbf{e}_t^\top, \left( \widehat{\mathbf{B}} - \mathbf{B}^* \right) \left( \mathbf{W}_j^* - \mathbf{W}_j^{(0)} \right) \right\rangle \\
& + \frac{1}{N_j} \sum_{t=1}^{T_j} \left\| \mathbf{X}_{t,j}^\top \left( \widehat{\mathbf{B}} - \mathbf{B}^* \right) \left( \mathbf{w}_{t,j}^{(0)} - \mathbf{w}_{t,j}^* \right) \right\| \cdot \left\| \mathbf{X}_{t,j}^\top (\widehat{\mathbf{B}} - \mathbf{B}^*) \mathbf{w}_{t,j}^* \right\|.
\end{aligned}
\tag{62}
$$

Since $\{\mathbf{X}_{t,j}/\sqrt{N_j}\}_{j \in [T_j]}$ satisfies Assumption 3.2, we obtain

$$
\begin{aligned}
\sigma_k^2 \left( \widehat{\mathbf{B}} \right) \left\| \mathbf{W}_j^* - \mathbf{W}_j^{(0)} \right\|_{\mathrm{F}}^2 \overset{(a)}{\lesssim} & \frac{\left[ \sigma_1(\widehat{\mathbf{B}}) + \left\| \widehat{\mathbf{B}} - \mathbf{B}^* \right\|_{\mathrm{F}} \right] \sigma^{(j)} \sqrt{T_j k}}{\sqrt{N_j}} \left\| \mathbf{W}_j^* - \mathbf{W}_j^{(0)} \right\|_{\mathrm{F}} \\
& + \sigma_1(\mathbf{W}_j^*) \left( \left\| \widehat{\mathbf{B}} - \mathbf{B}^* \right\| + \sigma_1(\widehat{\mathbf{B}}) \right) \left\| \widehat{\mathbf{B}} - \mathbf{B}^* \right\|_{\mathrm{F}} \left\| \mathbf{W}_j^* - \mathbf{W}_j^{(0)} \right\|_{\mathrm{F}} \\
\overset{(b)}{\lesssim} & \frac{\left[ \sigma_1(\mathbf{B}^*) + \left\| \widehat{\mathbf{B}} - \mathbf{B}^* \right\|_{\mathrm{F}} \right] \sigma^{(j)} \sqrt{T_j k}}{\sqrt{N_j}} \left\| \mathbf{W}_j^* - \mathbf{W}_j^{(0)} \right\|_{\mathrm{F}}
\end{aligned}
$$

$$+ \sigma_1(\mathbf{W}_j^*) \left( \left\| \widehat{\mathbf{B}} - \mathbf{B}^* \right\| + \sigma_1(\mathbf{B}^*) \right) \left\| \widehat{\mathbf{B}} - \mathbf{B}^* \right\|_{\mathrm{F}} \left\| \mathbf{W}_j^* - \mathbf{W}_j^{(0)} \right\|_{\mathrm{F}}, \qquad (63)$$

with high probability by using Eq. (62), where (a) is derived from Lemma B.1 and Cauchy-Schwarz inequality, and (b) is obtained by $\sigma_1(\widehat{\mathbf{B}}) \leq \sigma_1(\mathbf{B}^*) + \left\| \widehat{\mathbf{B}} - \mathbf{B}^* \right\|_{\mathrm{F}}$. Finally, we derive the following estimation for $\left\| \mathbf{W}_j^* - \mathbf{W}_j^{(0)} \right\|_{\mathrm{F}}$ with high probability:

$$\left\| \mathbf{W}_j^* - \mathbf{W}_j^{(0)} \right\|_{\mathrm{F}} \lesssim \frac{\kappa(\mathbf{B}^*)}{\sigma_k(\mathbf{B}^*)} \cdot \left( \frac{\sigma^{(j)} \sqrt{T_j k}}{\sqrt{N_j}} + \sigma_1(\mathbf{W}_j^*) \left\| \widehat{\mathbf{B}} - \mathbf{B}^* \right\|_{\mathrm{F}} \right). \qquad (64)$$

Combining Eq. (64) with Eq. (60) yields that $\left\| \mathbf{W}_j^* - \mathbf{W}_j^{(0)} \right\|_{\mathrm{F}} \leq \sigma_k^{1/2}(\mathbf{B}^*\mathbf{W}_j^*)/16$. Because the initialization $(\mathbf{B}^{(0)}, \mathbf{W}_j^{(0)})$ satisfies the condition of Theorem A.5 for the level $j$ curriculum, we can apply that theorem directly to obtain

$$\left[ \mathrm{dist} \left( \mathbf{W}_j^{(K_j)}, \mathbf{W}_j^* \right) \right]^2 \lesssim \frac{\left( \sigma^{(j)} \right)^2 \mathrm{tr}(\mathbf{B}^*\mathbf{W}_j^*) T_j}{\sigma_k^2(\mathbf{B}^*\mathbf{W}_j^*) N},$$

with high probability. Finally, following the same argument as in the proof of Corollary 5.3, we complete the proof. $\qquad \square$

## B. Technical Lemmas

**Lemma B.1.** *Under Assumption 3.2, the noise matrix $\frac{1}{N} \sum_{t=1}^T \mathbf{X}_t \mathbf{z}_t (\mathbf{v}_t^*)^\top$ satisfies*

$$\frac{1}{N} \left\| \sum_{t=1}^T \mathbf{X}_t \mathbf{z}_t (\mathbf{v}_t^*)^\top \right\| \leq \mathcal{O} \left( \frac{\sigma(1+\delta)^{\frac{1}{2}} \left( 1 + \log \left( \widetilde{\delta}^{-1} \right) \right)^{\frac{1}{2}} \log \left( \widetilde{\delta}^{-1} \right) \log(T)}{\sqrt{N}} \cdot \left( \max \{d, T\} + d \right)^{\frac{1}{2}} \right),$$

*with probability at least $1 - \widetilde{\delta}$.*

*Proof.* We define $\mathbf{Y} := \frac{1}{\sqrt{N}} \sum_{t=1}^T \underbrace{\mathbf{X}_t \mathbf{z}_t (\mathbf{v}_t^*)^\top}_{\sqrt{N} \mathbf{R}_t} = \sum_{t=1}^T \mathbf{R}_t$. Moreover, we derive that

$$\mathbb{E}_{\mathbf{z}} \left[ \mathbf{Y} \mathbf{Y}^\top \right] = \frac{1}{N} \mathbb{E}_{\mathbf{z}} \left[ \sum_{t=1}^T \mathbf{X}_t \mathbf{z}_t \mathbf{z}_t^\top \mathbf{X}_t^\top \right] \overset{(a)}{=} \frac{\sigma^2}{N} \sum_{t=1}^T \mathbf{X}_t \mathbf{X}_t^\top,$$

$$\mathbb{E}_{\mathbf{z}} \left[ \mathbf{Y}^\top \mathbf{Y} \right] = \frac{1}{N} \mathbf{V}^* \mathbb{E}_{\mathbf{z}} \left[ \begin{pmatrix} \mathbf{z}_1^\top \mathbf{X}_1^\top \mathbf{X}_1 \mathbf{z}_1 & \cdots & \mathbf{z}_1^\top \mathbf{X}_1^\top \mathbf{X}_1 \mathbf{z}_1 \\ \vdots & \ddots & \vdots \\ \mathbf{z}_T^\top \mathbf{X}_T^\top \mathbf{X}_1 \mathbf{z}_1 & \cdots & \mathbf{z}_T^\top \mathbf{X}_T^\top \mathbf{X}_T \mathbf{z}_T \end{pmatrix} \right] (\mathbf{V}^*)^\top$$

$$\overset{(b)}{=} \frac{\sigma^2}{N} \mathbf{V}^* \mathbf{diag} \left\{ \mathrm{tr} \left( \mathbf{X}_1^\top \mathbf{X}_1 \right), \cdots, \mathrm{tr} \left( \mathbf{X}_T^\top \mathbf{X}_T \right) \right\} (\mathbf{V}^*)^\top,$$

where $\mathbf{V}^* = (\mathbf{v}_1^*, \cdots, \mathbf{v}_T^*)$, (a) is derived from that $\mathbb{E} \left[ \mathbf{z}_t \mathbf{z}_t^\top \right] = \sigma^2 \mathbf{I}_N$ for any $t \in [T]$; (b) follows from that $\mathbb{E} \left[ \mathbf{z}_t^\top \mathbf{A} \mathbf{z}_t \right] = \sigma^2 \mathrm{tr}(\mathbf{A})$ for any $t \in [T]$ and matrix $A \in \mathbb{R}^{N \times N}$, and the fact that $\mathbf{z}_t$ is independent with $\mathbf{z}_{t'}$ for any $t \neq t'$. For simplicity, we denote $\mathbf{V}_1 = \mathbb{E}_{\mathbf{z}} \left[ \mathbf{Y} \mathbf{Y}^\top \right]$ and $\mathbf{V}_2 = \mathbb{E}_{\mathbf{z}} \left[ \mathbf{Y}^\top \mathbf{Y} \right]$. According to Assumption 3.2, $\alpha_t(\cdot) := \frac{1}{\sqrt{N}} \mathbf{X}_t^\top \circ$ satisfies

$$|\langle \alpha_t(\mathbf{v}), \alpha_t(\mathbf{u}) \rangle| \leq |\langle \mathbf{v}, \mathbf{u} \rangle| + \delta \|\mathbf{v}\| \|\mathbf{u}\|, \qquad (65)$$

utilizing Lemma D.1 for any $t \in [T]$ and $\mathbf{u}, \mathbf{v} \in \mathbb{R}^d$. Therefore, for any $t \in [T]$, we can obtain $(1-\delta)d \leq \frac{1}{N} \mathrm{tr}(\mathbf{X}_t^\top \mathbf{X}_t) \leq (1+\delta)d$ by applying $\mathbf{v} = \mathbf{u} = \mathbf{e}_i \in \mathbb{R}^d$ to Eq. (65) over $i \in [d]$. Similarly, one can also notice that $\mathrm{tr} \left( \frac{1}{N} \sum_{t=1}^T \mathbf{X}_t \mathbf{X}_t^\top \right) \leq (1+\delta)dT$ and $(1-\delta)T \leq \left\| \frac{1}{N} \sum_{t=1}^T \mathbf{X}_t \mathbf{X}_t^\top \right\| \leq (1+\delta)T$. Denote $\mathbf{V} = \mathbf{diag} \{\mathbf{V}_1, \mathbf{V}_2\}$ and its intrinsic dimension $d_{\mathbf{V}} = \mathrm{tr}(\mathbf{V})/\|\mathbf{V}\|$. We have $\mathrm{tr}(\mathbf{V}) \leq 2(1+\delta)\sigma^2 dT$, and $\|\mathbf{V}\| \geq (1-\delta)d$. Therefore $d_{\mathbf{V}} \leq (1+\delta)T/(1-\delta)$.

Note that

$$\|\mathbf{R}_t\|^2 = \frac{1}{N} \max_{u \in \mathbb{R}^T} \frac{\langle \mathbf{v}_t^*, \mathbf{u} \rangle^2}{\|\mathbf{u}\|^2} \|\mathbf{X}_t \mathbf{z}_t\|^2 = \frac{1}{N} \mathbf{z}_t^\top \mathbf{X}_t^\top \mathbf{X}_t \mathbf{z}_t \cdot \max_{u \in \mathbb{R}^T} \frac{\langle \mathbf{v}_t^*, \mathbf{u} \rangle^2}{\|\mathbf{u}\|^2}.$$

Based on Hanson-Wright inequality, each $\mathbf{R}_t$ has the following $\ell_2$ norm upper bound

$$\left\|\mathbf{R}_t\right\|^2 \lesssim \frac{\sigma^2}{N} \operatorname{tr}\left(\mathbf{X}_t^\top \mathbf{X}_t\right) + \frac{\sigma^2 \log(\widetilde{\delta}^{-1})}{N}\left\|\mathbf{X}_t^\top \mathbf{X}_t\right\| + \frac{\sigma^2 \log^{\frac{1}{2}}(\widetilde{\delta}^{-1})}{N}\left\|\mathbf{X}_t^\top \mathbf{X}_t\right\|_{\mathrm{F}},$$

with probability at least $1 - \widetilde{\delta}$. Since $\left\|\mathbf{X}_t^\top \mathbf{X}_t\right\|_{\mathrm{F}} \le \operatorname{tr}\left(\mathbf{X}_t^\top \mathbf{X}_t\right)$, we define

$$
\begin{aligned}
L &:= \left[(1+\delta)\left(1 + 2\log\left(\widetilde{\delta}^{-1}\right)\right) d\right]^{\frac{1}{2}} \\
&\ge \left[\frac{\left(1 + \log^{\frac{1}{2}}\left(\widetilde{\delta}^{-1}\right)\right)}{N} \operatorname{tr}\left(\mathbf{X}_t^\top \mathbf{X}_t\right) + \frac{\log(\widetilde{\delta}^{-1})}{N}\left\|\mathbf{X}_t^\top \mathbf{X}_t\right\|\right]^{\frac{1}{2}} \ge \sigma^{-1}\left\|\mathbf{R}_t\right\|.
\end{aligned}
$$

According to the matrix bernstein (Theorem 7.3.1 in Tropp et al. (2015)), we have

$$\|\mathbf{Y}\| \lesssim \sigma\left[(\mathbf{Var})^{\frac{1}{2}} + L\right] \log\left(\widetilde{\delta}^{-1}\right) \log\left(d_{\mathbf{V}}\right),$$

with probability at least $1 - \widetilde{\delta}$, where $\mathbf{Var} := (1+\delta)\max\{d, T\} \ge \sigma^{-2}\max\{\|\mathbf{V}_1\|, \|\mathbf{V}_2\|\}$. $\qquad\square$

**Lemma B.2.** *For diagonal positive definite matrix* $\boldsymbol{\Sigma}_* \in \mathbb{R}^{k \times k}$ *and matrices* $\mathbf{U}, \mathbf{V} \in \mathbb{R}^{k \times k}$, *suppose* $\max\{\|\mathbf{U}\|, \|\mathbf{V}\|\} \le C_1 \sigma_{\max}^{\frac{1}{2}}(\boldsymbol{\Sigma}_*)$, $\left\|\mathbf{U}\mathbf{V}^\top - \boldsymbol{\Sigma}_*\right\| \le \epsilon_0$ *and* $\|\mathbf{U} - \mathbf{V}\| \le \epsilon_1$. *Then we have*

$$\operatorname{dist}\left(\begin{pmatrix}\mathbf{U} \\ \mathbf{V}\end{pmatrix}, \begin{pmatrix}\boldsymbol{\Sigma}_*^{\frac{1}{2}} \\ \boldsymbol{\Sigma}_*^{\frac{1}{2}}\end{pmatrix}\right) \lesssim \operatorname{tr}^{\frac{1}{2}}(\boldsymbol{\Sigma}_*)\left(C_1 \kappa^{\frac{1}{2}}(\boldsymbol{\Sigma}_*) + 1\right)\left[\left(\kappa(\boldsymbol{\Sigma}_*) + \sigma_{\min}^{-\frac{1}{2}}(\boldsymbol{\Sigma}_*)\right)\epsilon_1 + \sigma_{\min}^{-1}(\boldsymbol{\Sigma}_*)\epsilon_0\right].$$

*Proof.* Since $\left\|\mathbf{U}\mathbf{V}^\top - \boldsymbol{\Sigma}_*\right\| \le \epsilon_0$ and $\|\mathbf{U} - \mathbf{V}\| \le \epsilon_1$, we have

$$\left\|\left(\boldsymbol{\Sigma}_*^{-\frac{1}{2}}\mathbf{U}\right)\left(\boldsymbol{\Sigma}_*^{-\frac{1}{2}}\mathbf{V}\right)^\top - \mathbf{I}_k\right\| \le \sigma_{\min}^{-1}(\boldsymbol{\Sigma}_*)\epsilon_0 \quad \text{and} \quad \left\|\boldsymbol{\Sigma}_*^{-\frac{1}{2}}\mathbf{U} - \boldsymbol{\Sigma}_*^{-\frac{1}{2}}\mathbf{V}\right\| \le \sigma_{\min}^{-\frac{1}{2}}(\boldsymbol{\Sigma}_*)\epsilon_1. \tag{66}$$

By denoting $\mathbf{U}_* = \boldsymbol{\Sigma}_*^{-\frac{1}{2}}\mathbf{U}$ and $\mathbf{V}_* = \boldsymbol{\Sigma}_*^{-\frac{1}{2}}\mathbf{V}$, we can obtain that

$$
\begin{aligned}
\left\|\mathbf{U}_*\mathbf{U}_*^\top - \mathbf{I}_k\right\| &\le \left\|\mathbf{U}_*\mathbf{U}_*^\top - \mathbf{U}_*\mathbf{V}_*^\top\right\| + \left\|\mathbf{U}_*\mathbf{V}_*^\top - \mathbf{I}_k\right\| \\
&\le \sigma_{\min}^{-\frac{1}{2}}(\boldsymbol{\Sigma}_*)\|\mathbf{U}_*\|\epsilon_1 + \sigma_{\min}^{-1}(\boldsymbol{\Sigma}_*)\epsilon_0 \le \kappa(\boldsymbol{\Sigma}_*)\epsilon_1 + \sigma_{\min}^{-1}(\boldsymbol{\Sigma}_*)\epsilon_0.
\end{aligned}
\tag{67}
$$

Consider the singular value decomposition (SVD) of $\mathbf{U}_*$ and $\mathbf{V}_*$:

$$\mathbf{U}_* = \mathbf{P}_{\mathbf{U}_*}\boldsymbol{\Sigma}_{\mathbf{U}_*}\mathbf{Q}_{\mathbf{U}_*}^\top \quad \text{and} \quad \mathbf{V}_* = \mathbf{P}_{\mathbf{V}_*}\boldsymbol{\Sigma}_{\mathbf{V}_*}\mathbf{Q}_{\mathbf{V}_*}^\top,$$

where $\boldsymbol{\Sigma}_{\mathbf{U}_*}$ and $\boldsymbol{\Sigma}_{\mathbf{U}_*}$ are diagonal positive definite matrices in $\mathbb{R}^k$, $\mathbf{P}_{\mathbf{U}_*}, \mathbf{P}_{\mathbf{V}_*}, \mathbf{Q}_{\mathbf{U}_*}$ and $\mathbf{Q}_{\mathbf{V}_*}$ are orthogonal matrices in $\mathbb{R}^k$. Therefore, by Eq. (67), it can be derived that

$$\left\|\boldsymbol{\Sigma}_{\mathbf{U}_*}^2 - \mathbf{I}_k\right\| \le \kappa(\boldsymbol{\Sigma}_*)\epsilon_1 + \sigma_{\min}^{-1}(\boldsymbol{\Sigma}_*)\epsilon_0,$$

which implicates that

$$\left\|\mathbf{U}_*\left(\mathbf{Q}_{\mathbf{U}_*}\mathbf{P}_{\mathbf{U}_*}^\top\right) - \mathbf{I}_k\right\| = \left\|\boldsymbol{\Sigma}_{\mathbf{U}_*} - \mathbf{I}_k\right\| \le \kappa(\boldsymbol{\Sigma}_*)\epsilon_1 + \sigma_{\min}^{-1}(\boldsymbol{\Sigma}_*)\epsilon_0. \tag{68}$$

Similar, one can notice that

$$\left\|\mathbf{V}_*\left(\mathbf{Q}_{\mathbf{V}_*}\mathbf{P}_{\mathbf{V}_*}^\top\right) - \mathbf{I}_k\right\| = \left\|\boldsymbol{\Sigma}_{\mathbf{V}_*} - \mathbf{I}_k\right\| \le \kappa(\boldsymbol{\Sigma}_*)\epsilon_1 + \sigma_{\min}^{-1}(\boldsymbol{\Sigma}_*)\epsilon_0. \tag{69}$$

According to Eq. (69), we have

$$
\begin{aligned}
\left\|\mathbf{Q}_{\mathbf{U}_*}\mathbf{P}_{\mathbf{U}_*}^\top - \mathbf{Q}_{\mathbf{V}_*}\mathbf{P}_{\mathbf{V}_*}^\top\right\| &\le \left\|\mathbf{Q}_{\mathbf{U}_*}\left(\mathbf{I}_k - \boldsymbol{\Sigma}_{\mathbf{U}_*}\right)\mathbf{P}_{\mathbf{U}_*}^\top\right\| + \|\mathbf{U}_* - \mathbf{V}_*\| + \left\|\mathbf{Q}_{\mathbf{V}_*}\left(\boldsymbol{\Sigma}_{\mathbf{V}_*} - \mathbf{I}_k\right)\mathbf{P}_{\mathbf{V}_*}^\top\right\| \\
&\overset{(a)}{\le} \left(2\kappa(\boldsymbol{\Sigma}_*) + \sigma_{\min}^{-\frac{1}{2}}(\boldsymbol{\Sigma}_*)\right)\epsilon_1 + 2\sigma_{\min}^{-1}(\boldsymbol{\Sigma}_*)\epsilon_0,
\end{aligned}
\tag{70}
$$

where (a) follows from combining Eq.(66) and Eqs. (68)-(69). Utilizing Eqs. (69)-(70), we obtain

$$
\begin{aligned}
\left\|\mathbf{V}_*\left(\mathbf{Q}_{\mathbf{U}_*}\mathbf{P}_{\mathbf{U}_*}^\top\right) - \mathbf{I}_k\right\| &\leq \left\|\mathbf{V}_*\left(\mathbf{Q}_{\mathbf{U}_*}\mathbf{P}_{\mathbf{U}_*}^\top\right) - \mathbf{V}_*\left(\mathbf{Q}_{\mathbf{V}_*}\mathbf{P}_{\mathbf{V}_*}^\top\right)\right\| + \left\|\mathbf{V}^*\left(\mathbf{Q}_{\mathbf{V}_*}\mathbf{P}_{\mathbf{V}_*}^\top\right) - \mathbf{I}_k\right\| \\
&\overset{(b)}{\leq} C_1\kappa^{\frac{1}{2}}(\boldsymbol{\Sigma}_*)\left\|\mathbf{Q}_{\mathbf{U}_*}\mathbf{P}_{\mathbf{U}_*}^\top - \mathbf{Q}_{\mathbf{V}_*}\mathbf{P}_{\mathbf{V}_*}^\top\right\| + \left\|\mathbf{V}^*\left(\mathbf{Q}_{\mathbf{V}_*}\mathbf{P}_{\mathbf{V}_*}^\top\right) - \mathbf{I}_k\right\| \\
&\leq \left(2C_1\kappa^{\frac{1}{2}}(\boldsymbol{\Sigma}_*) + 1\right)\left[\left(\kappa(\boldsymbol{\Sigma}_*) + \sigma_{\min}^{-\frac{1}{2}}(\boldsymbol{\Sigma}_*)\right)\epsilon_1 + \sigma_{\min}^{-1}(\boldsymbol{\Sigma}_*)\epsilon_0\right],
\end{aligned}
$$

where (b) is derived from $\|\mathbf{V}_*\| \leq \left\|\boldsymbol{\Sigma}_*^{-\frac{1}{2}}\right\| \cdot \|\mathbf{V}\|$. Based on the definitions of $\mathbf{U}_*$ and $\mathbf{V}_*$, we complete the proof. $\qquad\square$

**Lemma B.3.** *Let the initial scale $\alpha$, step-size $\eta_1$ and sample size $N$ satisfy the conditions specified in Theorem A.1, and suppose Assumption 3.2 holds with $\delta$ such that Eq. (13) holds. Then the last-iterate output $\left(\mathbf{B}^{(K_1/2)}, \mathbf{W}^{(K_1/2)}\right)$ of the first phase of Algorithm 2 satisfies*

$$
\begin{aligned}
\left\|\widetilde{\mathbf{B}}_{[k],[k]}^{(\tau)} - \widetilde{\mathbf{W}}_{[k],[k]}^{(\tau)}\right\| &\lesssim \alpha + \frac{\delta\sigma_1^{3/2}(\boldsymbol{\Sigma}^*)}{\sigma_k(\boldsymbol{\Sigma}^*)} + \frac{\sigma\left((1+\delta)\sigma_1(\boldsymbol{\Sigma}^*)(\max\{d,T\}+d)\right)^{\frac{1}{2}}}{\sigma_k(\boldsymbol{\Sigma}^*)\sqrt{N}}, \\
\max\left\{\left\|\widetilde{\mathbf{B}}_{[k+1:d],[k]}^{(\tau)}\right\|, \left\|\widetilde{\mathbf{W}}_{[k],[k+1:T]}^{(\tau)}\right\|\right\} &\lesssim \alpha + \frac{\delta\sigma_1^{3/2}(\boldsymbol{\Sigma}^*)}{\sigma_k(\boldsymbol{\Sigma}^*)} + \frac{\sigma\left((1+\delta)\sigma_1(\boldsymbol{\Sigma}^*)(\max\{d,T\}+d)\right)^{\frac{1}{2}}}{\sigma_k(\boldsymbol{\Sigma}^*)\sqrt{N}},
\end{aligned}
\tag{71}
$$

*for any $\tau \leq K_1/2$, where $\widetilde{\mathbf{B}}^{(\tau)} = (\mathbf{U}^*)^\top \mathbf{B}^{(\tau)}$ and $\widetilde{\mathbf{W}}^{(\tau)} = \mathbf{W}^{(\tau)}\mathbf{V}^*$.*

*Proof.* Applying Lemma B.4 directly, we complete the proof. $\qquad\square$

**Lemma B.4.** *Let $\alpha, \eta_1$ and $N$ satisfy the conditions specified in Theorem A.1, and suppose Assumption 3.2 holds with $\delta$ such that*

$$
\delta \leq \min\left\{\frac{c_1}{\kappa^6(\boldsymbol{\Sigma}^*)\log\left(\frac{\sqrt{\sigma_k(\boldsymbol{\Sigma}^*)}}{max\{d,T\}\alpha}\right)}, \frac{c_1}{\kappa^3(\boldsymbol{\Sigma}^*)\sqrt{k}}, \frac{1}{128}\right\},
\tag{72}
$$

*For simplicity, let $\mathbf{U}^{(\tau)}$ and $\mathbf{V}^{(\tau)}$ denote the submatrices consisting of the first $k$ rows of $\widetilde{\mathbf{B}}^{(\tau)}$ and $(\widetilde{\mathbf{W}}^{(\tau)})^\top$, respectively. Similarly, let $\mathbf{J}^{(\tau)}$ and $\mathbf{K}^{(\tau)}$ represent the submatrices formed by the remaining $d-k$ rows of $\widetilde{\mathbf{B}}^{(\tau)}$ and the remaining $T-k$ rows of $(\widetilde{\mathbf{W}}^{(\tau)})^\top$. Then during the first phase of Algorithm 2, we have Then during the gradient descent process (12) when $\tau \leq T_0 = \tilde{\mathcal{O}}\left(\frac{1}{\eta_1\sigma_k(\boldsymbol{\Sigma}^*)}\right)$, we have*

$$
\left\|\mathbf{U}^{(\tau)} - \mathbf{V}^{(\tau)}\right\| \lesssim \alpha + \frac{\delta\sigma_1^{3/2}(\boldsymbol{\Sigma}^*)}{\sigma_k(\boldsymbol{\Sigma}^*)} + \frac{\sqrt{\sigma_1(\boldsymbol{\Sigma}^*)}}{\sigma_k(\boldsymbol{\Sigma}^*)}\|\mathbf{E}\|,
\tag{73}
$$

$$
\left\|\mathbf{J}^{(\tau)}\right\| \lesssim \alpha + \frac{\delta\sigma_1^{3/2}(\boldsymbol{\Sigma}^*)}{\sigma_k(\boldsymbol{\Sigma}^*)} + \frac{\sqrt{\sigma_1(\boldsymbol{\Sigma}^*)}}{\sigma_k(\boldsymbol{\Sigma}^*)}\|\mathbf{E}\|,
\tag{74}
$$

*for any $\tau \leq K_1/2$.*

*Proof.* Denote $\Delta(\mathbf{M}) = \mathcal{A}^*\mathcal{A}(\mathbf{M}) - \mathbf{M}$, and $\Delta_1(\mathbf{M})$. Partition $\Delta(\mathbf{M}), \boldsymbol{\Sigma}^*$ and $\mathbf{E}$ into four sub-matrices as

$$
\Delta(\mathbf{M}) = \begin{bmatrix} \Delta_1(\mathbf{M}) & \Delta_2(\mathbf{M}) \\ \Delta_3(\mathbf{M}) & \Delta_4(\mathbf{M}) \end{bmatrix}, \qquad \boldsymbol{\Sigma}^* = \begin{bmatrix} \boldsymbol{\Sigma}_{[k]}^* & 0 \\ 0 & 0 \end{bmatrix}, \qquad \mathbf{E} = \begin{bmatrix} \mathbf{E}_1 & \mathbf{E}_2 \\ \mathbf{E}_3 & \mathbf{E}_4 \end{bmatrix}.
$$

Where $\Delta_1(\mathbf{M}), \mathbf{E}_1, \boldsymbol{\Sigma}_k^* \in \mathbb{R}^{k\times k}, \Delta_2(\mathbf{M}), \mathbf{E}_2 \in \mathbb{R}^{k\times(T-K)}$. Then, the iterations of $\widetilde{\mathbf{B}}^{(\tau)}$ and $\widetilde{\mathbf{W}}^{(\tau)}$ can be represented by the following equations:

$$
\begin{aligned}
\mathbf{U}^{(\tau+1)} = \mathbf{U}^{(\tau)} &+ \eta_1\boldsymbol{\Sigma}_k^*\mathbf{V}^{(\tau)} - \eta_1\mathbf{U}^{(\tau)}\left((\mathbf{V}^{(\tau)})^\top\mathbf{V}^{(\tau)} + (\mathbf{K}^{(\tau)})^\top\mathbf{K}^{(\tau)}\right) \\
&- \eta_1\Delta_1\left(\widetilde{\mathbf{B}}^{(\tau)}\widetilde{\mathbf{W}}^{(\tau)} - \boldsymbol{\Sigma}^*\right)\mathbf{V}^{(\tau)} + \eta_1\Delta_2\left(\widetilde{\mathbf{B}}^{(\tau)}\widetilde{\mathbf{W}}^{(\tau)} - \boldsymbol{\Sigma}^*\right)\mathbf{K}^{(\tau)} \\
&- \eta_1\mathbf{E}_1\mathbf{V}^{(\tau)} - \eta_1\mathbf{E}_2\mathbf{K}^{(\tau)},
\end{aligned}
\tag{75}
$$

$$\mathbf{V}^{(\tau+1)} = \mathbf{V}^{(\tau)} + \eta_1 \boldsymbol{\Sigma}_k^* \mathbf{U}^{(\tau)} - \eta_1 \mathbf{V}^{(\tau)} \left( (\mathbf{U}^{(\tau)})^\top \mathbf{U}^{(\tau)} + (\mathbf{J}^{(\tau)})^\top \mathbf{J}^{(\tau)} \right)$$
$$- \eta_1 \Delta_1^\top \left( \widetilde{\mathbf{B}}^{(\tau)} \widetilde{\mathbf{W}}^{(\tau)} - \boldsymbol{\Sigma}^* \right) \mathbf{U}^{(\tau)} - \eta_1 \Delta_3^\top \left( \widetilde{\mathbf{B}}^{(\tau)} \widetilde{\mathbf{W}}^{(\tau)} - \boldsymbol{\Sigma}^* \right) \mathbf{J}^{(\tau)} \qquad (76)$$
$$- \eta_1 \mathbf{E}_1^\top \mathbf{U}^{(\tau)} - \eta_1 \mathbf{E}_3^\top \mathbf{J}^{(\tau)},$$

$$\mathbf{J}^{(\tau+1)} = \mathbf{J}^{(\tau)} - \eta_1 \mathbf{J}^{(\tau)} \left( (\mathbf{V}^{(\tau)})^\top \mathbf{V}^{(\tau)} + (\mathbf{K}^{(\tau)})^\top \mathbf{K}^{(\tau)} \right)$$
$$- \eta_1 \Delta_3 \left( \widetilde{\mathbf{B}}^{(\tau)} \widetilde{\mathbf{W}}^{(\tau)} - \boldsymbol{\Sigma}^* \right) \mathbf{V}^{(\tau)} - \eta_1 \Delta_4 \left( \widetilde{\mathbf{B}}^{(\tau)} \widetilde{\mathbf{W}}^{(\tau)} \right) - \boldsymbol{\Sigma}^* \right) \mathbf{K}^{(\tau)} \qquad (77)$$
$$- \eta_1 \mathbf{E}_3 \mathbf{V}^{(\tau)} - \eta_1 \mathbf{E}_4 \mathbf{K}^{(\tau)},$$

$$\mathbf{K}^{(\tau+1)} = \mathbf{K}^{(\tau)} - \eta_1 \mathbf{K}^{(\tau)} \left( (\mathbf{U}^{(\tau)})^\top \mathbf{U}^{(\tau)} + (\mathbf{J}^{(\tau)})^\top \mathbf{J}^{(\tau)} \right)$$
$$+ \eta_1 \Delta_2^\top \left( \widetilde{\mathbf{B}}^{(\tau)} \widetilde{\mathbf{W}}^{(\tau)} - \boldsymbol{\Sigma}^* \right) \mathbf{U}^{(\tau)} + \eta_1 \Delta_4^\top \left( \widetilde{\mathbf{B}}^{(\tau)} \widetilde{\mathbf{W}}^{(\tau)} \right) - \boldsymbol{\Sigma}^* \right) \mathbf{J}^{(\tau)} \qquad (78)$$
$$- \eta_1 \mathbf{E}_1^\top \mathbf{U}^{(\tau)} - \eta_1 \mathbf{E}_3^\top \mathbf{J}^{(\tau)}.$$

Next, we will inductively prove Eq. (73) and Eq. (74) when $\tau \le K_1/2$.

**Proof of Eq.** (73) By Eq. (75) and Eq. (76), we have

$$\left\| \mathbf{U}^{(\tau+1)} - \mathbf{V}^{(\tau+1)} \right\| \le \left\| \mathbf{U}^{(\tau)} - \mathbf{V}^{(\tau)} \right\| \left\| \mathbf{I}_k - \eta_1 \boldsymbol{\Sigma}_{[k]}^* - \eta_1 \left( (\mathbf{V}^{(\tau)})^\top \mathbf{V}^{(\tau)} + (\mathbf{K}^{(\tau)})^\top \mathbf{K}^{(\tau)} \right) \right\|$$
$$+ \eta_1 \left\| \mathbf{V}^{(\tau)} \right\| \left\| (\mathbf{U}^{(\tau)})^\top \mathbf{U}^{(\tau)} + (\mathbf{J}^{(\tau)})^\top \mathbf{J}^{(\tau)} - (\mathbf{V}^{(\tau)})^\top \mathbf{V}^{(\tau)} - (\mathbf{K}^{(\tau)})^\top \mathbf{K}^{(\tau)} \right\|$$
$$+ 4\eta_1 \delta \left\| \widetilde{\mathbf{B}}^{(\tau)} \widetilde{\mathbf{W}}^{(\tau)} - \boldsymbol{\Sigma}^* \right\| \max \left\{ \left\| \mathbf{U}^{(\tau)} \right\|, \left\| \mathbf{V}^{(\tau)} \right\|, \left\| \mathbf{J}^{(\tau)} \right\|, \left\| \mathbf{K}^{(\tau)} \right\| \right\}$$
$$+ \eta_1 \left\| \mathbf{E}_1 \mathbf{U}^{(\tau)} \right\| + \eta_1 \left\| \mathbf{E}_1^\top \mathbf{V}^{(\tau)} \right\|$$
$$\le (1 - \eta_1 \sigma_k(\boldsymbol{\Sigma}^*)) \left\| \mathbf{U}^{(\tau)} - \mathbf{V}^{(\tau)} \right\| + 2\eta_1 \alpha^2 \cdot 2\sqrt{\sigma_1(\boldsymbol{\Sigma}^*)}$$
$$+ 4\eta_1 \delta \cdot \left( \left\| \widetilde{\mathbf{B}}^{(\tau)} \right\| \left\| \widetilde{\mathbf{W}}^{(\tau)} \right\| + \|\boldsymbol{\Sigma}^*\| \right) \cdot 2\sqrt{\sigma_1(\boldsymbol{\Sigma}^*)} + \eta_1 \|\mathbf{E}\| \cdot 2\sqrt{\sigma_1(\boldsymbol{\Sigma}^*)}$$
$$\le (1 - \eta_1 \sigma_k(\boldsymbol{\Sigma}^*)) \left\| \mathbf{U}^{(\tau)} - \mathbf{V}^{(\tau)} \right\| + 2\eta_1 \alpha^2 \cdot 2\sqrt{\sigma_1(\boldsymbol{\Sigma}^*)} + 40\eta_1 \delta \cdot \sigma_1^{3/2}(\boldsymbol{\Sigma}^*) + 2\eta_1 \sqrt{\sigma_1(\boldsymbol{\Sigma}^*)} \|\mathbf{E}\|.$$

The third inequality holds by $\max\{\|\widetilde{\mathbf{B}}^{(\tau)}\|, \|\widetilde{\mathbf{W}}^{(\tau)}\|\} \le 2\sqrt{\sigma_1(\boldsymbol{\Sigma}^*)}$. Thus, since $\alpha = \mathcal{O}(\delta\sigma_1^{3/2}(\boldsymbol{\Sigma}^*)/\sigma_k(\boldsymbol{\Sigma}^*))$, we can get $\|\mathbf{U}^{(0)} - \mathbf{V}^{(0)}\| \le 4\alpha \le 4\alpha + \frac{40\sigma_1^{3/2}(\boldsymbol{\Sigma}^*)}{\sigma_k(\boldsymbol{\Sigma}^*)} \delta + \frac{2\sqrt{\sigma_1(\boldsymbol{\Sigma}^*)}}{\sigma_k(\boldsymbol{\Sigma}^*)} \|\mathbf{E}\|$. If

$$\left\| \mathbf{U}^{(\tau)} - \mathbf{V}^{(\tau)} \right\| \le 4\alpha + \frac{40\sigma_1^{3/2}(\boldsymbol{\Sigma}^*)}{\sigma_k(\boldsymbol{\Sigma}^*)} \delta + \frac{2\sqrt{\sigma_1(\boldsymbol{\Sigma}^*)}}{\sigma_k(\boldsymbol{\Sigma}^*)} \|\mathbf{E}\|,$$

we know that

$$\left\| \mathbf{U}^{(\tau+1)} - \mathbf{V}^{(\tau+1)} \right\| \le (1 - \eta_1 \sigma_k(\boldsymbol{\Sigma}^*)) \left( 4\alpha + \frac{40\sigma_1^{3/2}(\boldsymbol{\Sigma}^*)}{\sigma_k(\boldsymbol{\Sigma}^*)} \delta + \frac{2\sqrt{\sigma_1(\boldsymbol{\Sigma}^*)}}{\sigma_k(\boldsymbol{\Sigma}^*)} \|\mathbf{E}\| \right) + 4\eta_1 \alpha^2 \sqrt{\sigma_1(\boldsymbol{\Sigma}^*)}$$
$$+ 40\eta_1 \delta \cdot \sigma_1^{3/2}(\boldsymbol{\Sigma}^*) + 2\eta_1 \sqrt{\sigma_1(\boldsymbol{\Sigma}^*)} \|\mathbf{E}\|$$
$$\le (1 - \eta_1 \sigma_k(\boldsymbol{\Sigma}^*)) \left( 4\alpha + \frac{40\sigma_1^{3/2}(\boldsymbol{\Sigma}^*)}{\sigma_k(\boldsymbol{\Sigma}^*)} \delta + \frac{2\sqrt{\sigma_1(\boldsymbol{\Sigma}^*)}}{\sigma_k(\boldsymbol{\Sigma}^*)} \|\mathbf{E}\| \right) + 4\eta_1 \sigma_k(\boldsymbol{\Sigma}^*)\alpha$$
$$+ 40\eta_1 \delta \sigma_1^{3/2}(\boldsymbol{\Sigma}^*) + 2\eta_1 \sqrt{\sigma_1(\boldsymbol{\Sigma}^*)} \|\mathbf{E}\|$$
$$\le 4\alpha + \frac{40\sigma_1^{3/2}(\boldsymbol{\Sigma}^*)}{\sigma_k(\boldsymbol{\Sigma}^*)} \delta + \frac{2\sqrt{\sigma_1(\boldsymbol{\Sigma}^*)}}{\sigma_k(\boldsymbol{\Sigma}^*)} \|\mathbf{E}\|. \qquad (79)$$

Hence, $\|\mathbf{U}^{(\tau)} - \mathbf{V}^{(\tau)}\| \leq 4\alpha + \frac{40\sigma_1^{3/2}(\mathbf{\Sigma}^*)}{\sigma_k(\mathbf{\Sigma}^*)}\delta$ for $\tau \leq K_1/2$ by induction. The second inequality holds by $\alpha = \mathcal{O}(\sigma_k(\mathbf{\Sigma}^*)/\sqrt{\sigma_1(\mathbf{\Sigma}^*)})$.

**Proof of Eq. (74)**

Now we prove that $\mathbf{J}^{(\tau)}$ and $\mathbf{K}^{(\tau)}$ are bounded for all $\tau \leq K_1/2$. By Eq. (77) and $\max\{\|\widetilde{\mathbf{B}}^{(\tau)}\|, \|\widetilde{\mathbf{W}}^{(\tau)}\|\} \leq 2\sqrt{\sigma_1(\mathbf{\Sigma}^*)}$, denoting $C_2 = \max\{21c_2, 32\} \geq 32$, we have

$$
\begin{aligned}
\left\|\mathbf{J}^{(\tau)}\right\| &\leq \left\|\mathbf{J}^{(0)}\right\| + 2\eta_1 \sum_{\tau=0}^{K_1/2-1} \max\left\{\left\|\mathbf{V}^{(\tau)}\right\|, \left\|\mathbf{K}^{(\tau)}\right\|\right\} \cdot \left(\delta\left\|\mathbf{B}^{(\tau)}\widetilde{\mathbf{W}}^{(\tau)} - \mathbf{\Sigma}^*\right\| + \max\left\{\|\mathbf{E}_3\|, \|\mathbf{E}_4\|\right\}\right) \\
&\leq \left\|\mathbf{J}^{(0)}\right\| + \eta_1 K_1 \cdot \left(10\delta\sigma_1^{3/2}(\mathbf{\Sigma}^*) + 2\sqrt{\sigma_1(\mathbf{\Sigma}^*)}\|\mathbf{E}\|\right) \\
&\leq \|\mathbf{J}^{(0)}\| + \eta_1 \left(20\sigma_1^{3/2}(\mathbf{\Sigma}^*) + 4\sqrt{\sigma_1(\mathbf{\Sigma}^*)}\|\mathbf{E}\|\right) \cdot \widetilde{\mathcal{O}}\left(\frac{1}{\eta_1\sigma_k(\mathbf{\Sigma}^*)}\right) \\
&= \widetilde{\mathcal{O}}\left(\alpha + \frac{\delta\sigma_1^{3/2}(\mathbf{\Sigma}^*)}{\sigma_k(\mathbf{\Sigma}^*)} + \frac{\sqrt{\sigma_1(\mathbf{\Sigma}^*)}}{\sigma_k(\mathbf{\Sigma}^*)}\|\mathbf{E}\|\right).
\end{aligned}
\tag{80}
$$

Similarly, we can prove that $\|\mathbf{K}^{(\tau)}\| \leq \widetilde{\mathcal{O}}\left(\alpha + \frac{\delta\sigma_1^{3/2}(\mathbf{\Sigma}^*)}{\sigma_k(\mathbf{\Sigma}^*)} + \frac{\sqrt{\sigma_1(\mathbf{\Sigma}^*)}}{\sigma_k(\mathbf{\Sigma}^*)}\|\mathbf{E}\|\right)$ for any $\tau \in [K_1/2]$. We complete the proof of Eq. (74). $\square$

**Lemma B.5.** *Let the initial scale $\alpha$, step-size $\eta_1$ and sample size $N$ satisfy the conditions specified in Theorem A.1, and suppose Assumption 3.2 holds with $\delta$ such that Eq. (13) holds. Then the last-iterate output $\left(\mathbf{B}^{(K_1/2)}, \mathbf{W}^{(K_1/2)}\right)$ of the first phase of Algorithm 2 satisfies*

$$
\left\|\widetilde{\mathbf{B}}^{(K_1/2)}\widetilde{\mathbf{W}}^{(K_1/2)} - \mathbf{\Sigma}^*\right\| \lesssim \frac{\sigma_k(\mathbf{\Sigma}^*)}{\sqrt{k}\kappa(\mathbf{\Sigma}^*)},
\tag{81}
$$

*with probability at least $1 - C_2\exp(-C_3 k) - C_4\epsilon$ for fixed numerical constants $C_2, C_3, C_4 > 0$ when $K_1 \gtrsim \frac{\kappa(\mathbf{\Sigma}^*)}{\eta_1\sigma_k(\mathbf{\Sigma}^*)}$, where $\widetilde{\mathbf{B}}^{(\tau)} = (\mathbf{U}^*)^\top \mathbf{B}^{(\tau)}$ and $\widetilde{\mathbf{W}}^{(\tau)} = \mathbf{W}^{(\tau)}\mathbf{V}^*$.*

*Proof.* Utilizing the following equality for any $\mathbf{M} \in \mathbb{R}^{d \times T}$

$$
\|\mathbf{M}\|_\mathrm{F}^2 = \left\|\mathbf{M}(\mathbf{V}^*)^\top\right\|_\mathrm{F}^2 = \sum_{t=1}^T \|\mathbf{M}\mathbf{v}_t^*\|^2,
$$

we have

$$
(1 - \delta)\left\|\mathbf{M}(\mathbf{V}^*)^\top\right\|_\mathrm{F}^2 \leq \|\mathcal{A}(\mathbf{M})\|_\mathrm{F}^2 \leq (1 + \delta)\left\|\mathbf{M}(\mathbf{V}^*)^\top\right\|_\mathrm{F}^2.
$$

This means that $\delta$ is the RIP constant of $\mathcal{A}$. By substituting $(\mathcal{A}^*\mathcal{A} - \mathcal{I})\left(\widetilde{\mathbf{B}}^{(\tau)}\widetilde{\mathbf{W}}^{(\tau)} - \mathbf{\Sigma}^*\right) + \mathbf{E}$ for the remainder error $(\mathcal{A}^*\mathcal{A} - \mathcal{I})\left(\widetilde{\mathbf{B}}^{(\tau)}\widetilde{\mathbf{W}}^{(\tau)} - \mathbf{\Sigma}^*\right)$ considered in Theorem 3 of Soltanolkotabi et al. (2023), and combining it with the hyper-parameter setting $N \gtrsim \frac{(\max\{d,T\}+d)k\kappa^4(\mathbf{\Sigma}^*)\sigma^2}{\sigma_k^2(\mathbf{\Sigma}^*)}$ and Lemma B.1, Eq. (81) can be derived directly with probability at least $1 - C_2\exp(-C_3 k) - C_4\epsilon$. $\square$

## C. Supplementary Experiments

The ablation study (Figure 3) shows that using only *Phase I* slows convergence, whereas using only *Phase II* increases the final loss. These results demonstrate that both components of TPGD are essential for fast convergence and optimal estimation error.

Table 2 compares the number of iterations required for loss stabilization between TPGD and the SVD/QR-based approaches. The results show that the proposed TPGD algorithm not only stabilizes in fewer iterations but also achieves a lower loss value at the point of stabilization.

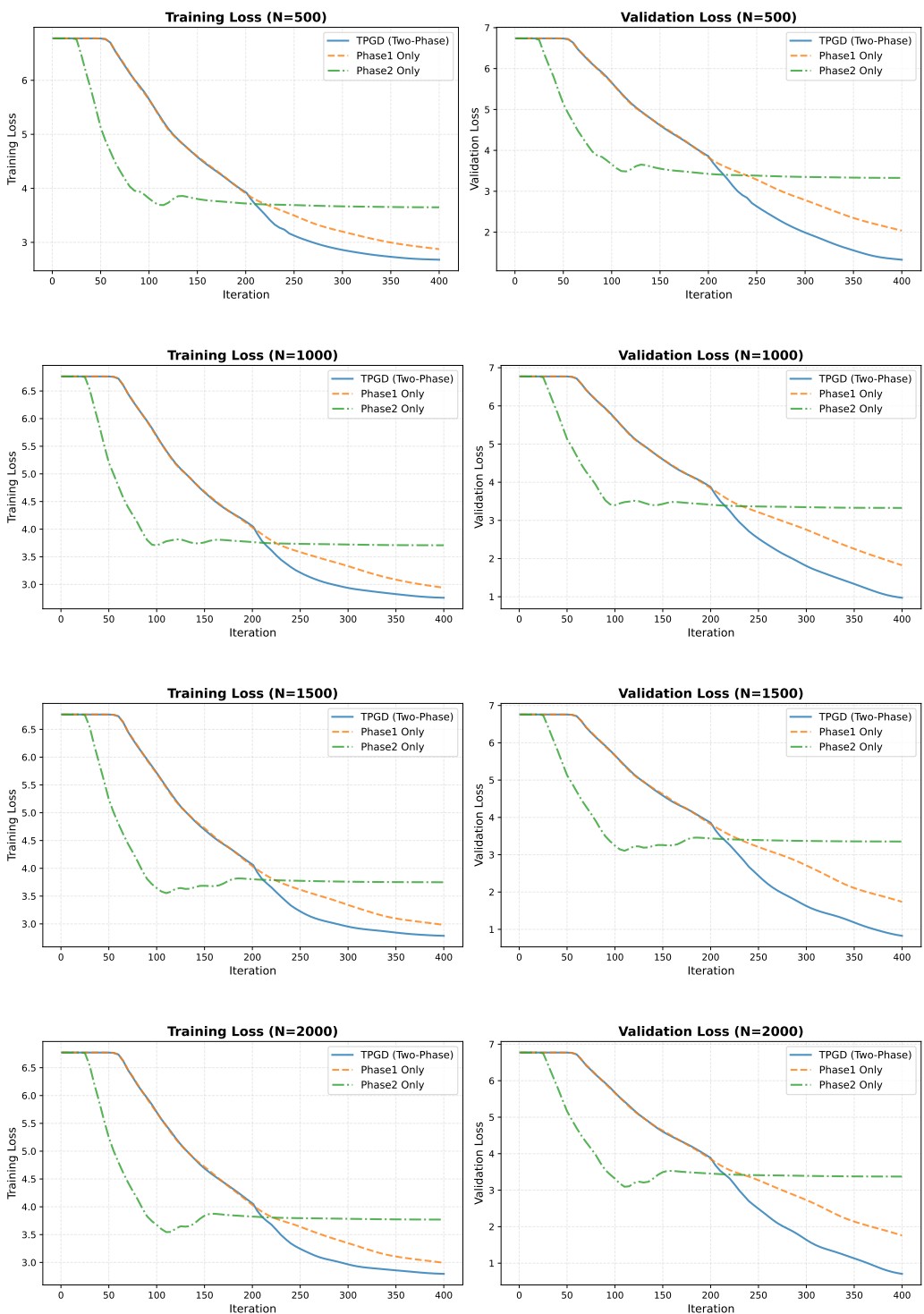

*Figure 3.* Comparison of training and validation loss trajectories for different sample sizes $N$ ($500, 1000, 1500, 2000$) under $d = 70$, $k = 70$, $T = 191$. The ablation contrasts TPGD (two-phase approach) against using only *Phase I* or only *Phase II*, illustrating the contribution of each phase to convergence behavior.

*Table 2.* Comparison of the number of iterations ($N = 1000, d = 100$) required for the loss to stabilize across different values of $T$ (from 100 to 500), with results averaged over three experiments. For each method—TPGD, AltminGD (Thekumparampil et al., 2021), and GD+SVD—the table reports both the iteration count and the corresponding loss value at stabilization.

| Method | $T = 100$ | $T = 200$ | $T = 300$ | $T = 400$ | $T = 500$ |
|---|---|---|---|---|---|
| AltminGD (with QR) (Thekumparampil et al., 2021) | Iter: 1662 Loss: 3.28 | Iter: 1826 Loss: 4.07 | Iter: 1896 Loss: 9.65 | Iter: 1954 Loss: 4.72 | Iter: 1966 Loss: 10.54 |
| GD + SVD | Iter: 476 Loss: 0.69 | Iter: 456 Loss: 0.96 | Iter: 455 Loss: 1.66 | Iter: 407 Loss: 2.25 | Iter: 444 Loss: 2.07 |
| **TPGD** **(This Work)** | **Iter: 365** **Loss: 0.14** | **Iter: 226** **Loss: 0.24** | **Iter: 276** **Loss: 0.32** | **Iter: 163** **Loss: 0.39** | **Iter: 176** **Loss: 0.47** |

# D. Auxiliary Lemmas

**Lemma D.1.** *[Lemma 7.3 (1) in Stöger & Soltanolkotabi (2021)] Suppose matrix $\mathbf{X}/\sqrt{N} \in \mathbb{R}^{d \times N}$ satisfies the $\delta$-RIP. Then, for any vector $\mathbf{u}, \mathbf{v} \in \mathbb{R}^d$, we have*

$$\left| \frac{1}{N} \left\langle \mathbf{X}^\top \mathbf{u}, \mathbf{X}^\top \mathbf{v} \right\rangle - \left\langle \mathbf{u}, \mathbf{v} \right\rangle \right| \leq \delta \left\| \mathbf{u} \right\| \left\| \mathbf{v} \right\|.$$

