# OpenReview forum: "Near-Optimal and Efficient First-Order Algorithm for Multi-Task Learning with Shared Linear Representation"
_ICML.cc/2026/Conference — ICML 2026 regular_

### Official Review · Reviewer_V8dx · 2026-03-09

[review text omitted: it was posted to a different submission]

---

> ### Author Rebuttal · Authors · 2026-03-25
>
> Dear Reviewer,
>
> Thank you very much for your time and effort in reviewing our manuscript.
>
> We would like to respectfully bring to your attention that your comments appear to be for a different manuscript. Your feedback refers to a method called "ConsisNovo" — a multi-view consistency learning framework for retrieval-augmented de novo peptide sequencing — and includes suggestions regarding full-spectrum teacher, masked-spectrum student, retrieval-augmented fusion view, masking ratios, and spectral library retrieval.
>
> In contrast, our paper focuses on **multi-task learning with shared linear representations**, where we propose a first-order algorithm with theoretical guarantees on computational efficiency and near-optimal estimation error. Our work does not involve peptide sequencing, spectral data, or the methodology described in your comments.
>
> We kindly ask if you might be able to review our manuscript based on its actual content. We greatly appreciate your time and assistance.
>
> Best regards,
>
> Authors

---

> > ### Author Rebuttal · Reviewer_V8dx · 2026-03-31
> >
> > This paper proposes a first-order algorithm that can simultaneously learn shared representations and task-specific parameters while ensuring high efficiency. Notably, the algorithm converges in only O(1) iterations and yields approximately optimal estimation results.
> > 1.The introduction section of the paper needs to highlight its innovativeness. For instance, designing an LBM with high computational efficiency and statistical optimality remains an unsolved challenge, even for linear representation settings. In what aspects is this challenge reflected?
> > 2.Although the paper provides a demonstration of the proposed algorithm, it is necessary to further derive the algorithm’s complexity and convergence properties.
> > 3.However, the experimental section is overly simplistic, as many experiments fail to verify the algorithm’s effectiveness. For example, the experiments only verify the convergence rates of TPGD compared with the baseline algorithm with a fixed number of task samples N, as well as the convergence rates of PGD compared with the baseline algorithm with a fixed total number of iterations.

---

> > > ### Author Response · Authors · 2026-04-02
> > >
> > > We sincerely thank the reviewer for the constructive follow-up questions. We address each point below and will incorporate the suggested improvements in the final version.
> > >
> > > ## 1. Highlighting Innovativeness
> > >
> > > In the final version, we will strengthen the introduction to more clearly emphasize the key novelty of our work. In particular, we will explicitly highlight that designing a likelihood-based method (LBM) that is both computationally efficient and statistically optimal remains an open problem—even in the linear representation setting. Our work resolves this gap by proposing the first first-order LBM that: achieves $\tilde O(1)$ iteration complexity, and attains the near-optimal estimation error $\tilde O(dk/(NT))$, matching the known lower bound up to logarithmic factors.
> > >
> > > ## 2. On the Core Challenges
> > > The difficulty of this problem stems from two fundamental challenges:
> > >
> > > - **Challenge 1: Computational inefficiency of LBM under non-convex setting.** Likelihood-based formulations lead to a non-convex matrix sensing problem, where jointly optimizing $(B,W)$ is highly nontrivial. Prior LBM approaches typically: require polynomial iteration complexity in $d$, or rely on carefully designed initialization. Our key technical contribution is to show that a simple first-order method can simultaneously drive both representation and task-specific parameters toward the global optimum, via a carefully designed two-phase gradient scheme. This enables: escaping the non-convex region efficiently (Phase I), and achieving fast local convergence (Phase II). This simultaneous convergence is the key to reducing iteration complexity to $\tilde O(1)$.
> > >
> > > - **Challenge 2: Achieving statistical optimality without auxiliary spectral operations.** Another major challenge is to ensure that: the learned representation $\hat B$ is statistically optimal, and it can be directly used for downstream transfer learning. Existing statistically optimal methods (e.g., moment-based or spectral methods) rely on recovering the principal subspace of $B^\star$, typically requiring SVD/QR operations. In contrast, prior LBM approaches: fail to achieve optimal rates, and require additional post-processing steps to ensure transferability. Our algorithm avoids this entirely. Specifically: the proposed TPGD directly achieves the optimal estimation error, and the learned $\hat B$ is immediately usable for transfer learning, without any additional decomposition or projection. To the best of our knowledge, this is the first LBM that simultaneously achieves computational efficiency, statistical optimality, and transferability without auxiliary operations.
> > >
> > > ## 3. Algorithmic Complexity and Convergence Properties
> > >
> > > We thank the reviewer for this important suggestion. The complexity and convergence guarantees are formally established in the appendix (Theorems A.1 and A.5), and we will make them more explicit in the main text (e.g., Theorem 5.1) in the final version.
> > >
> > > Phase I (Theorem A.1):
> > > Starting from random initialization, the algorithm reaches a constant-radius neighborhood of the global optimum within $\tilde O(1)$ iterations, with sufficient samples.
> > >
> > > Phase II (Theorem A.5):
> > > We establish a linear convergence rate for the optimization error:
> > > $$
> > > \mathrm{Opt Error}^{(K_1)}\leq\left(1-\frac{\sigma_k(\Sigma^\star)}{8}\eta_2\right)^{K_1/2}\mathrm{Opt Error}^{(K_1/2)}.
> > > $$
> > > This shows that once entering the local regime, the algorithm enjoys geometric convergence.
> > >
> > > We will revise the paper to clearly summarize these results in Theorem 5.1 and provide additional discussion on their implications.
> > >
> > > ## 4. Experimental Evaluation (Strengthened Plan)
> > >
> > > We appreciate the reviewer’s insightful comment. We will expand the experiments in the appendix of the final version.
> > >
> > > Specifically, we will include the following additional experiments:
> > >
> > > - **Ablation on algorithm design.** Our supplementary ablation study shows that using only Phase I slows convergence, whereas using only Phase II increases the final loss. These results demonstrate that both components of TPGD are essential for fast convergence and optimal estimation error. We will include the complete set of experimental figures in the final version of the manuscript.
> > >
> > > - **Efficiency evaluation.** We compare the number of iterations required for the loss to stabilize between TPGD and approaches based on SVD/QR decompositions:
> > >
> > > | Method | $T=100$ | $T=200$ | $T=300$ | $T=400$ | $T=500$ |
> > > |:------:|:-------:|:-------:|:-------:|:-------:|:-------:|
> > > |AltminGD (with QR)| Iter: 1662 & Loss: 3.28| Iter: 1826 & Loss: 4.07| Iter: 1896 & Loss: 9.65| Iter: 1954 & Loss: 4.72| Iter: 1966 & Loss: 10.54|
> > > |GD+SVD| Iter: 476 & Loss: 0.69| Iter: 456 & Loss: 0.96| Iter: 455 & Loss: 1.66| Iter: 407 & Loss: 2.25| Iter: 444 & Loss: 2.07|
> > > |**TPGD**| **Iter: 365 & Loss: 0.14**| **Iter: 226 & Loss: 0.24** | **Iter: 276 & Loss: 0.32**| **Iter: 163 & Loss: 0.39** | **Iter: 176 & Loss: 0.47** |

---

### Official Review · Reviewer_Fqu5 · 2026-03-12

**Soundness:** 2
**Presentation:** 2
**Significance:** 2
**Originality:** 2
**Overall Recommendation:** 4
**Confidence:** 3

**Summary:**

This paper studies likelihood-based optimization for linear multi-task learning with a shared low-rank representation. The main claim is that the proposed two-phase gradient descent method (TPGD) is the first first-order likelihood-based method that attains the near-optimal population loss rate
$\widetilde O\left(\frac{dk}{NT}\right)$, improving over the
$\widetilde O\left(\frac{dk^2}{NT}\right)$
rate of prior likelihood-based methods. The algorithm combines a warm-start Phase I on the plain likelihood with a Phase II that adds the balancing penalty
$||B^\top B - WW^\top||_F^2$.
The paper also includes a transfer-learning result based on the learned representation.

**Compliance With Llm Reviewing Policy:**

Affirmed.

**Ethical Review Concerns:**

After checking other reviewers' comments and reevaluating the experiments, I now view this as a theoretically-motivated paper. The experimental results support the theory very well. Therefore, I would like to raise my score.

**Final Justification:**

After checking other reviewers' comments and reevaluating the experiments, I now view this as a theoretically-motivated paper. The experimental results support the theory very well. Therefore, I would like to raise my score.

**Key Questions For Authors:**

1. Can the authors explain much more directly where the improvement from
   $\widetilde O\left(\frac{dk^2}{NT}\right)$
   to
   $\widetilde O\left(\frac{dk}{NT}\right)$
   comes from?

2. Can the authors highlight the main technical novelty in the proof?

**Limitations:**

Representation learning is a big topic in MTL and has been studied a lot. The current framework is still limited. It would be interesting for the authors to discuss whether their ideas could extend to more general settings, for example when the representation is only partially shared and partially task-specific, or when the representations across tasks are only similar.

**Strengths And Weaknesses:**

Strengths

1. The paper tackles an important open problem: obtaining both computational efficiency and near-optimal statistical error for likelihood-based linear MTL.

2. The algorithm is appealing: it stays within first-order optimization and avoids extra SVD/QR procedures, unlike some competing approaches.

Weaknesses

1. The main technical contribution is not explained clearly enough.
   The key improvement is the jump from
   $\widetilde O\left(\frac{dk^2}{NT}\right)$
   to
   $\widetilde O\left(\frac{dk}{NT}\right)$,
   but the paper does not explain enough where this gain really comes from.

2. The paper should explain the novelty relative to prior work, especially Tian et al. (2025). Table 1 is helpful, but the discussion is still not enough. In particular, Tian et al. (2025) provides a more general framework on the spectral side. So the authors need to be much more explicit about what is genuinely new here.

3. There is not enough ablation or discussion isolating the roles of Phase I and Phase II. Also, it's unclear how to tune $\eta_1$ and $\eta_2$ in practice.

---

> ### Author Rebuttal · Authors · 2026-03-31
>
> # Response to Reviewer Fqu5
>
> We thank the reviewer for the insightful feedback and constructive suggestions. We address all concerns point by point.
>
> ## 1. Main technical contribution
>
> Our key result improves the error rate from $\tilde O(dk^2/(NT))$ to $\tilde O(dk/(NT))$ via simultaneous learning of the representation $B*$ and task-specific parameter $W*$ under rotational invariance, enabled by a carefully designed two-phase gradient method.
>
> Phase I (feature learning). We analyze projected variables $\tilde B,\tilde W^\top$ (defined in Eq. (10)–(11)) in the subspace spanned by $U*,V*$. Using Lemma B.1 to control stochastic noise and a block-wise analysis of updates, we show key quantities (e.g., $||\tilde B_{[k],[k]}-\tilde W_{[k],[k]}^{\top}||$) remain small. Combined with Lemma B.2, this implies $(B,W^\top)$ converges to a neighborhood of $(F,G)$. This simultaneous convergence via gradient methods is the core novelty, a capability not achieved in prior multi-task learning studies.
>
> Phase II (refinement). We prove linear convergence via a Regularity Condition (Def. A.2). The error $\sum_t||v_t^{(\tau)}-v_t*||^2/T$ is controlled by $dist(Z^{(\tau)},J)^2/T$, which decomposes into a linear decay term dependent on the step size $\eta_2$ and a statistical term $\eta_2\tilde O(dk/(NT))$. After $K_1/2$ iterations, we obtain $\tilde O(dk/(NT))$.
>
> Comparison. Prior likelihood-based methods (e.g., Du et al., Tian et al.) either lack regularization or only regularize $B$. In Du et al., the error $||\hat B\hat w_t-B* w_t*||$ is dominated by $||P_{X_t\hat B}X_tB*||_F$. Their upper bound for this term is $\tilde O(dk/(N\sigma_k^2(W*)))$ and their additional Assumption 4.3 implies $1/(\sigma_k^2(W*))\lesssim k/T$. Hence, their ERM method yields an error of $\tilde O(dk^2/(NT))$ in the multi-task setting. Under the setting where all tasks share the same representation, the pERM method by Tian et al. is similar to the ERM approach of Du et al., and thus achieves the same $\tilde O(dk^2/(NT))$ error.
>
> ## 2. Novelty vs. Tian et al.
> We appreciate the reviewer’s insightful question. The difference and novelty are summarized from three points.
> - Setting. Tian et al. study heterogeneous/corrupted tasks, focusing on adaptivity and robustness of algorithms. We consider the classical shared low-rank setting, targeting **computational tractability** and **optimal estimation error**. Our algorithm can serve as an efficient and optimal baseline for future extensions (see **Q6** for extensions).
> - Algorithm novelty. We propose the first pure first-order two-phase GD (TPGD) with known best iteration complexity. Prior likelihood-based methods either have intractable iteration complexity, require specific initialization  (e.g., starting in a neighborhood of the optimum) or additional constraints (e.g., orthogonality), or involve extra costs such as SVD or QR decomposition during iterations.
> - Technique. Our key novelty is proving rapidly global convergence from random initialization via a two-stage design (warm-up + regularized refinement) and a new analysis technique: establishing the simultaneous convergence of $B$ and $W$ to the optimum under the $dist$ metric by combining a block-wise analysis of the iterative dynamics of the representation parameter $B$ and task-specific parameter $W$ with a quantification of their coupling effect. Detailed technical explanations are provided in our response to the first comment.
>
> ## 3. Role of Phase I vs Phase II
> Phase I: learns features of $B*$ and $W*$, and provides a warm start near $(F,G)$.
>
> Phase II: uses regularization to decouple $B,W$ and achieve linear convergence without inflating statistical error.
>
> Empirically (Fig. 1), Phase II clearly accelerates convergence.
>
> ## 4. Step sizes $\eta_1,\eta_2$
>
> Theorem 5.1 allows both to be small constants. In practice, setting $\eta_1\geq\eta_2$ works robustly; performance is insensitive as long as they are sufficiently small.
>
> ## 5. Technical novelty in proof
>
> The main novelty lies in establishing simultaneous convergence of $B$ and $W$ under rotational invariance, via coupling-aware analysis and the two-phase framework (see **Q1–Q2**). We will clarify this in the next version.
>
> ## 6. Extensions
>
> As noted above, our paper focuses on a classical and fundamental setting. We are interested in extending our framework to the settings considered by Tian et al. where representations are only partially shared and partially task-specific or only similar across tasks, via a two-stage approach:
>
> Stage I: learn a central representation
> (i) via regularized joint loss $\sum_t L_t(B_tw_t)+\gamma_1\sum_t||B_t^\top B_t-WW^\top||_F^2$, or
> (ii) reuse Phase I to obtain $\bar B$.
>
> Stage II: task-specific fine-tuning
> $L_t(v_t)+\gamma_2||v_t-B_tw_t||^2$ or $L_t(v_t)+\gamma_2||v_t-\bar B w_t||^2$.
>
> This step captures task-specific deviations from the central representation, and retains robustness guarantees similar to Tian et al.

---

> > ### Author Rebuttal · Reviewer_Fqu5 · 2026-04-03
> >
> > Thanks for the response. While the two-stage approach is clear, I remain concerned that the overall contribution is somewhat limited. The field of shared representation is already very mature and highly competitive. Given the current setting, I am not convinced that the proposed model provides a significant advancement over existing literature. I will re-evaluate the paper's impact in light of the other reviewers' comments.

---

> > > ### Author Response · Authors · 2026-04-05
> > >
> > > We sincerely thank the reviewer for the thoughtful feedback and for raising the important question regarding the overall impact.
> > >
> > > We agree that the literature on shared representation learning is well-established and highly competitive. However, we would like to clarify that our contribution addresses a fundamental gap that remains largely unresolved in prior work.
> > >
> > > As highlighted in our paper, existing theoretical advances in multi-task learning predominantly focus on statistical optimality, achieved via moment-based or spectral methods. While these approaches attain optimal estimation error, they typically rely on specialized procedures (e.g., SVD) and do not reflect the first-order optimization paradigm that is most widely used in modern machine learning practice.
> > >
> > > In contrast, understanding whether likelihood-based first-order methods can simultaneously achieve computational efficiency and statistical optimality has remained an open question, even in the linear representation setting. Prior likelihood-based approaches either suffer from suboptimal error rates or require strong assumptions. Moreover, even in closely related problems such as matrix sensing, achieving both optimal sample complexity and fast first-order convergence remains challenging.
> > >
> > > Our work makes progress on this question by showing that a simple first-order algorithm can, in fact, achieve:
> > >
> > > - Near-optimal estimation error $\tilde O(dk/(NT))$, matching the information-theoretic lower bound.
> > > - Fast convergence with only $\tilde O(1)$ iterations, without relying on additional operations such as SVD or QR decomposition.
> > >
> > > We believe this result is conceptually important, as it helps bridge the gap between **statistical optimality** and **computational tractability** in multi-task representation learning, and provides theoretical justification for the empirical success of first-order methods in related non-convex problems.
> > >
> > > More broadly, our analysis offers new insights into the feature learning dynamics of gradient-based methods (via the two-phase design), which may serve as a foundation for studying more complex shared representation models.
> > >
> > > We hope this clarification helps better position our contribution, and we appreciate the reviewer’s consideration in the final evaluation.

---

### Official Review · Reviewer_L9aq · 2026-03-13

**Soundness:** 4
**Presentation:** 3
**Significance:** 3
**Originality:** 3
**Overall Recommendation:** 5
**Confidence:** 3

**Summary:**

The authors provide a first-order algorithm for learning a linear representation shared between tasks.
Their algorithm learns the representation and task predictors jointly. The algorithm is non-convex i.e. on a task level bi-linear. So, as common in this type of work, they need a good initial starting location to converge to a global minima. Using this initialization obtained from SVD their algorithm in two stages. First, it does unregularized GD on the negative log-likelihood. Their algorithm, in contrast to prior work can use constant step sizes to converge to a small neighborhood of the global minima in $\widetilde{\mathcal{O}}(1)$ iterations. Second, a regularization correction step is performed to disentangle coupling from the first stage.

For their algorithm, it is the first that jointly learns the representation and task parameters with guaranteed computational and statistical efficiency. Furthermore the computational rates are statistically efficient. The work also removes a $k$ factor that is present in prior work for likelihood-based methods. The authors also extend their work to curriculum learning. This is done under Gaussian noise and the restricted isometry property common in compressed sensing (which is natural given the connection between the two subject areas).

**Compliance With Llm Reviewing Policy:**

Affirmed.

**Key Questions For Authors:**

* Can the authors clarify the remark I made above about L.139.
* At R.197 how are we numerically computing the solution? It's in Expectation.
* In my understanding L.315 is a statement about the learning of $B$ and $W$ as compared to the $U$ and $V$. Thus we have learning up to $U$ and $V$ modulo an intermediate diagonalization matrix, which is already injected (at least in a maximal sense) by the initialization point, and therefore learning the entire end-to-end predictor (factored). Is my understanding correct?
* Please provide details on the numerical simulations.

**Limitations:**

The authors should note the circularity in the initial starting point (which as I point out is common in this type of work).

**Strengths And Weaknesses:**

**STRENGTH**
* The paper gives a novel algorithm with significant novel rates for likelihood-based methods.
* The assumptions are standard.
* The allowance of constant step size seems to be a significant benefit.
* The relaxing of the RIP condition in the setting to be less than $k^{-1/2}$ versus $k^{-1}$ is significant in reducing dimensional dependence and may extend to other work.
* The work also gives weaker moment conditions in transfer learning in compared to prior work.
* The paper is well written. I've put some editing remarks below.

**WEAKNESSES**
* I found the Numerical Simulations section confusing and lacking of details. To my eye the task or dataset is not specified. Without such details it's difficult to determine the quality of the empirical work. More details in the appendix at least are necessary.


**minor remarks**
* I was confused after reading L.110 as it's not factored. This of course doesn't match with the actual method of learning the factored representation and predictor separately as the authors point out later.
* When the gradient updates are introduced at L. 135 a forward reference to R.272 may be appropriate, else the reader doesn't need to backwards construct the objective..
* At L.139: "Consequently, the proposed algorithm has an iteration complexity of only $\widetilde{\mathcal{O}}(1)$" doesn't to my eye follow from the above.
* L.192 "There exists linear MTL study leverage this connection."
* A space after "In contrast,l" on the bottom of page 2.
* Should introduce LBM in the main body.

---

> ### Author Rebuttal · Authors · 2026-03-31
>
> # Response to Reviewer L9aq:
> We sincerely thank Reviewer L9aq for the constructive and thoughtful feedback, and for recognizing our work. Below we respond point by point.
>
> ## 1. Minor remarks.
> We apologize for typos due to unclear presentation or writing errors, and thank the reviewer for identifying them. All typos and presentation issues (e.g., L.110, L.135, L.192, spacing, and LBM introduction) will be corrected for clarity and readability.
>
> ## 2. More discussions for L.110.
> We apologize for not providing a more detailed explanation on how task predictors $[\hat v_t]_{t\in[T]}$ are obtained, due to an oversight in writing. We thank the reviewer for pointing out this source of confusion.
> At L.110, the predictor $\hat v_t$ should indeed be understood as the **factorized** predictor constructed from the learned representation and task-specific parameter,
> namely $\hat v_t = \hat B \hat w_t$, where $\hat w_t$ denotes the $t$-th column vector of $\hat W$. The convergence guarantees are given in Corollary 5.3. The confusion arises because the factorized structure
> is not explicitly stated at that point in the text. We will add a clarification in the revised manuscript—both at L.110 and after Theorem 5.1—on how the learned $\hat B$ and $\hat W$ are used to construct the predictor $\hat v_t$ for each task $t$.
>
> ## 3. Remark in L.139.
> We sincerely thank the reviewer for the careful observation.
> More precisely, Theorem 5.1 shows that the total number of iterations satisfies $K_1 \geq \tilde \Omega(\kappa^6(\Sigma*))$. Thus, the iteration complexity depends polynomially on the condition number.
>
> In the discussion at the end of Section 3, we adopt the standard convention in the literature of treating $\kappa(\Sigma*)$ as a problem-dependent constant, which leads to the simplified statement $\tilde O(1)$.
>
> We agree that this simplification was not sufficiently justified in the introduction. In the next revision, we will (i) explicitly state the dependence on $\kappa(\Sigma*)$,
> and (ii) clarify under which regimes this dependence can be regarded as constant.
>
> ## 4. Numerically computing the solution at R.197.
> The reviewer raises an important point regarding practical implementation. When optimizing the loss function $L_{\rho}$ using Algorithm 2, at each iteration $\tau$, we sample a single data $(x_{T+1}^{(\tau)}, y_{T+1}^{(\tau)})$ and update $w_{T+1}$ using stochastic gradient descent. This process does not require exact computation of the expectation $\mathbb{E}[\cdot]$, thus maintaining computational efficiency.
>
> ## 5. Statement at L.315.
> We thank the reviewer for this insightful question, and their understanding is essentially correct.
>
> Due to the non-identifiability of matrix factorization,
> the learned parameters $(\hat B, \hat W)$ are only identifiable up to an invertible transformation $P\in\mathbb{R}^{k\times k}$. Specifically, for any invertible matrix $P$, $(\hat{B}, \hat{W})$ and $(\hat B P, P^{-1}\hat W)$
> represent the same end-to-end predictor. Our theory guarantees that the product $\hat B\hat W$ converges to the ground-truth predictor $B* W*$ at the optimal rate, regardless of this transformation ambiguity.
>
> Importantly, this implies that the learned representation $\hat B$ remains sufficient for downstream adaptation:
> $\hat w_{T+1}$ recovers $w_{T+1}*$ up to the same transformation $P^{-1}$.
>
> ## 6. Details on the numerical simulations.
> We consider a linear representation model $\phi(x)=B*^\top x$ with $B*\in\mathbb{R}^{d\times k}$. Data are generated as
> $$
> y_t = \langle w_t*, \phi(x_t) \rangle + z_t = \langle w_t*, B*^\top x_t\rangle  + z_t, \quad z_t \sim \mathcal{N}(0, \sigma^2),
> $$
> For each task $t\in[T]$, we sample $N$ training and $N_{test}$ test points. Inputs are i.i.d. $x_{t,i}\sim\mathcal{N}(0,x_{std}^2I_d)$. Clean labels follow $y_{t,i}^{clean}=\langle w_t*, B*^\top x_{t,i}\rangle$; training labels include noise $z_{t,i}\sim\mathcal{N}(0,\sigma^2)$, while test labels are noise-free.
> We use four settings:
>
> - para_0: $d=50,k=5,T=50,N=500,N_{test}=20,\sigma=0.4$;
> - para_1: $d=100,k=10,T=100,N=1000,N_{test}=20,\sigma=0.4$;
> - para_2: $d=200,k=20,T=200,N=1000,N_{test}=20,\sigma=0.2$;
> - para_3: $d=400,k=50,T=400,N=2000,N_{test}=20,\sigma=0.1$.
>
> Hyperparameters follow our theory. We estimate the condition number $\kappa$ from sampled $B,W$ (or conservatively upper-bound it in practice). Learning rates are
> $$
> \eta_1 = \min\left[\frac{c_1}{\kappa^5 \sigma_1}, 0.1\right], \quad \eta_2 = \min\left[\frac{c_2}{\sigma_1}, 0.1\right],
> $$
> with $c_1=1$ and $c_2=1$ (0.1 for para_0). The Phase I iteration count is $K_1=c_3/(\eta_1\sigma_k)$, where $c_3$ decreases from 5 to 2 across settings to reduce computation in high dimensions while preserving convergence. Empirically, Phase II converges rapidly to near-optimality. All comparative algorithms use the recommended or default hyperparameters from their original publications. We will provide detailed descriptions of these settings in the appendix.

---

> > ### Author Rebuttal · Reviewer_L9aq · 2026-04-04
> >
> > I thank the authors for their thoughtful reply.
> >
> > Reading over the reply my concerns are addressed. I'll keep an eye on the other reviews for further discussion and maintain my score given current information.

---

> > > ### Author Response · Authors · 2026-04-06
> > >
> > > Thank you again for your thoughtful response and recognition. We are very glad that our rebuttal has addressed your concerns. If you have any further questions, please don't hesitate to let us know—we would be happy to provide additional clarification.

---

### Decision · Program_Chairs · 2026-04-30

**Decision:**

Accept (regular)

**Comment:**

This paper studies linear multi-task learning and proposes a two-phase first-order method that achieves a near-optimal ~O(dk/(NT)) error rate without SVD or QR steps.
Reviewers viewed the paper as technically strong. They also asked for a clearer explanation of the main contribution and of the novelty relative to prior work.
The main concerns were the limited empirical scope and the need to explain what each phase contributes. The rebuttal made these points clearer. The empirical support still remains limited to simulations.
The paper makes a solid theoretical contribution. I therefore recommend acceptance.